# Learning protein constitutive motifs from sequence data

**Jérôme Tubiana, Simona Cocco, Rémi Monasson\***

Laboratory of Physics of the Ecole Normale Supérieure, CNRS UMR 8023 & PSL Research, Paris, France

**Abstract** Statistical analysis of evolutionary-related protein sequences provides information about their structure, function, and history. We show that Restricted Boltzmann Machines (RBM), designed to learn complex high-dimensional data and their statistical features, can efficiently model protein families from sequence information. We here apply RBM to 20 protein families, and present detailed results for two short protein domains (Kunitz and WW), one long chaperone protein (Hsp70), and synthetic lattice proteins for benchmarking. The features inferred by the RBM are biologically interpretable: they are related to structure (residue-residue tertiary contacts, extended secondary motifs ($\alpha$-helixes and $\beta$-sheets) and intrinsically disordered regions), to function (activity and ligand specificity), or to phylogenetic identity. In addition, we use RBM to design new protein sequences with putative properties by composing and 'turning up' or 'turning down' the different modes at will. Our work therefore shows that RBM are versatile and practical tools that can be used to unveil and exploit the genotype–phenotype relationship for protein families.

DOI: https://doi.org/10.7554/eLife.39397.001

## Introduction

In recent years, the sequencing of many organisms' genomes has led to the collection of a huge number of protein sequences, which are catalogued in databases such as UniProt or PFAM *Finn et al., 2014*). Sequences that share a common ancestral origin, defining a family (*Figure 1A*), are likely to code for proteins with similar functions and structures, providing a unique window into the relationship between genotype (sequence content) and phenotype (biological features). In this context, various approaches have been introduced to infer protein properties from sequence data statistics, in particular amino-acid conservation and coevolution (correlation) (*Teppa et al., 2012*; *de Juan et al., 2013*).

A major objective of these approaches is to identify positions carrying amino acids that have critical impact on the protein function, such as catalytic sites, binding sites, or specificity-determining sites that control ligand specificity. Principal component analysis (PCA) of the sequence data can be used to unveil groups of coevolving sites that have a specific functional role *Russ et al., 2005*; *Rausell et al., 2010*; *Halabi et al., 2009*. Other methods rely on phylogeny *Rojas et al., 2012*, entropy (variability in amino-acid content) *Reva et al., 2007*, or a hybrid combination of both *Mihalek et al., 2004*; *Ashkenazy et al., 2016*.

Another objective is to extract structural information, such as the contact map of the three-dimensional fold. Considerable progress was brought by maximum-entropy methods, which rely on the computation of direct couplings between sites reproducing the pairwise coevolution statistics in the sequence data *Lapedes et al., 1999*; *Weigt et al., 2009*; *Jones et al., 2012*; *Cocco et al., 2018*. Direct couplings provide very good estimators of contacts *Morcos et al., 2011*; *Hopf et al., 2012*; *Kamisetty et al., 2013*; *Ekeberg et al., 2014* and capture the pairwise epistasis effects necessary to model the fitness changes that result from mutations *Mann et al., 2014*; *Figliuzzi et al., 2016*; *Hopf et al., 2017*.

**\*For correspondence:** monasson@lpt.ens.fr

**Competing interests:** The authors declare that no competing interests exist.

**eLife digest** Almost every process that keeps a cell alive depends on the activities of several proteins. All proteins are made from chains of smaller molecules called amino acids, and the specific sequence of amino acids determines the protein's overall shape, which in turn controls what the protein can do. Yet, the relationships between a protein's structure and its function are complex, and it remains unclear how the sequence of amino acids in a protein actually determine its features and properties.

Machine learning is a computational approach that is often applied to understand complex issues in biology. It uses computer algorithms to spot statistical patterns in large amounts of data and, after 'learning' from the data, the algorithms can then provide new insights, make predictions or even generate new data.

Tubiana et al. have now used a relatively simple form of machine learning to study the amino acid sequences of 20 different families of proteins. First, frameworks of algorithms –known as Restricted Boltzmann Machines, RBM for short – were trained to read some amino-acid sequence data that coded for similar proteins. After 'learning' from the data, the RBM could then infer statistical patterns that were common to the sequences. Tubiana et al. saw that many of these inferred patterns could be interpreted in a meaningful way and related to properties of the proteins. For example, some were related to known twists and loops that are commonly found in proteins; others could be linked to specific activities. This level of interpretability is somewhat at odds with the results of other common methods used in machine learning, which tend to behave more like a 'black box'.

Using their RBM, Tubiana et al. then proposed how to design new proteins that may prove to have interesting features. In the future, similar methods could be applied across computational biology as a way to make sense of complex data in an understandable way.

DOI: https://doi.org/10.7554/eLife.39397.002

Despite these successes, we still do not have a unique, accurate framework that is capable of extracting the structural and functional features common to a protein family, as well as the phylogenetic variations specific to sub-families. Hereafter, we consider Restricted Boltzmann Machines (RBM) for this purpose. RBM are a powerful concept coming from machine learning *Ackley et al., 1987*; *Hinton, 2012*; they are unsupervised (sequence data need not be annotated) and generative (able to generate new data). Informally speaking, RBM learn complex data distributions through their statistical features (*Figure 1B*).

In the present work, we have developed a method to train efficiently RBM from protein sequence data. To illustrate the power and versatility of RBM, we have applied our approach to the sequence alignments of 20 different protein families. We report the results of our approach, with special emphasis on four families — the Kunitz domain (a protease inhibitor that is historically important for protein structure determination *Ascenzi et al., 2003*, the WW domain (a short module binding different classes of ligands (*Sudol et al., 1995*, Hsp70 (a large chaperone protein *Bukau and Horwich, 1998*), and lattice-protein in silico data *Shakhnovich and Gutin, 1990*; *Mirny and Shakhnovich, 2001* — to benchmark our approach on exactly solvable models *Jacquin et al., 2016*. Our study shows that RBM are able to capture: (1) structure-related features, be they local (such as tertiary contacts), extended such as secondary structure motifs ($\alpha$-helix and $\beta$-sheet)) or characteristic of intrinsically disordered regions; (2) functional features, that is groups of amino acids controling specificity or activity; and (3) phylogenetic features, related to sub-families sharing evolutionary determinants. Some of these features involve only two residues (as direct pairwise couplings do), others extend over large and not necessarily contiguous portions of the sequence (as in collective modes extracted with PCA). The pattern of similarities of each sequence with the inferred features defines a multidimensional representation of this sequence, which is highly informative about the biological properties of the corresponding protein (*Figure 1C*). Focusing on representations of interest allows us, in turn, to design new sequences with putative functional properties. In summary, our work shows that RBM offer an effective computational tool that can be used to characterize and exploit quantitatively the genotype–phenotype relationship that is specific to a protein family.

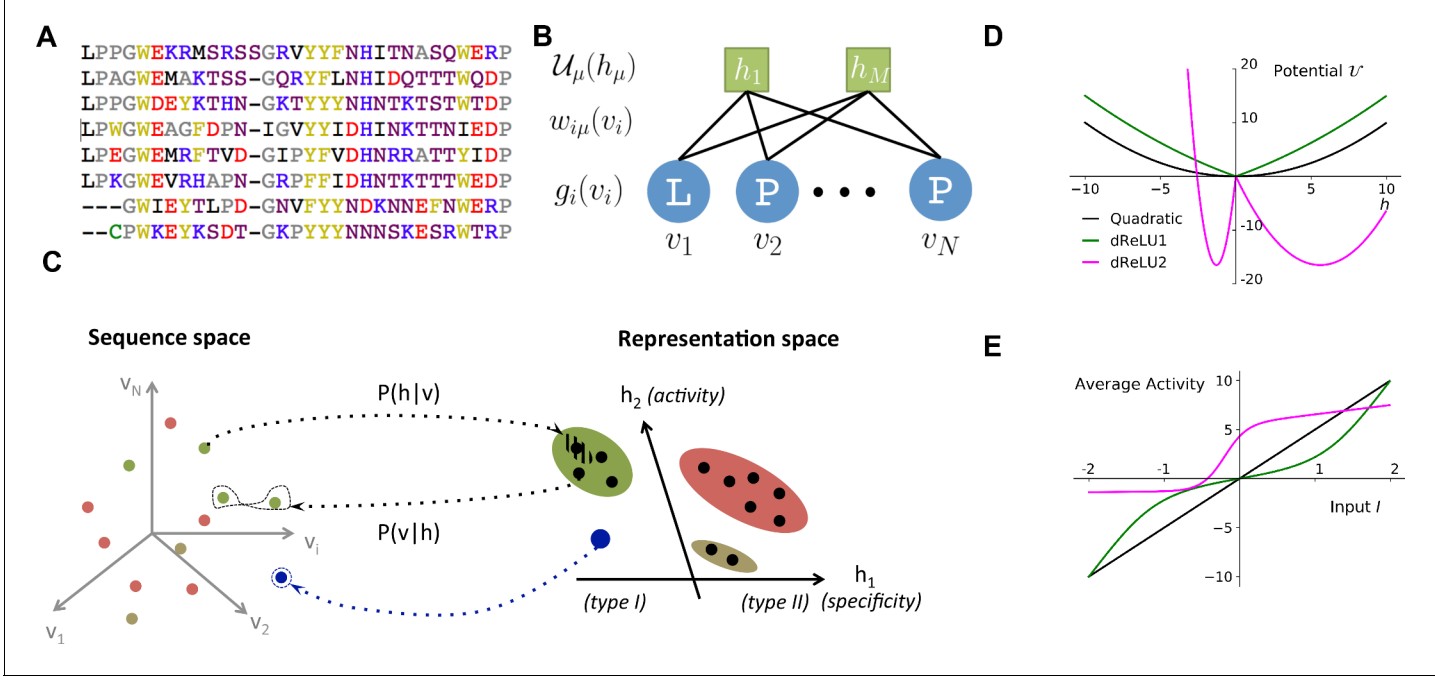

**Figure 1.** Reverse and forward modeling of proteins. (**A**) Example of Multiple-Sequence Alignment (MSA), here of the WW domain (PF00397). Each column $i = 1, ..., N$ corresponds to a site on the protein, and each line to a different sequence in the family. The color code for amino acids is as follows: red = negative charge (E,D), blue = positive charge (H, K, R), purple = non charged polar (hydrophilic) (N, T, S, Q), yellow = aromatic (F, W, Y), black = aliphatic hydrophobic (I, L, M, V), green = cysteine (C), grey = other, small amino acids (A, G, P). (**B**) In a Restricted Boltzmann Machine (RBM), weights $w_{i\mu}$ connect the visible layer (carrying protein sequences $\mathbf{v}$) to the hidden layer (carrying representations $\mathbf{h}$). Biases on the visible and hidden units are introduced by the local potentials $g_i(v_i)$ and $\mathcal{U}_\mu(h_\mu)$. Owing to the bipartite nature of the weight graph, hidden units are conditionally independent given a visible configuration, and vice versa. (**C**) Sequences $\mathbf{v}$ in the MSA (dots in sequence space, left) code for proteins with different phenotypes (dot colors). RBM define a probabilistic mapping from sequences $\mathbf{v}$ onto the representation space $\mathbf{h}$ (right), which is indicative of the phenotype of the corresponding protein and encoded in the conditional distribution $P(\mathbf{h}|\mathbf{v})$, **Equation (3)** (black arrow). The reverse mapping from representations to sequences is $P(\mathbf{v}|\mathbf{h})$, **Equation (4)** (black arrow). In turn, sampling a subspace in the representation space (colored domains) defines a complex subset of the sequence space, and allows the design of sequences with putative phenotypic properties that are either found in the MSA (green circled dots) or not encountered in Nature (arrow out of blue domain). (**D**) Three examples of potentials $\mathcal{U}$ defining the hidden-unit type in RBM (see **Equation (1)** and panel (B)): quadratic (black, $\gamma = 0.2$, $\theta = 0$) and double Rectified Linear Unit (dReLU) (dReLU1 (green), $\gamma_+ = \gamma_- = 0.1$, $\theta_+ = -\theta_- = 1$; and dReLU2 (purple), $\gamma_+ = 1$, $\gamma_- = 20$, $\theta_+ = -6$, $\theta_- = 25$) potentials. In practice, the parameters of the hidden unit potentials are fixed through learning of the sequence data. (**E**) Average activity of hidden unit $h$, calculated from **Equation (3)**, as a function of the input $I$ defined in **Equation (2)**. The three curves correspond to the three choices of potentials in panel (A). For the quadratic potential (black), the average activity is a linear function of $I$. For dReLU1 (green), small inputs $I$ barely activate the hidden unit, whereas dReLU2 (Purple) essentially binarizes the inputs $I$.

DOI: https://doi.org/10.7554/eLife.39397.003

## Results

### Restricted Boltzmann Machines

#### Definition

A Restricted Boltzmann Machine (RBM) is a joint probabilistic model for sequences and representations (see **Figure 1C**). It is formally defined on a bipartite, two-layer graph (**Figure 1B**). Protein sequences $\mathbf{v} = (v_1, v_2, ..., v_N)$ are displayed on the Visible layer, and representations $\mathbf{h} = (h_1, h_2, ..., h_M)$ on the Hidden layer. Each visible unit takes one out of 21 values (20 amino acids + 1 alignment gap). Hidden-layer unit values $h_\mu$ are real. The joint probability distribution of $\mathbf{v}, \mathbf{h}$ is:

$$P(\mathbf{v}, \mathbf{h}) \propto \exp \left( \sum_{i=1}^{N} g_i(v_i) - \sum_{\mu=1}^{M} \mathcal{U}_\mu(h_\mu) + \sum_{i,\mu} h_\mu w_{i\mu}(v_i) \right), \qquad (1)$$

up to a normalization constant. Here, the weight matrix $w_{i\mu}$ couples the visible and the hidden layers,

and $g_i(v_i)$ and $\mathcal{U}_\mu(h_\mu)$ are local potentials biasing the values of, respectively, the visible and the hidden variables (**Figure 1B,D**).

## From sequence to representation, and back

Given a sequence $\mathbf{v}$ on the visible layer, the hidden unit receives the input

$$I_\mu(\mathbf{v}) = \sum_i w_{i\mu}(v_i) \, . \tag{2}$$

This expression is analogous to the score of a sequence with a position-specific weight matrix. Large positive or negative $I_\mu$ values signal a good match between the sequence and, respectively, the positive and the negative components of the weights attached to unit $\mu$, whereas small $|I_\mu|$ values correspond to a bad match.

The input $I_\mu$ determines, in turn, the conditional probability of the activity $h_\mu$ of the hidden unit,

$$P(h_\mu|\mathbf{v}) \propto \exp\left(-\mathcal{U}_\mu(h_\mu) + h_\mu I_\mu(\mathbf{v})\right) \, , \tag{3}$$

up to a normalization constant. The nature of the potential $\mathcal{U}$ is crucial in determining how the average activity $h$ varies with the input $I$ (see **Figure 1E** and below).

In turn, given a representation (set of activities) $\mathbf{h}$ on the hidden layer, the residues on site $i$ are distributed according to:

$$P(v_i|\mathbf{h}) \propto \exp\left(g_i(v_i) + \sum_\mu h_\mu w_{i\mu}(v_i)\right) \, . \tag{4}$$

Hidden units with large activities $h_\mu$ strongly bias this probability, and favor values of $v_i$ corresponding to large weights $w_{i\mu}(v_i)$.

Use of **Equation (3)** allows us to sample the representation space given a sequence, while **Equation (4)** defines the sampling of sequences given a representation (see both directions in **Figure 1C**). Iterating this process generates high-probability representations, which, in turn, produce very likely sequences, and so on.

## Probability of a sequence

The probability of a sequence, $P(\mathbf{v})$, is obtained by summing (integrating) $P(\mathbf{v}, \mathbf{h})$ over all its possible representations $\mathbf{h}$.

$$P(\mathbf{v}) = \int \prod_{\mu=1}^M dh_\mu P(\mathbf{v}, \mathbf{h}) \propto \exp\left[\sum_{i=1}^N g_i(v_i) + \sum_{\mu=1}^M \Gamma_\mu(I_\mu(\mathbf{v}))\right] \, , \tag{5}$$

where $\Gamma_\mu(I) = \log \int dh\, e^{-U_\mu(h) + hI}$ is the cumulant-generating function associated with the potential $\mathcal{U}_\mu$ and is a function of the input to hidden unit $\mu$ (see **Equation (2)**).

For quadratic potentials $\mathcal{U}_\mu(h) = \frac{\gamma_\mu}{2} h^2 + \theta_\mu h$ (**Figure 1E**), the conditional probability $P(h_\mu|\mathbf{v})$ is Gaussian, and the RBM is said to be Gaussian. The cumulant-generating functions $\Gamma_\mu(I) = \frac{1}{2\gamma_\mu}(I - \theta_\mu)^2$ are quadratic, and their sum in **Equation (5)** gives rise to effective pairwise couplings between the visible units, $J_{ij}(v_i, v_j) = \sum_\mu \frac{1}{\gamma_\mu} w_{i\mu}(v_i) w_{j\mu}(v_j)$. Hence, a Gaussian RBM is equivalent to a Hopfield-Potts model **Cocco et al., 2013**, where the number $M$ of hidden units plays the role of the number of Hopfield-Potts 'patterns'.

Non-quadratic potentials $\mathcal{U}_\mu$, and, hence, non-quadratic $\Gamma(I)$, introduce couplings to *all orders* between the visible units, all generated from the weights $w_{i\mu}$. RBM thus offer a practical way to go beyond pairwise models, and express complex, high-order dependencies between residues, based on the inference of a limited number of interaction parameters (controlled by $M$). In practice, for each hidden unit, we consider the class of 4-parameter potentials,

$$\mathcal{U}_\mu(h) = \frac{1}{2}\gamma_{\mu,+} h_+^2 + \frac{1}{2}\gamma_{\mu,-} h_-^2 + \theta_{\mu,+} h_+ + \theta_{\mu,-} h_- \, , \quad \text{where} \quad h_+ = \max(h, 0) \, , \quad h_- = \min(h, 0) \, , \tag{6}$$

hereafter called double Rectified Linear Unit (dReLU) potentials (**Figure 1E**). Varying the parameters

allows us to span a wide class of behaviors, including quadratic potentials, double-well potentials (leading to bimodal distributions for $h_\mu$) and hard constraints (e.g. preventing $h_\mu$ from being negative).

RBM can thus be thought of both as a framework to extract representations from sequences through *Equation (3)*, and as a way to model complex interactions between residues in sequences through *Equation (5)*. They constitute a natural candidate to unify (and improve) PCA-based and direct-coupling-based approaches to protein modeling.

## Learning

The weights $w_{i\mu}$ and the defining parameters of the potentials $g_i$ and $\mathcal{U}_\mu$ are learned by maximizing the average log-probability $\langle \log P(\mathbf{v}) \rangle_{MSA}$ of the sequences $\mathbf{v}$ in the Multiple Sequence Alignment (MSA). In practice, estimating the gradients of the average log-probability with respect to these parameters requires sampling from the model distribution $P(\mathbf{v})$, which is done through Monte Carlo simulation of the RBM (see 'Materials and methods').

We also introduce penalty terms over the weights $w_{i\mu}(v)$ (and the local potentials $g_i(v)$ on visible units) to avoid overfitting and to promote sparse weights. Sparsity facilitates the biological interpretation of weights and, thus, emphasizes the correspondence between representation and phenotypic spaces (*Figure 1C*). Crucially, imposing sparsity also forces the RBM to learn a so-called compositional representation, in which each sequence is characterized by a subset of strongly activated hidden units, which are of size large compared to 1 but small compared to $M$ (*Tubiana and Monasson, 2017*. All technical details about the learning procedure are reported in the 'Materials and methods'.

In the next sections, we present results for selected values of the number of hidden units and of the regularization penalty. The values of these (hyper-)parameters are justified afterwards.

## Kunitz domain

### Description

The majority of natural proteins are obtained by concatenating functional building blocks, called protein domains. The Kunitz domain, with a length of about 50–60 residues (protein family PF00014 *Finn et al., 2014*)) is present in several genes and its main function is to inhibit serine proteases such as trypsin. Kunitz domains play a key role in the regulation of many important processes in the body, such as tissue growth and remodeling, inflammation, body coagulation and fibrinolysis. They are implicated in several diseases, such as tumor growth, Alzheimer's disease, and cardiovascular and inflammatory diseases and, therefore, have been largely studied and shown to have a large potential in drug design *Shigetomi et al., 2010*; *Bajaj et al., 2001*).

Some examples of proteins containing a Kunitz-domain include the Basic Pancreatic Trypsin Inhibitor (BPTI, which has one Kunitz domain), Bikunin (two domains) *Fries and Blom, 2000*, Hepatocyte growth factor activator inhibitor (HAI, two domains) and tissue factor pathway inhibitor (TFPI, three domains) *Shigetomi et al., 2010*; *Bajaj et al., 2001*).

*Figure 2A* shows the MSA sequence logo and the secondary structure of the Kunitz domain. It is characterized by two $\alpha$ helices and two $\beta$ strands. cysteine-cysteine disulfide bridges largely contribute to the thermodynamic stability of the domain, as frequently observed in small proteins. The structure of BPTI was the first one ever resolved (*Ascenzi et al., 2003*, and is often used to benchmark folding predictions on the basis of simulations *Levitt and Warshel, 1975*) and coevolutionary approaches *Morcos et al., 2011*; *Hopf et al., 2012*; *Kamisetty et al., 2013*; *Cocco et al., 2013*; *Haldane et al., 2018*. We train a RBM with $M = 100$ dReLU on the MSA of PF00014, constituted by $B = 8062$ sequences with $N = 53$ consensus sites.

### Inferred weights and interpretations

*Figure 2B* shows the weights $w_{i\mu}(v)$ attached to five selected hidden units. Each logo identifies the amino-acid motifs in the sequences $\mathbf{v}$ that give rise to large (positive or negative) inputs ($I$) onto the associated hidden unit( see *Equation (2))*.

Weight 1 in *Figure 2B* has large components on sites 45 and 49 that are in contact in the final $\alpha_2$ helix (*Figure 2A and D*). The distribution of the inputs ($I_1$) partitions the MSA into three subfamilies (*Figure 2C*, top panel, dark blue histogram). The two peaks in $I_1 \simeq -2.5$ and $I_1 \simeq 1.5$ identify

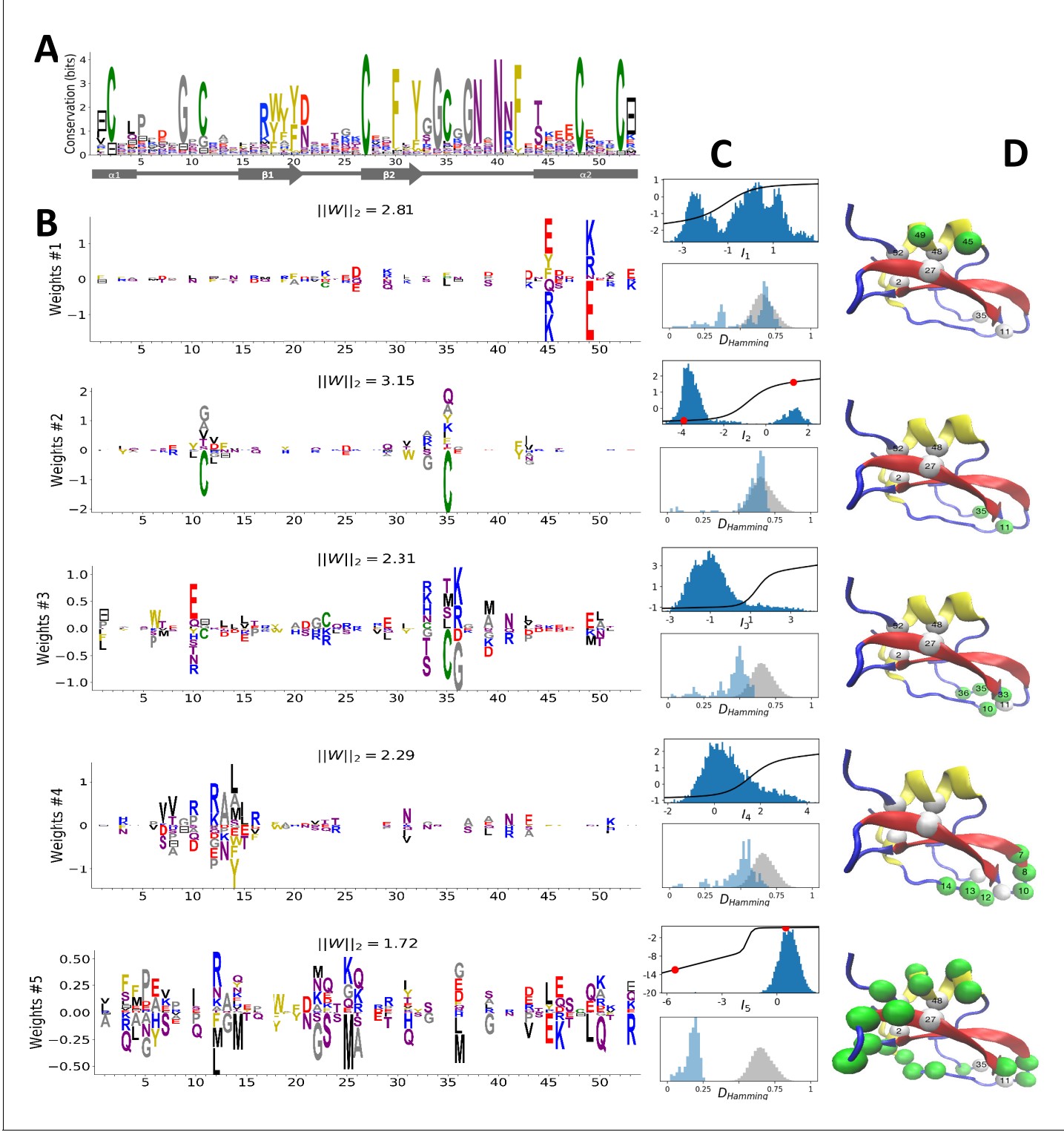

**Figure 2.** Modeling Kunitz Domain with RBM. (**A**) Sequence logo and secondary structure of the Kunitz domain (PF00014), showing two α-helices and two β-strands. Note the presence of the three C-C disulfide bridges between positions 11&35, 2&52 and 27&48. (**B**) Weight logos for five hidden units (see text). Positive and negative weights are shown by letters located, respectively, above and below the zero axis. Values of the norms $\|W_\mu\|_2 = \sqrt{\sum_{i,v} w_{i\mu}(v)^2}$ are given. The color code for the amino acids is the same as that in *Figure 1A*. (**C**) Top: distribution of inputs $I_\mu(\mathbf{v})$ over the sequences $\mathbf{v}$ in the MSA (dark blue), and average activity vs. input function (full line, left scale); red points correspond to the activity levels used for design in *Figure 5*. Bottom: histograms of Hamming distances between sequences in the MSA (grey) and between the 20 sequences (light blue) with

*Figure 2 continued on next page*

*Figure 2 continued*

largest (for unit 2,3,4) or smallest (1,5) $I_\mu$. (D 3D visualization of the weights, shown on PDB structure 2knt *Merigeau et al., 1998* using VMD *Humphrey et al., 1996*. White spheres denote the positions of the three disulfide bridges in the wildtype sequence. Green spheres locate residues $i$ such that $\sum_v |w_{i\mu}(v)| > S$, with $S = 1.5$ for hidden units $\mu = 1, 2, 3$, $S = 1.25$ for $\mu = 4$, and $S = 0.5$ for $\mu = 5$.

DOI: https://doi.org/10.7554/eLife.39397.004

sequences in which the contact is due to an electrostatic interaction with, respectively, $(+, -)$ and $(-, +)$ charged amino acids on sites 45 and 49; the other peak in $I_1 \simeq 0$ identifies sequences realizing the contact differently, for example with an aromatic amino acid on site 45. Weight 1 also shows a weaker electrostatic component on site 53 (*Figure 2B*); the four-site separation interval between sites 45, 49– and 53 fits well with the average helix turn of 3.6 amino acids (*Figure 2D*).

Weight 2 focuses on the contact between residues 11 and 35, realized in most sequences by a C-C disulfide bridge (*Figure 2B* and a negative $I_2$ peak in *Figure 2C* (top). A minority of sequences in the MSA, corresponding to $I_2 > 0$ and mostly coming from nematode organisms (*Appendix 1—figure 19*), do not show the C-C bridge. A subset of these sequences strongly and positively activate hidden unit 3 (*Appendix 1—figure 19* and $I_3 > 0$ peak in *Figure 2C*). Positive components in the weight 3 logo suggest that these proteins stabilize their structure through electrostatic interactions between sites 10 (− charge) and site 33–36 (+ charges both) (see *Figure 2B and D*) that compensates for the absence of a C–C bridge on the neighbouring sites 11–35.

Weight 4 describes a feature that is mostly localized on the loop preceding the $\beta_1$-$\beta_2$ strands (sites 7 to 16) (see *Figure 2B and D*). Structural studies of the trypsin–trypsin inhibitor complex have shown that this loop binds to proteases *Marquart et al., 1983*): site 12 is in contact with the active site of the protease and is therefore key to the inhibitory activity of the Kunitz domain. The two amino acids (R, K) having a large positive contribution to weight 4 in position 12 are basic and bind to negatively charged residues (D, E) on the active site of trypsin-like serine proteases. Although several Kunitz domains with known trypsin inhibitory activity, such as BPTI, TFPI, TPPI-2 and so on, give rise to large and positive inputs ($I_4$), Kunitz domains with no trypsin/chymotrypsin inhibition activity, such as those associated with the *COL7A1* and *COL6A3* genes *Chen et al., 2001*; *Kohfeldt et al., 1996*, correspond to negative or vanishing values of $I_4$. Hence, hidden unit 4 possibly separates the Kunitz domains that have trypsin-like protease inhibitory activity from the others.

This interpretation is also in agreement with mutagenesis experiments carried out on sites 7 to 16 to test the inhibitory effects of Kunitz domains BPT1, HAI-1, and TFP1 against trypsine-like proteases *Bajaj et al., 2001*; *Kirchhofer et al., 2003*; *Shigetomi et al., 2010*; *Grzesiak et al., 2000*; *Chand et al., 2004*). *Kirchhofer et al. (2003)* showed that mutation R12A on the first domain (out of two) of HAI-1 destroyed its inhibitory activity; a similar effect was observed with R12X, where X is a non-basic residue, in the first two domains (out of three) of TFP1 as discussed by *Bajaj et al. (2001)*. *Grzesiak et al. (2000)* showed that for BPTI, the mutations G9F, G9S, G9P reduced its affinity with human serine proteases . Conversely, in *Kohfeldt et al. (1996)* it was shown that the set of mutations P10R, D13A & F14R could convert the COL6A3 domain into a trypsin inhibitor. All of these results are in agreement with the above interpretation and the logo of weight 4. Note that, although several sequences have large $I_4$ (top histogram in *Figure 2C*), many correspond to small or negative values. This may be explained by the facts that (i) many of the Kunitz domains analyzed are present in two or more copies, and as such, not all of them need to bind strongly to trypsin (*Bajaj et al., 2001* and (ii) a Kunitz domain may have other specificities that are encoded by other hidden units. In particular, weight 34 in 'Supporting Information', displays on site 12 large components that are associated with medium- to large-sized hydrophobic residues (L, M, Y), and is possibly related to other serine protease specificity classes such as chymotrypsin (*Appel, 1986*).

Weight 5 codes for a complex extended mode. To interpret this feature, we display in *Figure 2C* (bottom histogram) the distributions of Hamming distances between all pairs of sequences in the MSA (gray histograms) and between the 100 sequences **v** in the MSA with largest inputs $|I_\mu(\mathbf{v})|$ to the corresponding hidden unit (light blue histograms). For hidden unit 5, the distances between those top-input sequences are smaller than those between random sequences in the MSA, suggesting that weight 5 is characteristic of a cluster of closely related sequences. Here, these sequences correspond to the protein Bikunin, which is present in most mammals and some other vertebrates

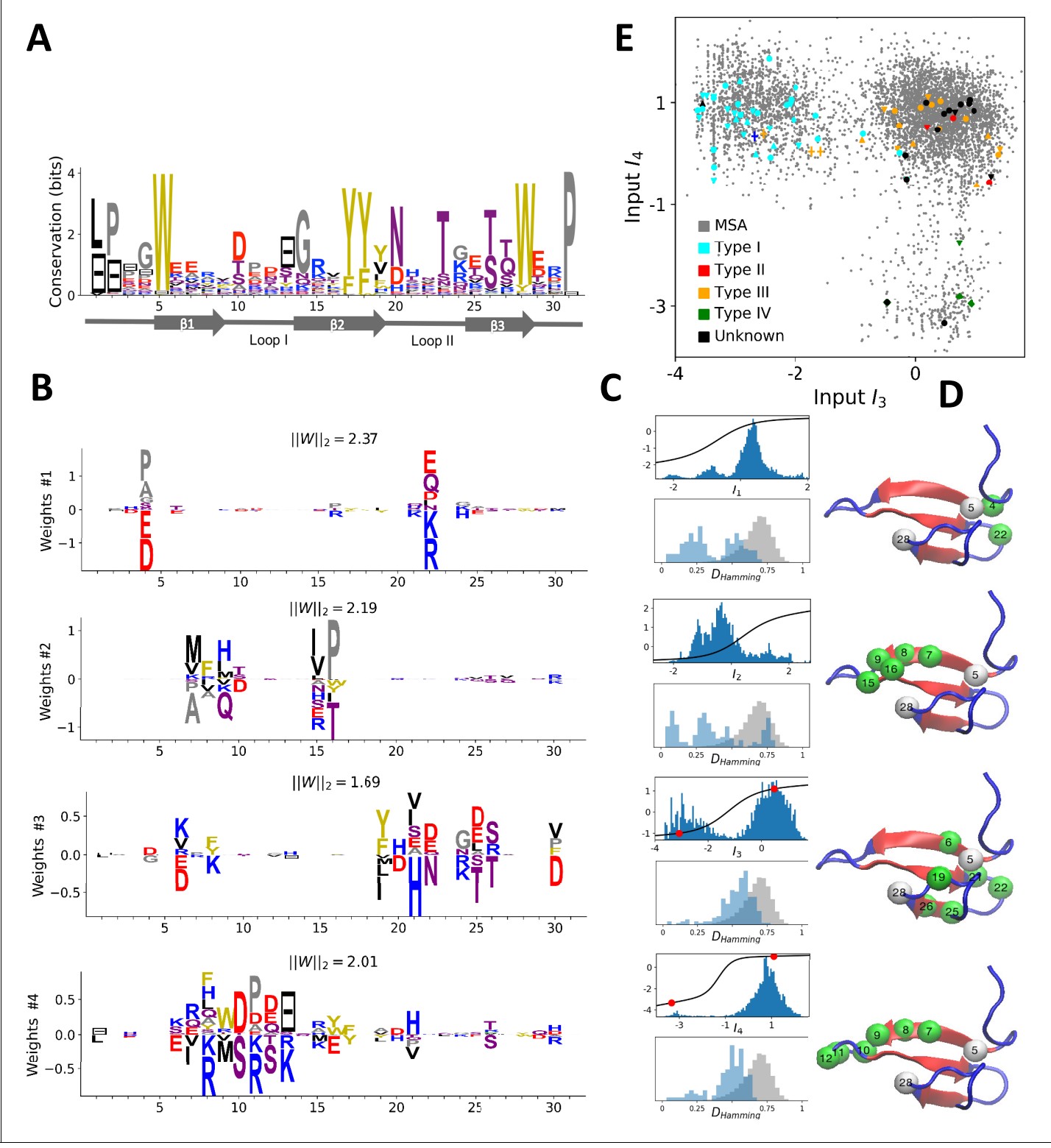

**Figure 3.** Modeling the WW domain with RBM. (**A**) Sequence logo and secondary structure of the WW domain (PF00397), which includes three $\beta$-strands. Note the two conserved W amino acids in positions 5 and 28. (**B**) Weight logos for four representative hidden units. (**C**) Corresponding inputs, average activities and distances between the top-20 feature-activating sequences. (**D**) 3D visualization of the features, shown on the PDB structure 1e0m *Macias et al., 2000*. White spheres locate the two W amino acids. Green spheres locate residues $i$ such that $\sum_v |w_{i\mu}(v)| > 0.7$ for each hidden unit $\mu$. (**E**) Scatter plot of inputs $I_3$ vs. $I_4$. Gray dots represent the sequences in the MSA; they cluster into three main groups. Colored dots show artificial or

*Figure 3 continued on next page*

*Figure 3 continued*

natural sequences whose specificities, given in the legend, were tested experimentally. Upper triangle: natural, from *Russ et al. (2005)*. Lower triangle: artificial, from *Russ et al. (2005)*. Diamond: natural, from *Otte et al. (2003)*. Crosses: YAP1 (0) and variants (1 and 2 mutations from YAP1), from *Espanel and Sudol (1999)*. The three clusters match the standard ligand-type classification.

DOI: https://doi.org/10.7554/eLife.39397.005

---

*Shigetomi et al., 2010*. Conversely, for other hidden units (e.g. 1,2), both histograms are quite similar, showing that the corresponding weight motifs are found in evolutionary distant sequences.

The five weights above were chosen on the basis of several criteria. (i) Weight norm, which is a proxy for the relevance of the hidden unit. Hidden units with larger weight norms contribute more to the likelihood, whereas weights with low norms may arise from noise or overfitting. (ii) Weight sparsity. Hidden units with sparse weights are more easily interpretable in terms of structural or functional constraints. (iii) Shape of input distributions. Hidden units with multimodal input distributions separate the family into subfamilies, and are therefore potentially interesting. (iv) Comparison with available literature. (v) Diversity. The remaining 95 inferred weights are shown in the 'Supporting Information'. We find a variety of both structural features, (for example pairwise contacts as in weights 1 and 2, that are also reminiscent of the localized, low-eigenvalue modes of the Hopfield-Potts model *Cocco et al., 2013*)) and phylogenetic features (activated by evolutionary related sequences as hidden unit 5). The latter, in particular, include stretches of gaps, mostly located at the extremities of the sequence *Cocco et al., 2013*. Several weights have strong components on the same sites as weight 4, showing the complex pattern of amino acids that controls binding affinity.

## WW domain

### Description

WW is a protein–protein interaction domain, found in many eukaryotes and human signaling proteins, that is involved in essential cellular processes such as transcription, RNA processing, protein trafficking, and receptor signaling. WW is a short domain of length 30–40 amino-acids (*Figure 3A*, PFAM PF00397, $B = 7503$ sequences, $N = 31$ consensus sites), which folds into a three-stranded anti-parallel $\beta$-sheet. The domain name stems from the two conserved tryptophans (W) at positions 5–28 (*Figure 3A*), which serve as anchoring sites for the ligands. WW domains bind to a variety of proline (P)-rich peptide ligands, and can be divided into four groups on the basis of their preferential binding affinity (*Sudol and Hunter, 2000*. Group I binds specifically to the PPXY motif, where X is any amino acid; Group II to PPLP motifs; Group III to proline-arginine-containing sequences (PR); Group IV to phosphorylated serine/threonine-proline sites (p(S/T)P). The modulation of binding properties allow hundreds of WW domain to specifically interact with hundreds of putative ligands in mammalian proteomes.

### Inferred weights and interpretation

Four weight logos of the inferred RBM are shown in *Figure 3B*; the remaining 96 weights are given in the 'Supporting Information'. Weight 1 codes for a contact between sites 4 & 22, which is realized either by two amino acids with oppositive charges ($I_1 < 0$) or by one small and one negatively charged amino acid ($I_1 > 0$). Weight 2 shows a $\beta$-sheet–related feature, with large entries defining a set of mostly hydrophobic ($I_2 > 0$) or hydrophilic ($I_2 < 0$) residues localized on the $\beta_1$ and $\beta_2$ strands (*Figure 3B*) and in contact on the 3D fold (see *Figure 3D*). The activation histogram in *Figure 3C*, with a large peak on negative $I_2$, suggests that this part of the WW domain is exposed to the solvent in most, but not all, natural sequences.

Weights 3 and 4 are supported by sites on the $\beta_2$-$\beta_3$ binding pocket and on the $\beta_1$-$\beta_2$ loop of the WW domain. The distributions of activities in *Figure 3C* highlight different groups of sequences in the MSA that strongly correlate with experimental ligand-type identification (see *Figure 3E*). We find that: (i) Type I domains are characterized by $I_3 < 0$ and $I_4 > 0$; (ii) Type II/III domains are characterized by $I_3 > 0$ and $I_4 > 0$; (iii) there is no clear distinction between Type II and Type III domains; and (iv) Type IV domains are characterized by $I_3 > 0$ and $I_4 < 0$. These findings are in good agreement with various studies:

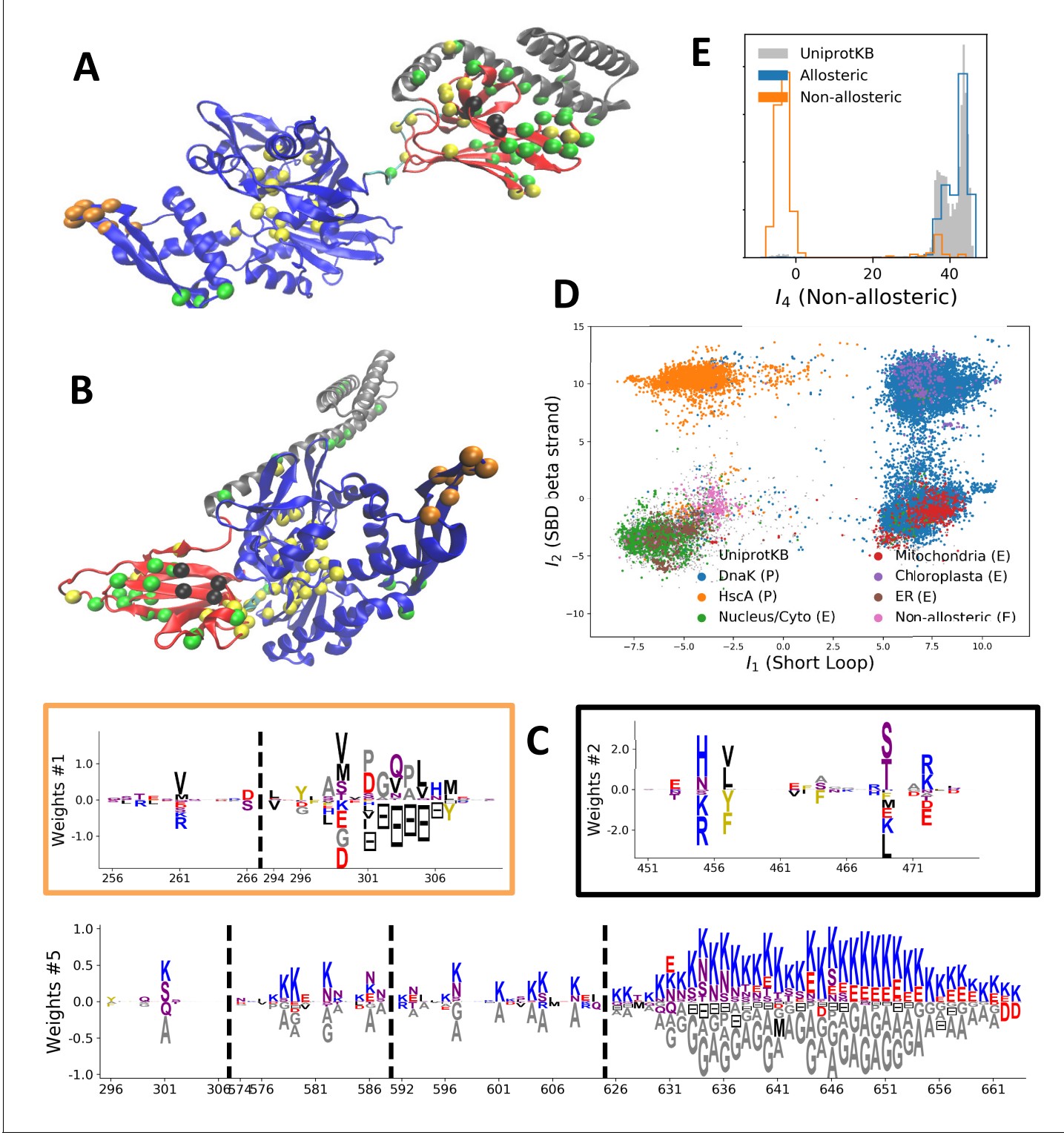

**Figure 4.** Modeling HSP70 with RBM. (**A, B**) 3D structures of the DNaK *E. coli* HSP70 protein in the ADP-bound (A: PDB: 2kho *Bertelsen et al., 2009*) and ATP-bound (B: PDB: 4jne *Qi et al., 2013*) conformations. The colored spheres show the sites carrying the largest entries in the weights in panel (C). (**C**) Weight logos for hidden units $\mu = 1, 2$ and 5 (see *Appendix 1—figure 21* for the other hidden units). Owing to the large protein length, we show only weights for positions $i$ with large weights ($\sum_v |w_{i\mu}(v)| > 0.4 \times \max_i \sum_v |w_{i\mu}(v)|$), with surrounding positions up to ±5 sites away; dashed lines vertical locate the left edges of the intervals. Protein backbone colors: blue = NBD; cyan = linker; red = SBD; gray = LID. Colors: orange = Unit 1 (NBD loop); black = Unit 2 (SBD β strand); green = Unit 3 (SBD/LID); yellow = Unit 4 (Allosteric). (**D**) Scatter plot of inputs $I_1$ vs. $I_2$. Gray dots represent the

*Figure 4 continued on next page*

*Figure 4 continued*
sequences in the MSA, and cluster into four main groups. Colored dots represent the main sequence categories based on gene phylogeny, function and expression. (E) Histogram of input $I_4$, showing separation between allosteric and non-allosteric protein sequences in the MSA.
DOI: https://doi.org/10.7554/eLife.39397.006

i. Mutagenesis experiments have shown the importance of sites 19, 21, 24 and 26 for binding specificity *Espanel and Sudol, 1999*; *Fowler et al., 2010*). For the YAP1 WW domain, as confirmed by various studies (see table 2 in *Fowler et al., 2010*), the mutations H21X and T26X reduce the binding affinity to Type I ligands, whereas Q24R increases it and S12X has no effect. This is in agreement with the negative components of weight 3 (*Figure 3B*): $I_3$ increases upon mutations H21X and T26X, decreases upon Q24R and is unaffected by S12X. Moreover the mutation L19W alone, or in combination with H21[D/G/K/R/S] could switch the specificity from Type I to Type II/III *Espanel and Sudol, 1999*. These results are consistent with *Figure 3E*: YAP1 (blue cross) is of Type I but one or two mutations move it to the right side, closer to the other cluster (orange crosses). *Espanel and Sudol (1999)* also proposed that Type II/III specifity required the presence of an aromatic amino acid (W/F/Y) on site 19, in good agreement with weight 3.

ii. The distinction between Types II and III is unclear in the literature, because WW domains often have high affinity with both ligand types.

iii. Several studies *Russ et al., 2005*; *Kato et al., 2002*; *Jäger et al., 2006*) have demonstrated the importance of the $\beta_1$-$\beta_2$ loop for achieving Type IV specificity, which requires a longer, more flexible loop, as opposed to a short rigid loop for other types. The length of the loop is encoded in weight 4 through the gap symbol on site 13: short and long loops correspond to, respectively, positive and negative $I_4$. The importance of residues R11 and R13 was shown by *Kato et al. (2002)* and *Russ et al. (2005)*, where removing R13 of Type IV hPin1 WW domain reduced its binding affinity to [p(S/T)P] ligands. These observations agree with the logo of weight 4, which authorizes substitutions between K and R on sites 11 and 13.

iv. A specificity-related sector of eight sites was identified in *Russ et al. (2005)*, five of which carry the top entries of weight 3 (green balls in *Figure 3D*). Our approach not only provides another specificity-related feature (weight 4) but also the motifs of amino acids that affectType I and IV specificity, in good agreement with the experimental findings of *Russ et al. (2005)*.

## Hsp70 protein

### Description

70-kDa heat shock proteins (Hsp70) form a highly-conserved family that is represented in essentially all organisms. Hsp70, together with other chaperone proteins, perform a variety of essential functions in the cell: they can assist the folding and assembly of newly synthetized proteins, trigger refolding cycles of misfolded proteins, transport unfolded proteins through organelle membranes, and when necessary, deliver non-functional proteins to the proteasome, endosome or lysosome for recycling *Bukau and Horwich, 1998*; *Young et al., 2004*; *Zuiderweg et al., 2017*. There are 13 HSP70s protein-encoding genes in humans, differing by where (nucleus/cytoplasm, mitochondria or endoplasmic reticulum) and when they are expressed. Some, such as HSPA8 (Hsc70), are constitutively expressed whereas others, such as HSPA1 and HSPA5, are stress-induced (respectively by heat shock and glucose deprivation). Notably, Hsc70 can make up to 3% of the total total mass of proteins within the cell, and thus is one of its most important housekeeping genes. Structurally, Hsp70 are multi-domain proteins of ength of 600–670 sites (631 for the *E. coli* DNaK gene). They consist of:

- A Nucleotide Binding Domain (NBD, 400 sites) that can bind and hydrolyse ATP.
- A Substrate Binding Domain (SBD sites), folded in a beta-sandwich structure, which binds to the target peptide or protein.
- A flexible, hydrophobic interdomain-linker linking the NBD and the SBD.
- A LID domain, constituted by several (up to 5) $\alpha$ helices, which encapsulates the target protein and blocks its release.
- An unstructured C-terminal tail of variable length, which is important for detection and interaction with other co-chaperones, such as Hop proteins (*Scheufler et al., 2000*.

Hsp70 functions by adopting two different conformations (see *Figure 4A and B*). When the NBD is bound to ATP, the NBD and the SBD are held together and the LID is open, such that the protein has low binding affinity for substrate peptides. After the hydrolysis of ATP to ADP, the NBD and the SBD detach from one another, and the LID is closed, yielding high binding affinity and effectively trapping the peptides between the SBD and the LID. By cycling between both conformations, Hsp70 can bind to misfolded proteins, unfold them by stretching (e.g. with two Hsp70 molecules bound at two ends of the protein) and release them for refold cycles. Since Hsp70 alone have low ATPase activity, this cycle requires another type of co-chaperone, J-protein, which simultaneously binds to the target protein and the Hsp70 to stimulate the ATPase activity of Hsp70, as well as a Nucleotide Exchange Factor (NEF) that favors conversion of the ADP back to ATP and hence release of the target protein (see *Figure 1* in *Zuiderweg et al. (2017)*).

We constructed an MSA for HSP70 with $N = 675$ consensus sites and $B = 32,170$ sequences, starting from the seeds of *Malinverni et al. (2015)*, and queried SwissProt and Trembl UniprotKB databases using HMMER3 *Eddy, 2011*. Annotated sequences were grouped on the basis of their phylogenetic origin and functional role. Prokaryotes mainly express two Hsp70 proteins: DnaK ($B = 17,118$ sequences in the alignment), which are the prototype Hsp70, and HscA ($B = 3,897$), which are specialized in chaperoning of iron-sulfur cluster containing proteins. Eukaryotes' Hsp70 were grouped by their location of expression (mitochondria, $B = 851$; chloroplasts, $B = 416$; endoplasmic reticulum, $B = 433$; nucleus or cytoplasm and others, $B = 1,452$). We also singled out Hsp110 sequences, which, despite the high homology with Hsp70, correspond to non-allosteric proteins ($B = 294$). We then trained a dReLU RBM over the full MSA with $M = 200$ hidden units. We show below the weight logos, structures and input distributions for ten selected hidden units (see *Figure 4* and *Appendix 1—figures 21–26*).

## Inferred weights and interpretation

Weight 1 encodes a variability of the length of the loop within the IIB subdomain of the NBD, see stretch of gaps from sites 301 to 306. As shown in *Figure 4D* (projection along x axis), it separates prokaryotic DNaK proteins (for which the loop is 4–5 sites longer) from most eukaryotic Hsp70 proteins and from prokaryotic HscA. An additional hidden unit (Weight 6 in *Appendix 1—figure 21*) further separates eukaryotic Hsp70 from HscA proteins, whose loops are 4–5 sites shorter (distribution of inputs $I_6$ in *Appendix 1—figure 26*). This structural difference between the three families was previously reported and is of high functional importance to the NBD (*Buchberger et al., 1994*; *Brehmer et al., 2001*. Shorter loops increase the nucleotide exchange rates (and thus the release of target protein) in the absence of NEF, and the loop size controls interactions with NEF proteins *Brehmer et al., 2001*; *Briknarová et al., 2001*; *Sondermann et al., 2001*). Hsp70 proteins that have long and intermediate loop sizes interact specifically with GrpE and Bag-1 NEF proteins, respectively, whereas short, HscA-like loops do not interact with any of them. This cochaperone specificity allows for functional diversification within the cell; for instance, eukaryotic Hsp70 proteins that are expressed within mitochondria and chloroplasts, such as the human gene HSPA9 and the *Chlamydomonas reinhardtii* HSP70B, share the long loop with prokaryotic DNaK proteins, and therefore do not interact with Bag proteins. Within the DNaK subfamily, two main variants of the loop can be isolated as well (Weight 7 in *Appendix 1—figure 22*), hinting at more NEF-protein specificities.

Weight 2 encodes a small collective mode localized on $\beta_4 - \beta_5$ strands, at the edge of the β sandwich within the SBD. The weights are quite large ($w \sim 2$), and the input distribution is bimodal, notably separating HscA and chloroplast Hsp70 ($I_2 > 0$) from mitochondrial Hsp70 and the other eukaryotic Hsp70 ($I_2 < 0$). We note also a similarity in structural location and amino-acid content with weight 3 of the WW–domain, which controls binding specificity (*Figure 3B*). Although we have found no trace of this motif in the literature, this evidence suggests that it could be important for substrate binding specificity. Endoplasmic-reticulum-specific Hsp70 proteins can also be separated from the other eukaryotic proteins by looking at appropriate hidden units (see Weight 8 in *Appendix 1—figure 22* and the distribution of input $I_8$ in *Appendix 1—figure 26*).

RBM can also extract collective modes of coevolution spanning multiple domains, as shown by Weight 3 (*Appendix 1—figure 21*). The residues supporting Weight 3 (green spheres in *Figure 4A and B*) are physically contiguous in the ADP conformation, but not in the ATP conformation. Hence, Weight 3 captures inter-domain coevolution between the SBD and the LID domains.

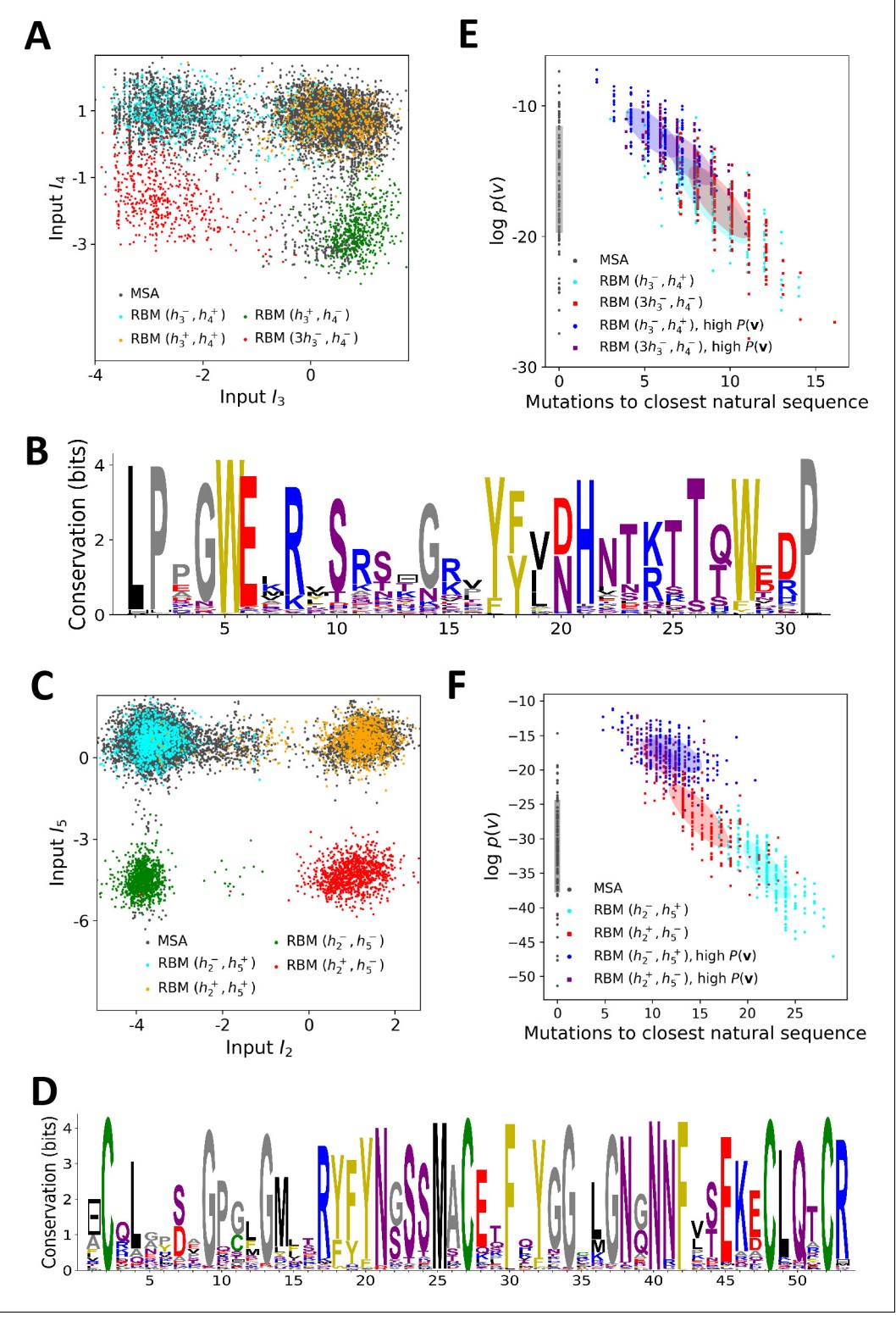

**Figure 5.** Sequence design with RBM. (**A**) Conditional sampling of WW domain-modeling RBM. Sequences are drawn according to *Equation (3)*, with activities $(h_3, h_4)$ fixed to $(h_4^-, h_4^+)$, $(h_3^+, h_4^-)$, $(h_3^+, h_4^+)$ and $(3h_3^-, h_4^-)$, see red points indicating the values of $h_3^\pm, h_4^\pm$ in *Figure 3C*. Natural sequences in the MSA are shown with gray dots, and generated sequences with colored dots. Four clusters of sequences are obtained; the first three are putatively associated to, respectively, ligand-specific groups I, II/III and IV. The sequences in the bottom left cluster,
*Figure 5 continued on next page*

*Figure 5 continued*

obtained through very strong conditioning, do not resemble any of the natural sequences in the MSA; their binding specificity is unknown. (B) Sequence logo of the red sequences in panel (A), with 'long $\beta_1$-$\beta_2$ loop' and 'type I' features. (C) Conditional sampling of Kunitz domain-modeling RBM, with activities $(h_2, h_5)$ fixed to $(h_2^\pm, h_5^\pm)$, see red dots indicating $h_2^\pm, h_5^\pm$ in *Figure 2C*. Red sequences combine the absence of the 11–35 disulfide bridge and a strong activation of the Bikunin-AMBP feature, although these two phenotypes are never found together in natural sequences. (D) Sequence logo of the red sequences in panel (C), with 'no disulfide bridge' and 'bikunin' features. (E) Scatter plot of the number of mutations to the closest natural sequence vs log-probability, for natural (gray) and artificial (colored) WW domain sequences. The color code is the same as that in panel (A); dark dots were generated with the high-probability trick, based on duplicated RBM (see 'Materials and methods'). Note the existence of many high-probability artificial sequences far away from the natural ones. (F) The same scatter plot as in panel (E) for natural and artificial Kunitz-domain sequences.

DOI: https://doi.org/10.7554/eLife.39397.007

---

Weight 4 (sequence logo in *Appendix 1—figure 21*) also codes for a wide, inter–domain collective mode, which is localized at the interface between the SBD and the NBD domains. When the Hsp70 protein is in the ATP conformation, the sites carrying weight 4 are physically contiguous, whereas in the ADP state they are far apart (see yellow spheres in *Figure 4A and B*). Moreover, its input distribution (shown in *Figure 4E*), separates the non-allosteric Hsp110 subfamily ($I_4 \sim 0$) from the other subfamilies ($I_4 \sim 40$), suggesting that this motif is important for allostery. Several mutational studies have highlighted 21 important sites for allostery within *E. coli* DNaK *Smock et al., 2010*; seven of these positions carry the top entries of Weight 3, four appear in another Hsp110-specific hidden unit (Weight 9 in *Appendix 1—figure 22*), and several others are highly conserved and do not coevolve at all.

Last, Weight 5 (*Figure 4C*) codes for a collective mode that is located mainly on the unstructured C-terminal tail, with a few sites on the LID domain. Its amino-acid content is strikingly similar across all sites: positive weights for hydrophilic residues (in particular, lysine) and negative weights for tiny, hydrophobic residues. Hydrophobic-rich or hydrophilic-rich sequences are found in the MSA (see *Appendix 1—figure 28*). This motif is consistent with the role of the tail in cochaperone interaction: hydrophobic residues are important for the formation of Hsp70–Hsp110 complexes via the Hop protein *Scheufler et al., 2000*. High-charge content is also frequently encountered, and is the basis of a recognition mechanism, in intrinsically disordered protein regions *Oldfield and Dunker, 2014*. This could suggest the existence of different protein partners.

Some of the results presented here were previously obtained with other coevolutionary methods. In *Malinverni et al. (2015)*, the authors showed that Direct Coupling Analysis could detect conformation-specific contacts; these are similar to hidden units 3 and 4 presented here which are located on contiguous sites in the ADP-bound and ATP-bound conformations, respectively. In *Smock et al. (2010)*, an inter-domain sector of sites discriminating between allosteric and non-allosteric sequences was found. This sector shares many sites with our weight 4, and is also localized at the SBD/NBD edge. However, only a sector could be retrieved with sector analysis, whereas many other meaningful collective modes could be extracted using RBM.

## Sequence design

The biological interpretation of the features inferred by the RBM guides us to sample new sequences $\mathbf{v}$ with putative functionalities. In practice, we sample from the conditional distribution $P(\mathbf{v}|\mathbf{h})$, *Equation (4)*, where a few hidden-unit activities in the representation $\mathbf{h}$ are fixed to desired values, whereas the others are sampled from *Equation (3)*. For WW domains, we condition on the activities of hidden units 3 and 4, which are related to binding specificity. Fixing $h_3$ and $h_4$ to levels corresponding to the peaks in the histograms of inputs in *Figure 3C* allows us to generate sequences belonging specifically to each one of the three ligand-specificity clusters (see *Figure 5A*).

In addition, sequences with combinations of activities that are not encountered in the natural MSA can be engineered. As an illustration, we used conditional sampling to generate hybrid WW-domain sequences with strongly negative values of $h_3$ and $h_4$, corresponding to a Type I-like $\beta_2$-$\beta_3$ binding pocket and a long, Type IV-like $\beta_1$-$\beta_2$ loop (see *Figure 5A and B*).

For Kunitz domains, the property 'no 11–35 disulfide bond' holds only for some sequences of nematode organisms, whereas the Bikunin-AMBP gene is present only in vertebrates; the two corresponding motifs are thus never observed simultaneously in natural sequences. Sampling our RBM conditioned to appropriate levels of $h_2$ and $h_5$ allows us to generate sequences with both features activated (see *Figure 5C and D*).

The sequences designed by RBM are far away from all natural sequences in the MSA, but have comparable probabilities (see *Figure 5E* (WW) and *Figure 5F* (Kunitz)). Their probabilities estimated with pairwise direct-coupling models (trained on the same data), whose ability to identify functional and artificial sequences has already been tested (*Balakrishnan et al., 2011*; *Cocco et al., 2018* andare also large (see *Appendix 1—figure 7*).

Our RBM framework can also be modified to design sequences with very high probabilities, even larger than in the MSA, by appropriate duplication of the hidden units (see 'Materials and methods'). This trick can be combined with conditional sampling (see *Figure 5E and F*).

## Contact predictions

As illustrated above, the co-occurrence of large weight components in highly sparse features often corresponds to nearby sites on the 3D fold. To extract structural information in a systematic way, we use our RBM to derive effective pairwise interactions between sites, which can then serve as estimators for contacts as approaches that are based on direct-coupling *Cocco et al., 2018*. The derivation is sketched in *Figure 6A*. We consider a sequence $\mathbf{v}^{a,b}$ with residues $a$ and $b$ on, respectively, sites $i$ and $j$. Single mutations $a \to a'$ or $b \to b'$ on, respectively, site $i$ or $j$ are accompanied by changes in the log probability of the sequence (indicated by the full arrows in *Figure 6A*). Comparison of the change resulting from the double mutation with the sum of the changes resulting from the two single mutations provides our RBM-based estimate of the epistatic interaction (see *Equations (15,16)* in 'Materials and methods'). These interactions are well correlated with the outcomes of the Direct-Coupling Analysis (see *Appendix 1—figure 9*).

*Figure 6* shows that the quality of the prediction of the contact maps of the Kunitz (*Figure 6B*) and the WW (*Figure 6C*) domains with RBM is comparable to state-of-the-art methods based on direct couplings (*Morcos et al., 2011*); predictions for long-range contacts are reported in *Appendix 1—figure 10*. The quality of contact prediction with RBM:

- Does not seem to depend much on the choice of the hidden-unit potential see the Gaussian and dReLU PPV performances in *Figure 6B,C and D*, although the latter have better performance in terms of sequence scoring than the former (see *Appendix 1—figures 1*, *2* and *5*).
- Strongly increases with the number of hidden units (see *Appendix 1—figures 11,12*). This dependence is not surprising, as the number $M$ of hidden units acts in practice as a regularizor over the effective coupling matrix between residues. In the case of Gaussian RBM, the value of $M$ fixes the maximal rank of the matrix $J_{ij}(v_i, v_j)$ (see 'Materials and methods'). The value $M = 100$ of the number of hidden units is small compared to the maximal ranks $R = 20 \times N$ of the couplings matrices of the Kunitz ($R = 1060$) and WW ($R = 620$) domains, and explains why Direct-Coupling Analysis gives slightly better performance than RBM in the contact predictions of *Figure 6B and C*.
- Worsens with stronger weight-sparsifying regularizations (see *Appendix 1—figure 12*) as expected.

We further tested RBM distant contact predictions in a fully blind setting on the 17 protein families (the Kunitz domain plus 16 other domains) that were used for to benchmark plmDCA (*Ekeberg et al., 2014*), a state-of-the-art procedure for inferring pairwise couplings in Direct-Coupling Analysis. The number of idden units was fixed to $M = 0.3\,R$, that is proportionally to the domain lengths, and the regularization strength was fixed to $\lambda_1^2 = 0.1$. Contact predictions averaged over all families are reported in *Figure 6D* for different choices of the hidden-unit potentials (Gaussian and dReLU). We find that performances are comparable to those of plmDCA, but the computational cost of training RBM is substantially higher.

## Benchmarking on lattice proteins

Lattice protein (LP) models were introduced in the $90's$ to study protein folding and design (*Mirny and Shakhnovich, 2001*. In one of those models *Shakhnovich and Gutin, 1990*, a 'protein'

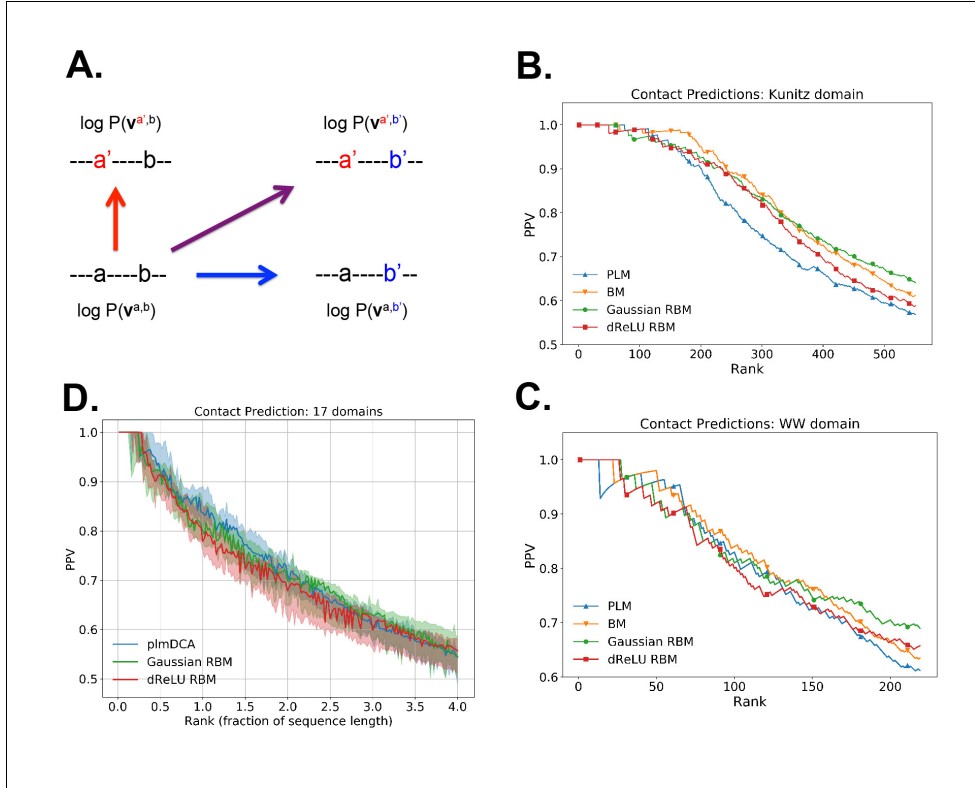

**Figure 6.** Contact predictions using RBM. (**A**) Sketch of the derivation with RBM of effective epistatic interactions between residues. The change in log probability resulting from a double mutation (purple arrow) is compared to the sum of the changes accompanying the single mutations (blue and red arrows) (see text and 'Materials and methods', *Equations (15,16)*). (**B**) Positive Predictive Value (PPV) vs. pairs $(i, j)$ of residues, ranked according to their scores for the Kunitz domain. RBM predictions with quadratic (Gaussian RBM) and dReLU potentials are compared to direct coupling-based methods, namely the Pseudo-Likelihood Method (plmDCA) *Ekeberg et al., 2014*) and Boltzmann Machine (BM) learning *Sutto et al., 2015*). (**C**) Same as panel (**B**) for the WW domain. (**D**) Distant contact predictions for the 17 protein domains used to benchmark plmDCA in *Ekeberg et al. (2014)* obtained using fixed regularization $\lambda_1^2 = 0.1$ and $M = 0.3 \times N \times 20$. PPV for contacts between residues separated by at least five sites along the protein backbone vs. ranks of the corresponding couplings, expressed as fractions of the protein length $N$; solid lines indicate the median PPV and colored areas the corresponding 1/3 to 2/3 quantiles.

DOI: https://doi.org/10.7554/eLife.39397.008

of $N = 27$ amino acids may fold into $\sim 10^5$ distinct structures on a $3 \times 3 \times 3$ cubic lattice, with probabilities depending on its sequence (see 'Materials and methods' and *Figure 7A and B*). LP sequence data were used to benchmark the Direct-Coupling Analysis in *Jacquin et al. (2016)*, and we follow the same approach here to assess the performances of RBM in a case where the ground truth is known. We first generate a MSA containing sequences that have large probabilities ($p_{nat} > 0.99$) of folding into one structure shown in *Figure 7A* (*Jacquin et al., 2016*). A RBM with $M = 100$ dReLU hidden units is then learned, (see Appendix 1 for details about regularization and cross-validation).

Various structural LP features are encoded by the weights as in real proteins, including complex negative-design related modes (see *Figure 7C and D* and the remaining weights in 'Supporting Information'). The performances in terms of contact predictions are comparable to state-of-the art methods on LP (see *Appendix 1—figure 11*).

The capability of RBM to design new sequences that have desired features and high values of fitness, exactly computable in LP as the probability of folding into the native structure in *Figure 7A*, can be quantitatively assessed. Conditional sampling allows us to design sequences with specific hidden-unit activity levels, or combinations of features that are not found in the MSA (*Figure 7E*). These designed sequences are diverse and have large fitnesses, comparable to those of the MSA

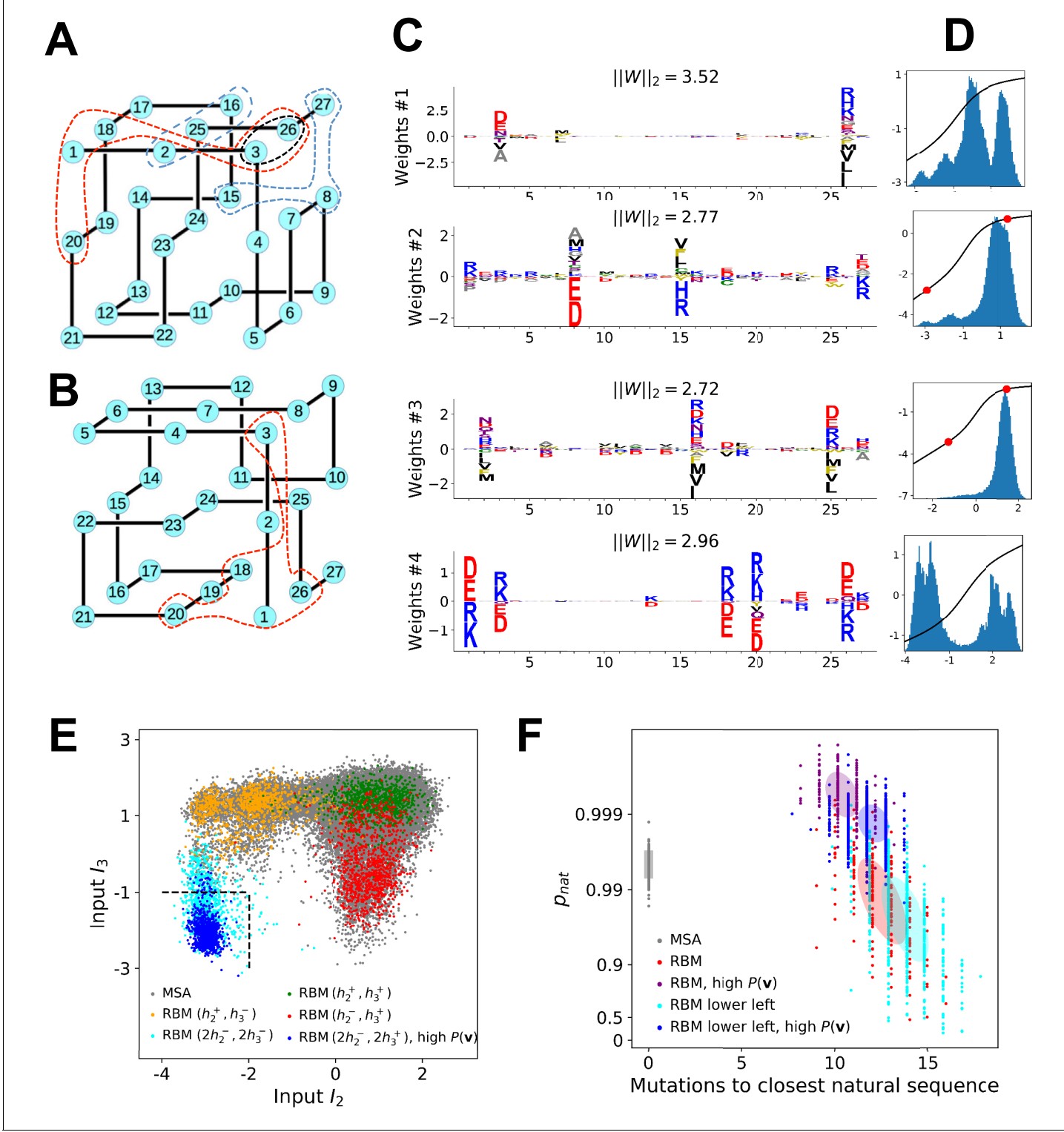

**Figure 7.** Benchmarking RBM with lattice proteins. (A) $S_A$, one of the 103,406 distinct structures that a 27-mer can adopt on the cubic lattice *Shakhnovich and Gutin, 1990*. Circled sites are related to the features shown in *Figure 6C*. (B)$S_G$, another fold with a contact map (set of neighbouring sites) close to $S_A$ *Jacquin et al., 2016*. (C) Four weight logos for a RBM inferred from sequences folding into $S_A$, see 'Supporting Information' for the remaining 96 weights. Weight 1 corresponds to the contact between sites 3 and 26, see black dashed contour in panel (A). The contact can be realized by amino acids of opposite (-+) charges ($I_1 > 0$) or by hydrophobic residues ($I_1 < 0$). Weights 2 and 3 are related to, respectively, the triplets of amino acids 8-15-27 and 2-16-25, each realizing two overlapping contacts on $S_A$ (blue dashed contours). Weight 4 codes for electrostatic

*Figure 7 continued on next page*

*Figure 7 continued*

contacts between sites 3 & 26, 1 & 18 and 1 & 20, and imposes the conditon that the charges of amino acids 1 and 26 have the same sign. The latter constraint is not due to the native fold (1 and 26 are 'far away' on $S_A$) but because folding must be impeded in the 'competing' structure, $S_G$ (*Figure 7B* and 'Materials and methods') in which sites 1 and 26 are neighbours *Jacquin et al., 2016*). (D) Distributions of inputs ($I$) and average activities (full line, left scale). All features are activated across the entire sequence space (not shown). (E) Conditional sampling with activities ($h_2, h_3$) fixed to ($h_2^\pm, h_3^\pm$), see red dots in panel (D). Designed sequences occupy specific clusters in the sequence space, corresponding to different realizations of the overlapping contacts encoded by weights 2 and 3 (*Figure 6C*). Conditioning to ($h_2^-, h_3^+$) makes it possible to generate sequences combining features that are not found together in the MSA (see bottom left corner), even with very high probabilities (see 'Materials and methods'). (F) Scatter plot of the number of mutations to the closest natural sequence vs. the probability $p_{nat}$ of folding into structure $S_A$ (see *Jacquin et al., 2016* for a precise definition) for natural (gray) and artificial (colored) sequences. Note the large diversity and the existence of sequences with higher $p_{nat}$ than those in the training sample.

DOI: https://doi.org/10.7554/eLife.39397.009

sequences and even higher when generated by duplicated RBM (*Figure 7F*), and well correlated with the RBM probabilities $P(\mathbf{v})$ (*Appendix 1—figure 6*).

## Cross-validation of the model and interpretability of the representations

Each RBM was trained on a randomly chosen subset of 80% of the sequences in the MSA, while the remaining 20% (the test set) were used for validation of its predictive power. In practice, we compute the average log-probability of the test set to assess the performances of the RBM for various values of the number $M$ of hidden units, for the regularization strength $\lambda_1^2$ and for different hidden-unit potentials. Results for the WW and Kunitz domains and for Lattice Proteins are reported in *Figure 8* and in Appendix 2 (Model Selection). The dReLU potential, which includes quadratic and Bernoulli (another popular choice for RBM) potentials as special cases, is consistently better than the quadratic and Bernoulli potentials individually. As expected, increasing $M$ allows RBM to capture more features in the data distribution and, therefore, improves performances up to a point, after which overfitting starts to occur.

The impact of the regularization strength $\lambda_1^2$ favoring weight sparsity (see definition in 'Materials and methods' *Equation (8)*) is two-fold (see *Figure 8A* for the WW domain). In the absence of regularization ($\lambda_1^2 = 0$) weights have components on all sites and residues, and the RBM overfit the data, as illustrated by the large difference between the log-probabilities of the training and test sets. Overfitting notably results in generated sequences that are close to the natural ones and not very diverse, as seen from the entropy of the sequence distribution (*Appendix 1—figure 8*). Imposing mild regularization allows the RBM to avoid overfitting and maximizes the log-probability of the test set ($\lambda_1^2 = 0.03$ in *Figure 8A*), but most sites and residues carry non-zero weights. Interestingly, imposing stronger regularizations has low impact on the generalization abilities of RBM (resulting in a small decrease in the test set log-probability), while making weights much sparser ($\lambda_1^2 = 0.25$ in *Figure 3*). For regularizations that are too large, too few non-zero weights remain available and the RBM is not powerful enough to model the data adequately (causing a drop in log-probability of the test set).

Favoring sparser weights in exchange for a small loss in log-probability has a deep impact on the nature of the representation of the sequence space by the RBM (see *Figure 8B*). Good representations are expected to capture the invariant properties of sequences across evolutionarily divergent organisms, rather than idiosyncratic features that are attached to a limited set of sequences (mixture model in *Figure 8C*). For sparse-enough weights, the RBM is driven into the compositional representation regime (see *Tubiana and Monasson, 2017*) of *Figure 8E*, in which each hidden unit encodes a limited portion of a sequence and the representation of a sequence is defined by the set of hidden units with strong inputs. Hence, the same hidden unit (e.g. weights 1 and 2 coding for the realizations of contacts in the Kunitz domain in *Figure 2B*) can be recruited in many parts of the sequence space corresponding to very diverse organisms (see bottom histograms attached to weights 1 and 2 in *Figure 2C*, which shows that the sequences corresponding to strong inputs are scattered all over the sequence space). In addition, silencing or activating one hidden unit affects only a limited number of residues (contrary to the entangled regime of *Figure 8D*), and a large diversity of sequences can be generated through combinatorial choices of the activity states of the hidden units, an approach that guarantees efficient sequence design.

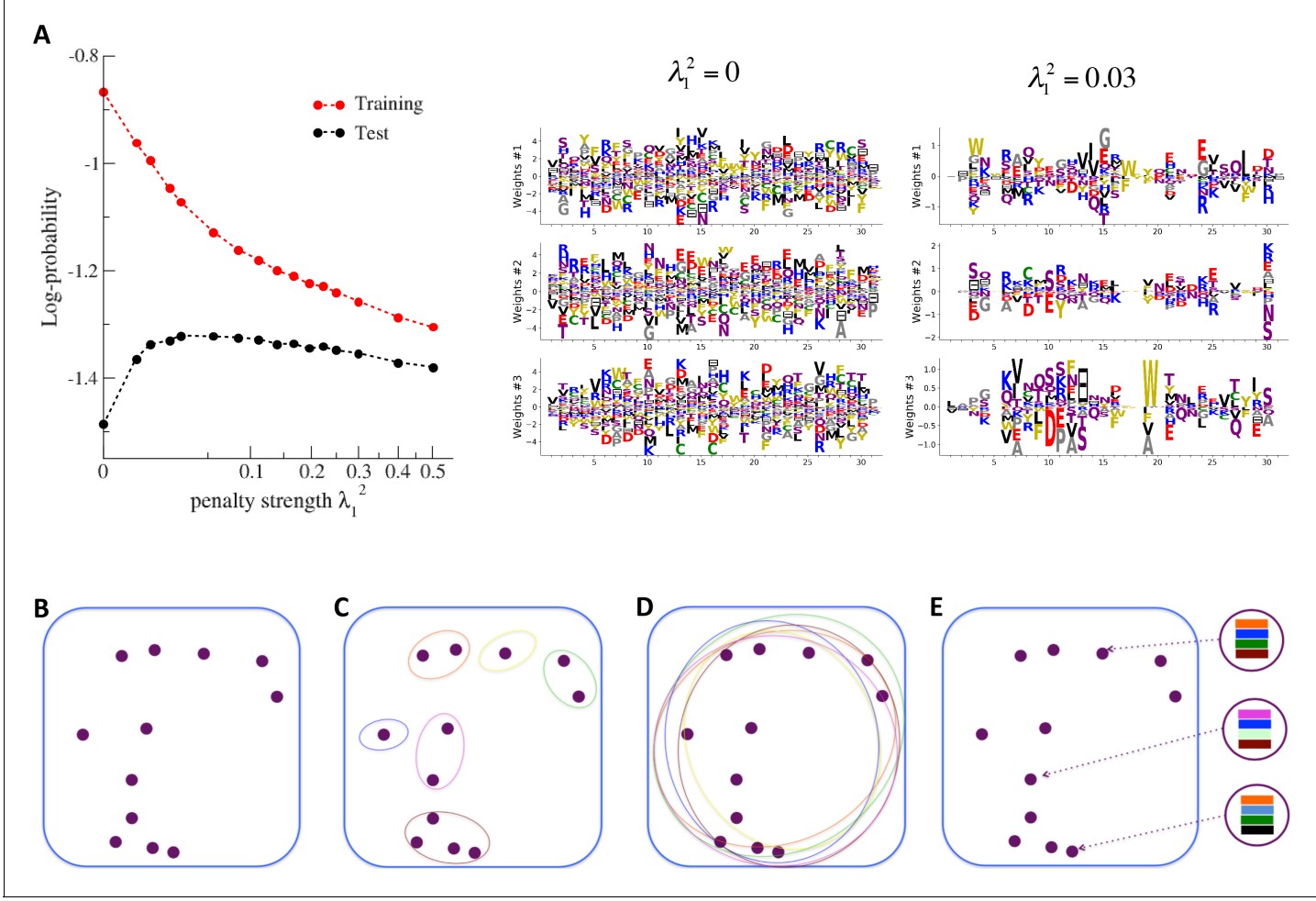

**Figure 8.** Nature of the representations built by RBM and interpretability of weights. (A) The effect of sparsifying regularization. Left: log-probability (see , *Equation (5)*) as a function of the regularization strength $\lambda_1^2$ (square root scale) for RBM with $M = 100$ hidden units trained on WW domain sequence data. Right: the weights attached to three representative hidden units are shown for $\lambda_1^2 = 0$ (no regularization) and 0.03 (optimal log-likelihood for the test set, see left panel); weights shown in *Figure 3* were obtained at higher regularization $\lambda_1^2 = 0.25$. For larger regularization, too many weights vanish, and the log-likelihood diminishes. (B) Sequences (purple dots) in the MSA attached to a protein family define a highly sparse subset of the sequence space (symbolized by the blue square), from which a RBM model is inferred. The RBM then defines a distribution over the entire sequence space, with high scores for natural sequences and over many more other sequences putatively belonging to the protein family. The representations of the sequence space by RBM can be of different types, three examples of which are sketched in the following panels. (C) *Mixture model:* each hidden unit focuses on a specific region in sequence space (color ellipses, different colors correspond to different units), and the attached weights form a template for this region. The representation of a sequence thus involves one (or a few) strongly activated hidden units, while all remaining units are inactive. (D) *Entangled model:* all hidden units are moderately active across the sequence space. The pattern of activities vary from one sequence to another in a complex manner. (E) *Compositional model:* a moderate number of hidden units are activated for each protein sequence, each recognizing one of the motifs (shown by colors) in the sequence and controling one of the protein's biological properties. Composing the different motifs in various ways (right circled compositions) generates a large diversity of sequences.
DOI: https://doi.org/10.7554/eLife.39397.010

In addition, inferring sparse weights makes their comparison across many different families easier. In *Figure 9 and 10*, we show some representative weights that were obtained after training RBMs with the MSAs of the 16 families considered by *Ekeberg et al. (2014)* (the 17th family, the Kunitz domain, is shown in *Figure 2*), which were chosen to illustrate the broad classes of encountered motifs; see 'Supporting information' for the other top weights of the 16 families. We find that weights may code for a variety of structural properties:

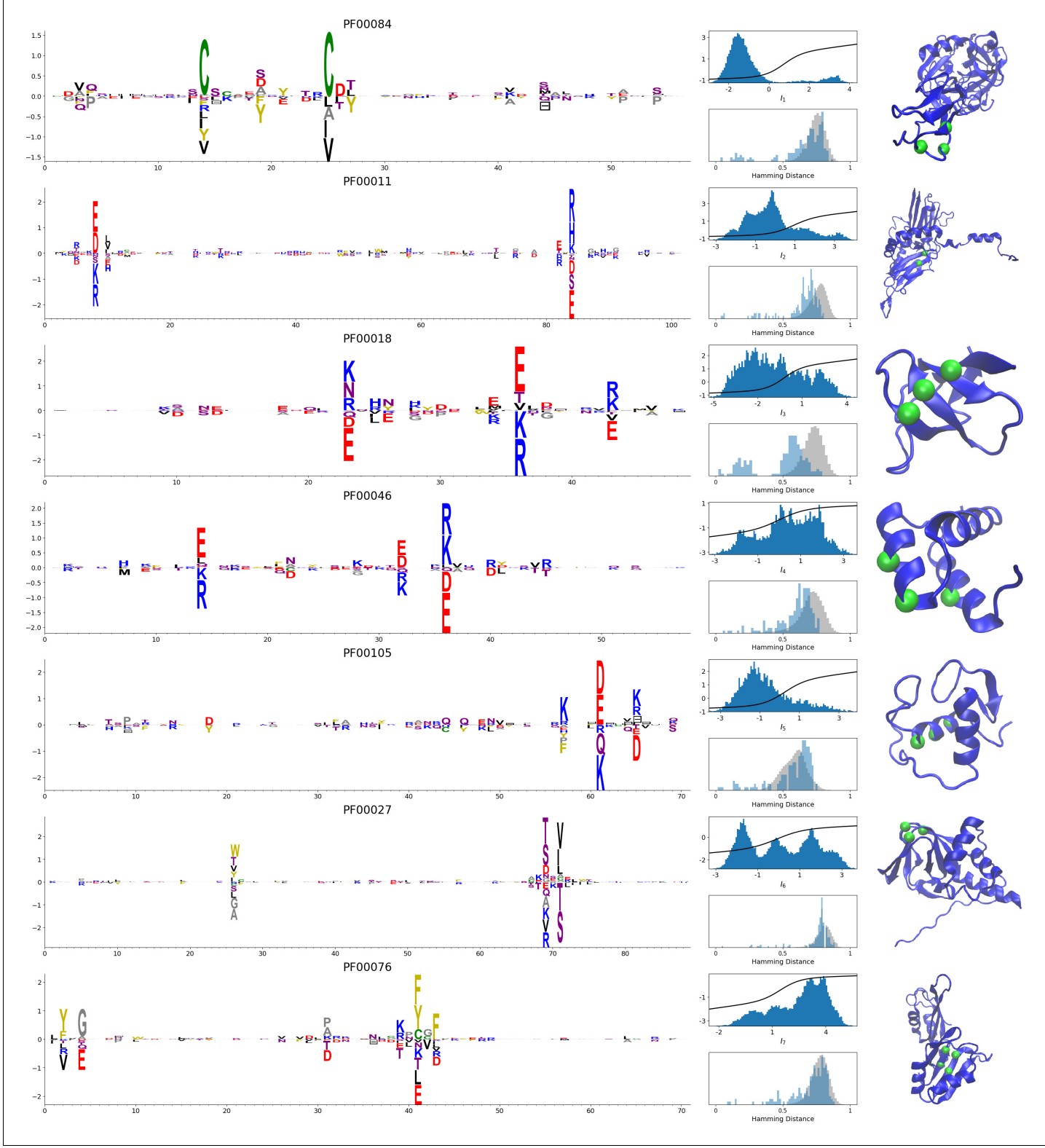

**Figure 9.** Representative weights of the protein families selected in *Ekeberg et al. (2014)*. RBM parameters: $\lambda_1^2 = 0.25$, $M = 0.05 \times N \times 20$. The format is the same as that used in *Figures 2B*, *3B* and *4B*. Weights are ordered by similarity, from top to bottom: Sushi domain (PF00084), Heat shock protein Hsp20 (PF00011), SH3 Domain (PF00018), Homeodomain protein (PF00046), Zinc finger–C4 type (PF00105), Cyclic nucleotide-binding domain (PF00027), and RNA recognition motif (PF00076). Green spheres show the sites that carry the largest weights on the 3D folds (in order, PDB: 1elv, 2bol, 2hda, 2vi6, 1gdc, 3fhi, 1g2e). The ten weights with largest norms in each family are shown in *Supplementary files 5–6*.
DOI: https://doi.org/10.7554/eLife.39397.011

- Pairwise contacts on the corresponding structures, realized by various types of residue-residue physico-chemical interactions (see *Figure 9A and B*). These motifs are similar to weights 2 of the Kunitz domain (*Figure 2B*) and weight 1 of the WW domain (*Figure 3B*).
- Structural triplets, carrying residues in proximity either on the tertiary structure or on the secondary structure (see *Figure 9C,D,E and F*). Many such triplets arise from electrostatic interactions and carry amino acids with alternating charges (*Figure 9C,D and E*); they are often found in α-helices and reflect their ∼4-site periodicity (*Figure 9E* and last two sites in *Figure 9D*), in agreement with weight 1 of the Kunitz domain (*Figure 2B*). Triplets may also involve residues with non-electrostatic interactions (*Figure 9F*).
- Other structural motifs involving four or more residues, for example between β-strands (see *Figure 9G*). Such motifs were also found in the WW domain (see weight 2 in *Figure 3B*).

In addition, weights may also reflect non-structural properties, such as:

- Stretches of gaps at the extremities of the sequences, indicating the presence of subfamilies containing shorter proteins (see *Figure 10A and B*).
- Stretches of gaps in regions corresponding to internal loops of the proteins (see *Figure 10C and D*). These motifs control the length of these loops, similarly to weight 1 of HSP70 (see *Figure 4C*).
- Contiguous residue motifs on loops (*Figure 10E and F*) and β–strands (*Figure 10G*). These motifs could be involved in binding specificity, as found in the Kunitz and WW domains (weights 4 in *Figure 2B and 3B*).
- Phylogenetic properties shared by a subset of evolutionary close sequences (see bottom histograms *Figure 10H and I*), contrary to the motifs listed above. These motifs are generally less sparse and scattered over the protein sequence, as weight 5 of the Kunitz domain in *Figure 2B*.

For all those motifs, the top histograms of the inputs on the corresponding hidden units indicate how the protein families cluster into distinct subfamilies with respect to the features.

## Discussion

In summary, we have shown that RBM are a promising, versatile, and unifying method for modeling and generating protein sequences. RBM, when trained on protein sequence data, reveal a wealth of structural, functional and evolutionary features. To our knowledge, no other method used to date has been able to extract such detailed information in a unique framework. In addition, RBM can be used to design new sequences: hidden units can be seen as representation-controling knobs, that are tunable at will to sample specific portions of the sequence space corresponding to desired functionalities. A major and appealing advantage of RBM is that the two-layer architecture of the model embodies the very concept of genotype-phenotype mapping (*Figure 1C*). Codes for learning and visualizing RBM are attached to this publication (see 'Materials and methods').

From a machine-learning point of view, the values of RBM that define parameters (such as class of potentials and number $M$ of hidden units, or regularization penalties) were selected on the basis of the log-probability of a test set of natural sequences not used for training and on the interpretability of the model. The dReLU potentials that we introduced in this work (*Equation (6)*) consistently outperform other potentials for generative purposes. As expected, increasing $M$ improves likelihood up to some level, after which overfitting starts to occur. Adding sparsifying regularization not only prevents overfitting but also facilitates the biological interpretation of weights (*Figure 8A*). It is thus an effective way to enhance the correspondence between representation and phenotypic spaces (*Figure 1C*). It also allows us to drive the RBM operation point at which most features can be activated across many regions of the sequence space (*Figure 8E*); examples are provided by hidden units 1 and 2 for the Kunitz domain in *Figure 2B and C* and hidden unit 3 for the WW domain in *Figure 3B and C*. Combining these features allows us to generate a variety of new sequences with high probabilities, such as those shown in *Figure 5*. Note that some inferred features, such as hidden unit 5 in *Figure 2C and D* and, to a lesser extent, hidden unit 2 in *Figure 3B and C*, are, by contrast, activated by evolutionary close sequences. Our inferred RBMs thus share some partial similarity with the mixture models of *Figure 8C*. Interestingly, the identification of specific sequence motifs with structural, functional or evolutionary meaning does not seem to be restricted to a few

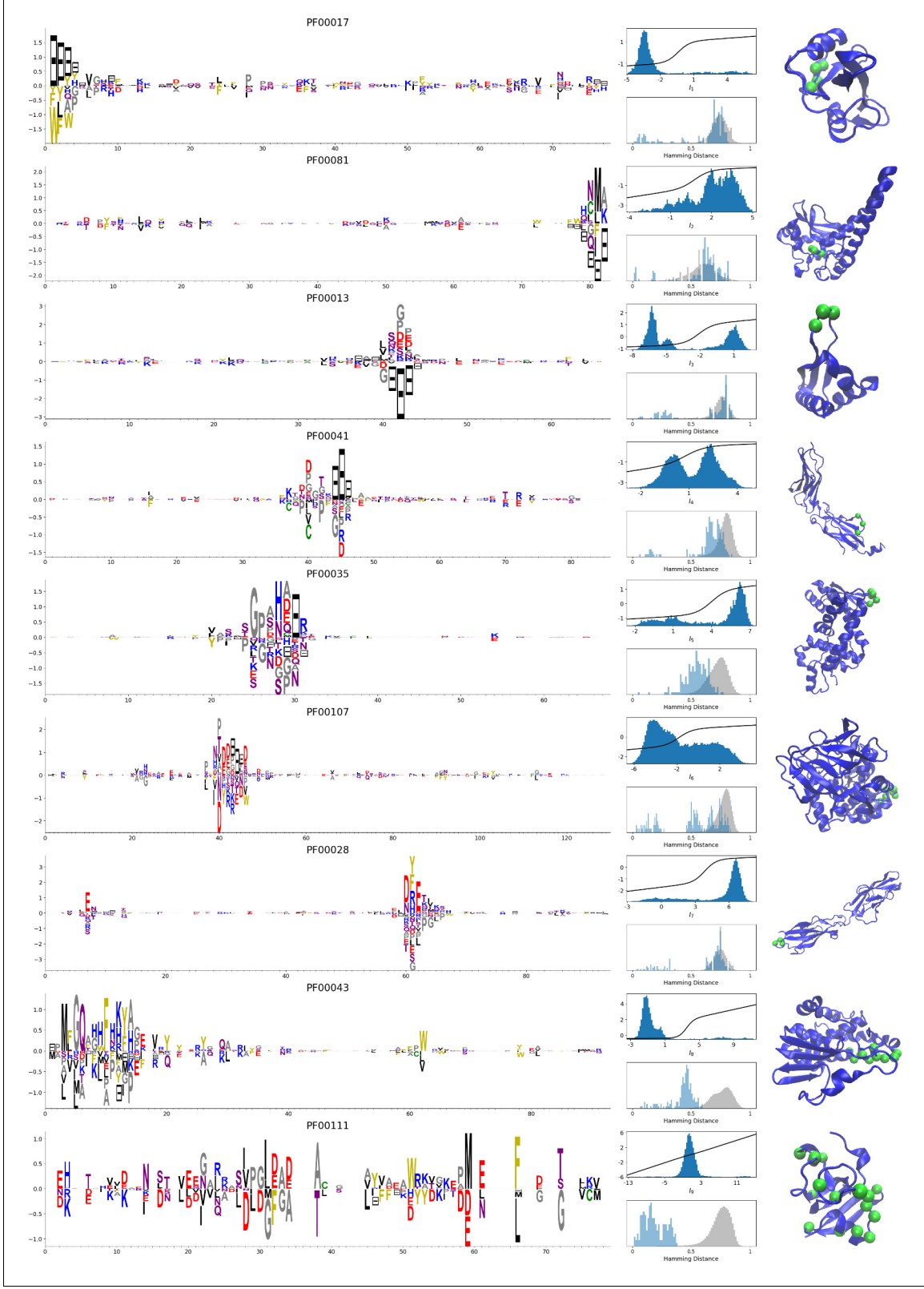

**Figure 10.** Representative weights of the protein families selected in *Ekeberg et al. (2014)*. RBM parameters: $\lambda_1^2 = 0.25$, $M = 0.05 \times N \times 20$. The format is the same as that used in *Figures 2B*, *3B* and *4B*. Weights are ordered by similarity (from top to bottom): SH2 domain (PF00017), superoxide dismutase (PF00081), K homology domain (PF00013), fibronectin type III domain (PF00041), double-stranded RNA-binding motif (PF00035), zinc-binding dehydrogenase (PF00107), cadherin (PF00028), glutathione S-transferase, C-terminal domain (PF00043), and 2Fe-2S iron-sulfur cluster binding domain

*Figure 10 continued on next page*

Figure 10 continued

(PF00111). Green spheres show the sites that carry the largest weights on the 3D folds (in order, PDB: 1o47, 3bfr, 1wvn, 1bqu, 1o0w, 1a71, 2o72, 6gsu, 1a70). The ten weights with largest norms in each family are shown in **Supplementary files 5–6**.

DOI: https://doi.org/10.7554/eLife.39397.012

protein domains or proteins, but could be a generic property as suggested by our study of 16 additional families (*Figure 9 and 10*).

Despite the algorithmic improvements developed in the present work (see 'Materials and methods'), training RBM is challenging as it requires intensive sampling. Generative models that are alternatives to RBM, and that do not require Markov Chain sampling, exist in machine learning; they include Generative Adversarial Networks (*Goodfellow et al., 2014*) and Variational Auto–encoders (VAE) (*Kingma and Welling, 2013*. VAE were recently applied to protein sequence data for fitness prediction (*Sinai et al., 2017*; *Riesselman et al., 2018*. Our work differs in several impo rtant points: our RBM is an extension of direct-based coupling approaches, requires much less hidden units (about 10 to 50 times fewer than were used in *Sinai et al., 2017* and *Riesselman et al., 2018*), has a simple architecture with two layers carrying sequences and representations, infers interpretable weights with biological relevance, and can be easily tweaked to design sequences with desired statistical properties. We have shown that RBM can successfully model small domains (of a few tens of amino acids) as well as much longer proteins (of several hundreds of residues). The reason is that, even for very large proteins, the computational effort can be controlled through the number $M$ of hidden units (see 'Materials and methods' for discussion about the running time of our learning algorithm). Choosing moderate values of $M$ makes the number of parameters to be learned reasonable and avoids overfitting, yet allows for the discovery of important functional and structural features. It is, however, unclear how $M$ should scale with $N$ to unveil 'all' the functional features of very complex and rich proteins (such as Hsp70).

From a computational biology point of view, RBM unifies and extends previous approaches in the context of protein coevolutionary analysis. From the one hand, the features extracted by RBM identify 'collective modes' that control the biological functionalities of the protein, in a similar way to the so-called sectors extracted by statistical coupling analysis (*Halabi et al., 2009*). However, contrary to sectors, the collective modes are not disjoint: a site may participate in different features, depending on the value of the residue it carries. On the other hand, RBM coincide with direct-coupling analysis (*Morcos et al., 2011* when the potential $\mathcal{U}(h)$ is quadratic in $h$. For non-quadratic potentials $\mathcal{U}$, couplings to all orders between the visible units are present. The presence of high-order interactions allows for a significantly better description of gap modes *Feinauer et al., 2014*, of multiple long-range couplings due to ligand binding, and of outliers sequences (*Appendix 1—figure 5*). Our dReLU RBM model offers an efficient way to go beyond pairwise coupling models, without an explosion in the number of interaction parameters to be inferred, as all high-order interactions (whose number, $\sim q^N$, is exponentially large in $N$) are effectively generated from the same $M \times N \times q$ weights $w_{i\mu}(v)$. RBM also outperforms the Hopfield-Potts framework *Cocco et al., 2013*, an approach previously introduced to capture both collective and localized structural modes. Hopfield-Potts 'patterns' were derived with no sparsity regularization and within the mean-field approximation, which made the Hopfield-Potts model insufficiently accurate for sequence design (see *Appendix 1—figures 14–18*).

The weights shown in *Figures 2B*, *3B* and *4B* are stable with respect to subsampling (*Appendix 1—figure 13*) and could be unambiguously interpreted and related to existing literature. However, the biological significance of some of the inferred features remains unclear, and would require experimental investigation. Similarly, the capability of RBM to design new functional sequences need experimental validation besides the comparison with past design experiments (*Figure 5E*) and the benchmarking on in silico proteins (*Figure 7*). Although recombining different parts of natural proteins sequences from different organisms is a well recognized procedure for protein design (*Stemmer, 1994*; *Khersonsky and Fleishman, 2016*, RBM innovates in a crucial aspect. Traditional approaches cut sequences into fragments at fixed positions on the basis of secondary structure considerations, but such parts are learned and need not be contiguous along the primary sequence in RBM models. We believe that protein design with detailed computational modeling methods, such

as Rosetta (**Simons et al., 1997**; **Khersonsky and Fleishman, 2016**, could be efficiently guided by our RBM-based approach, in much the same way as protein folding greatly benefited from the inclusion of long-range contacts found by direct-coupling analysis (**Marks et al., 2011**; **Hopf et al., 2012**.

Future projects include developing systematic methods for identifying function-determining sites, and analyzing more protein families. As suggested by the analysis of the 16 families shown in **Figure 9 and 10**, such a study could help to establish a general classification of motifs into broad classes with structural or functional relevance, shared by distinct proteins. In addition, it would be very interesting to use RBM to determine evolutionary paths between two, or more, protein sequences in the same family, but with distinct phenotypes. In principle, RBM could reveal how functionalities continuously change along the paths, and could provide a measure of viability of intermediary sequences.

## Materials and methods

### Data preprocessing

We use the PFAM sequence alignments of the V31.0 release (March 2017) for both Kunitz (PF00014) and WW (PF00397) domains. All columns with insertions are discarded, then duplicate sequences are removed. We are left with, respectively, $N = 53$ sites and $B = 8062$ unique sequences for Kunitz, and $N = 31$ and $B = 7503$ for WW; each site can carry $q = 21$ different symbols. To correct for the heterogeneous sampling of the sequence space, a reweighting procedure is applied: each sequence $\mathbf{v}^\ell$ with $\ell = 1, ..., B$ is assigned a weight $w_\ell$ equal to the inverse of the number of sequences with more than 90% amino-acid identity (including itself). In all that follows, the average over the sequence data of a function $f$ is defined as

$$\langle f(\mathbf{v}) \rangle_{MSA} = \left( \sum_{\ell=1}^{B} w_\ell f(\mathbf{v}^\ell) \right) / \left( \sum_{\ell=1}^{B} w_\ell \right). \tag{7}$$

### Learning procedure

Objective function and gradients

Training is performed by maximizing, through stochastic gradient ascent, the difference between the log-probability of the sequences in the MSA and the regularization costs,

$$\langle \log P(\mathbf{v}) \rangle_{MSA} - \frac{\lambda_f}{2} \sum_{i,v} g_i(v)^2 - \frac{\lambda_1^2}{2qN} \sum_{\mu} \left( \sum_{i,v} |w_{i\mu}(v)| \right)^2, \tag{8}$$

Regularization terms include a standard $L_2$ penalty for the potentials acting on the visible units, and a custom $L_2/L_1$ penalty for the weights. The latter penalty corresponds to an effective $L_1$ regularization with an adaptive strength that increases with the weights, thus promoting homogeneity among hidden units. (This can be seen from the gradient of the regularization term, which reads $\lambda_1^2 \left( \sum_{i,v'} |w_{i\mu}(v')|/qN \right) \text{sign}(w_{i\mu}(v))$.) Besides, it prevents hidden units from ending up entirely disconnected ($w_{i\mu}(v) = 0 \, \forall i, v$), and makes the determination of the penalty strength $\lambda_1^2$ more robust (see **Appendix 1—figure 2**).

According to **Equation (5)**, the probability of a sequence $\mathbf{v}$ can be written as,

$$P(\mathbf{v}) = e^{-E_{\text{eff}}(\mathbf{v})} / \left( \sum_{\mathbf{v}'} e^{-E_{\text{eff}}(\mathbf{v}')} \right), \quad \text{where} \quad E_{\text{eff}}(\mathbf{v}) = -\sum_{i=1}^{N} g_i(v_i) - \sum_{\mu=1}^{M} \Gamma(I_\mu(\mathbf{v})) \tag{9}$$

is the effective 'energy' of the sequence, which depends on all the model parameters. The gradient of $\langle \log P(\mathbf{v}) \rangle_{MSA}$ over one of these parameters, denoted generically by $\psi$, is therefore

$$\frac{\partial}{\partial \psi} \langle \log P(\mathbf{v}) \rangle_{MSA} = \sum_{\mathbf{v}} P(\mathbf{v}) \frac{\partial E_{\text{eff}}}{\partial \psi}(\mathbf{v}) - \langle \frac{\partial E_{\text{eff}}}{\partial \psi}(\mathbf{v}) \rangle_{MSA}. \tag{10}$$

Hence, the gradient is the difference between the average values of the derivative of $E_{eff}$ with respect to $\psi$ over the model and the data distributions.

## Moment evaluation

Several methods have been developed to evaluate the model average in the gradient ( see *Equation (10)) Fischer and Igel, 2012*. The naive approach is to run for each gradient iteration a full Markov Chain Monte Carlo (MCMC) simulation of the RBM until the samples reach equilibrium, then use these samples to compute the model average *Ackley et al., 1987*. A more efficient approach is the Persistent Constrastive Divergence *Tieleman, 2008*: the samples obtained from the previous simulation are used to initialize for the next MCMC simulation, and only a small number of Gibbs updates ($N_{MC} \sim 10$) are performed between each gradient evaluation. If the model parameters evolve slowly, the samples are always at equilibrium, and we obtain the same accuracy as that provided the naive approach at a fraction of the computational cost. In practice, the Persistent Contrastive Divergence (PCD) algorithm succeeds if the mixing rate of the Markov Chain — which depends on the nature and dimension of the data, and the model parameters — is fast enough. In our training sessions, PCD proved sufficient to learn relevant features and good generative models for small proteins and regularized RBM.

## Stochastic gradient ascent

The optimization is carried out by Stochastic Gradient Ascent. At each step, the gradient is evaluated using a mini-batch of the data, as well as a small number of MCMC configurations. In most of our training sessions, we used the same batch size (=100) for both sets. The model is initialized as follows:

- Weights $w_{i\mu}(v)$ are randomly and independently drawn from a Gaussian distribution with zero mean and variance equal to $\frac{0.1}{N}$. The scaling factor $\frac{1}{N}$ ensures that the initial input distribution has variance of the order of 1.
- The potentials $g_i(v)$ are given their values in the independent-site model: $g_i(v) = \log \langle \delta_{v_i,v} \rangle_{\text{MSA}}$, where $\delta$ denotes the Kronecker function.
- For all hidden-unit potentials, we set $\gamma_+ = \gamma_- = 1$, $\theta_+ = \theta_- = 0$.

The learning rate is initially set to $0.1$, and decays exponentially after a fraction of the total training time (e.g. 50%) until it reaches a final, small value, for example $10^{-4}$.

## Dynamic reparametrization

For Gaussian and dReLU potentials, there is a redundancy between the slope of the hidden unit average activity and the global amplitude of the weight vector. Indeed, for the Gaussian potential, the model distribution is invariant under rescaling transformations $\gamma_\mu \to \lambda^2 \gamma_\mu$, $w_{i\mu} \to \lambda w_{i\mu}$, $\theta_\mu \to \lambda \theta_\mu$ and offset transformation $\theta_\mu \to \theta_\mu + K_\mu$, $g_i \to g_i - \sum_\mu w_{i\mu} \frac{K_\mu}{\gamma_\mu}$. Though we can set $\gamma_\mu = 1$, $\theta_\mu = 0 \, \forall \mu$ without loss of generality, it can lead either to numerical instability (at high learning rate) or slow learning (at low learning rate). A significantly better choice is to adjust the slope and offset dynamically so that $\langle h_\mu \rangle \sim 0$ and $\text{Var}(h_\mu) \sim 1$ at all times. This new approach, reminiscent of batch normalization for deep networks, is implemented in the training algorithm released with this work. Detailed equations and benchmarks will be available online soon.

## Gauge choice

Since the conditional probability *Equation 4* is normalized, the transformations $g_i(v) \to g_i(v) + \lambda_i$ and $w_{i\mu}(v) \to w_{i\mu}(v) + K_{i\mu}$ leave the conditional probability invariant. We choose the zero-sum gauges, defined by $\sum_v g_i(v) = 0$, $\sum_v w_{i\mu}(v) = 0$. Since the regularization penalties over the fields and weight depend on the gauge choice, the gauge must be enforced throughout all training and not only at the end. The updates on the fields leave the gauge invariant, so the transformation $g_i(v) \to g_i(v) - \frac{1}{q} \sum_{v'} g_i(v')$ can be used only once, after initialization. On the other hand, this is not the case for the updates on the weights, so the transformation $w_{i\mu}(v) - \frac{1}{q} \sum_{v'} w_{i\mu}(v')$ must be applied after each gradient update.

## Evaluating the partition function

Evaluating $P(\mathbf{v})$ requires knowledge of the partition function $Z = \sum_{\mathbf{v}} \exp(-E_{\text{eff}}(\mathbf{v}))$ (see denominator in *Equation (9))*. The later expression, which involves summing over $q^N$ terms is not tractable. Instead, we estimate $Z$ using the Annealed Importance Sampling algorithm (AIS) *Neal, 2001*; *Salakhutdinov and Murray, 2008*. Briefly, the idea is to estimate partition function ratios. Let $P_1(\mathbf{v}) = \frac{P_1^*(\mathbf{v})}{Z_1}$, $P_0 = \frac{P_0^*(\mathbf{v})}{Z_0}$ be two probability distributions with partition functions $Z_1$, $Z_0$. Then:

$$\left\langle \frac{P_1^*(\mathbf{v})}{P_0^*(\mathbf{v})} \right\rangle_{\mathbf{v} \sim P_0} = \sum_{\mathbf{v}} \frac{P_1^*(\mathbf{v})}{P_0^*(\mathbf{v})} \frac{P_0^*(\mathbf{v})}{Z_0} = \frac{1}{Z_0} \sum_{\mathbf{v}} P_1^*(\mathbf{v}) = \frac{Z_1}{Z_0} \tag{11}$$

Therefore, provided that $Z_0$ is known (e.g. if $P_0$ is an independent model with no couplings), one can in principle estimate $Z_1$ through Monte Carlo sampling. The difficulty lies in the variance of the estimator: if $P_1$, $P_0$ are very different from one another, then some configurations can be very likely for $P_1$ and have very low probability with $P_0$; these configurations almost never appear in the Monte Carlo estimate of $\langle . \rangle$, but the probability ratio can be exponentially large. In Annealed Importance Sampling, we address this problem by constructing a continuous path of interpolating distributions $P_\beta(\mathbf{v}) = P_1(\mathbf{v})^\beta P_0(\mathbf{v})^{1-\beta}$, and estimate $Z_1$ as a product of the ratios of the partition functions:

$$Z_1 = \frac{Z_1}{Z_{\beta_{l_{max}}}} \frac{Z_{\beta_{l_{max}-1}}}{Z_{\beta_{l_{max}-2}}} \dots \frac{Z_{\beta_1}}{Z_0} \times Z_0 \ , \tag{12}$$

where we choose a linear set of interpolating inverse temperatures of the form $\beta_l = \frac{l}{l_{max}}$. To evaluate the successive expectations, we use a fixed number $C$ of samples initially drawn from $P_0$, and gradually anneal them from $P_0$ to $P_1$ by successive applications of Gibbs sampling at $P_\beta$. Moreover, all computations are done in logarithmic scales for numerical stability purposes: we estimate $\log \frac{Z_1}{Z_0} \approx \left\langle \log \frac{P_1^*(\mathbf{v})}{P_0^*(\mathbf{v})} \right\rangle_{\mathbf{v} \sim P_0}$, which is justified if $P_1$ and $P_0$ are close. In practice, we used $C = 20$ chains, $n_\beta = 5 \times 10^4$ steps. For the initial distribution $P_0$, we take the closest (in terms of KL divergence) independent model to the data distribution $P_{MSA}$. The visible layer fields are those of the independent model inferred from the MSA, and the weights are $\mathbf{w}^{\beta=0} = 0$. For the hidden potential values, we infer the parameters from the statistics of the hidden layer activity conditioned to the data.

## Explicit formula for sampling and training RBM

Training, sampling and computing the probability of sequences with RBM requires: (1) sampling from $P(\mathbf{v}|\mathbf{h})$, (2) sampling from $P(\mathbf{h}|\mathbf{v})$, and (3) evaluating the effective energy $E_{\text{eff}}(\mathbf{v})$ and its derivatives. This is done as follows:

1. Each sequence site $i$ is encoded as a categorical variable taking integer values $v_i \in [0, 20]$, with each integer corresponding to one of the 20 amino-acids + 1 gap. Similarly, the fields and weights are encoded as a $N \times 21$ matrix and a $M \times N \times 21$ tensor, respectively. Owing to the bipartite structure of the graph, $P(\mathbf{v}|\mathbf{h}) = \prod_i P(\mathbf{v}_i|\mathbf{h})$ (see *Equation (4))*. Therefore, sampling from $P(\mathbf{v}|\mathbf{h})$ is done in three steps: compute the inputs received from the hidden layer, then the conditional probabilities $P(v_i|\mathbf{h})$ given the inputs, and sample each visible unit independently the corresponding conditional distributions.
2. The conditional probability $P(\mathbf{h}|\mathbf{v})$ factorizes. Given a visible configuration $\mathbf{v}$, each hidden unit is sampled independently from the others via $P(h_\mu|\mathbf{v})$ (see *Equation (3))*. For a quadratic potential $\mathcal{U}(h) = \frac{1}{2}\gamma h^2 + \theta h$, this conditional distribution is Gaussian. For the dReLU potential $\mathcal{U}(h)$ in *Equation (6)*, we introduce first

$$\Phi(x) = \exp(\frac{x^2}{2}) \left[ 1 - \text{erf}(\frac{x}{\sqrt{2}}) \right] \sqrt{\frac{\pi}{2}}$$

Some useful properties of $\Phi$ are:
- $\Phi(x) \sim_{x \to -\infty} \exp(\frac{x^2}{2})\sqrt{2\pi}$
- $\Phi(x) \sim_{x \to \infty} \frac{1}{x} - \frac{1}{x^3} + \frac{3}{x^5} + \mathcal{O}(\frac{1}{x^7})$
- $\Phi'(x) = x\Phi(x) - 1$

To avoid numerical issues, $\Phi$ is computed in practice with its definition for $x < 5$ and with its asymptotic expansion otherwise. We also write $\mathcal{TN}(\mu, \sigma^2, a, b)$, the truncated Gaussian distribution of mode $\mu$, width $\sigma$ and support $[a, b]$.

Then, $P(h|I)$ is given by a mixture of two truncated Gaussians:

$$P(h|I) = p^+ \, \mathcal{TN}\left(\frac{I - \theta^+}{\gamma_+}, \frac{1}{\gamma_+}, 0, +\infty\right) + p^- \, \mathcal{TN}\left(\mu = \frac{I - \theta^-}{\gamma^-}, \sigma^2 = \frac{1}{\gamma^-}, -\infty, 0\right) \tag{13}$$

where $Z^\pm = \Phi\left(\frac{\mp(I - \theta^\pm)}{\sqrt{\gamma^\pm}}\right)\frac{1}{\sqrt{\gamma^\pm}}$, and $p^\pm = \frac{Z^\pm}{Z^+ + Z^-}$.

3. Evaluating $E_{\text{eff}}$ and its derivatives requires an explicit expression for the cumulant–generating function $\Gamma(I)$. For quadratic potentials, $\Gamma(I)$ is quadratic too. For dReLU potentials, we have $\Gamma(I) = \log(Z^+ + Z^-)$, where $Z^\pm$ is defined above.

## Computational complexity

The computational complexity is of the order of $M \times N \times B$, with more accurate variants taking more time. The algorithm scales reasonably to large protein sizes, and was tested successfully for $N$ up to $\sim 700$, taking in the order of 1–2 days on an Intel Xeon Phi processor with $2 \times 28$ cores.

## Sampling procedure

Sampling from $P$ in **Equation (5)** is done with Markov Chain Monte Carlo methods, with the standard alternate Gibbs sampler described in the main text and in **Fischer and Igel (2012)**. Conditional sampling, that is sampling from $P(\mathbf{v}|h_\mu = h_\mu^c)$, is straightforward with RBM: it is achieved by the same Gibbs sampler while keeping $h_\mu$ fixed.

The RBM architecture can be modified to generate sequences with high probabilities (as in **Figure 5E&F**). The trick is to duplicate the hidden units, the weights, and the local potentials acting on the visible units, as shown in **Figure 11**. By doing so, the sequences $\mathbf{v}$ are distributed according to

$$P_2(\mathbf{v}) \propto \int \prod_\mu dh_{\mu 1}\, dh_{\mu 2}\, P(\mathbf{v}|\mathbf{h}_1)\, P(\mathbf{v}|\mathbf{h}_2) = P(\mathbf{v})^2 . \tag{14}$$

Hence, with the duplicated RBM, sequences with high probabilities in the original RBM model are given a boost when compared to low-probability sequences (**Figure 11**). Note that more subtle biases can be introduced by duplicating some (but not all) of the hidden units in order to give more importance in the sampling to the associated statistical features.

## Contact map estimation

RBM can be used for contact prediction in a manner similar to pairwise coupling models, after derivation of an effective coupling matrix $J_{ij}^{\text{eff}}(a, b)$. Consider a sequence $\mathbf{v}$, and two sites $i, j$. Define the set of mutated sequences $\mathbf{v}^{a,b}$ with amino acid content: $v_k^{a,b} = v_k$ if $k \neq i, j$, $a$ if $k = i$, $b$ if $k = j$ (**Figure 6A**). The differential likelihood ratio

$$\Delta\Delta R_{ij}(\mathbf{v}; a, a', b, b') \equiv \log\left[\frac{P(\mathbf{v}^{a,b}) P(\mathbf{v}^{a',b'})}{P(\mathbf{v}^{a',b}) P(\mathbf{v}^{a,b'})}\right], \tag{15}$$

where $P$ is the marginal distribution in **Equation (5)**, measures epistatic contributions to the double mutation $a \to a'$ and $b \to b'$ on sites $i$ and $j$, respectively, in the background defined by sequence $\mathbf{v}$ (see **Figure 6A**). The effective coupling matrix is then defined as

$$J_{ij}^{\text{eff}}(a, b) = \left\langle \frac{1}{q^2} \sum_{a', b'} \Delta\Delta R_{ij}(\mathbf{v}; a, a', b, b') \right\rangle_{MSA}, \tag{16}$$

where the average is taken over the sequences $\mathbf{v}$ in the MSA. For a pairwise model, $\Delta\Delta R_{ij}$ does not depend on the background sequence $\mathbf{v}$, and **Equation (16)** coincides with the true coupling in the zero-sum gauge. Contact estimators are based on the Frobenius norms of $J^{\text{eff}}$, with the Average Product Correction (see **Cocco et al., 2018**).

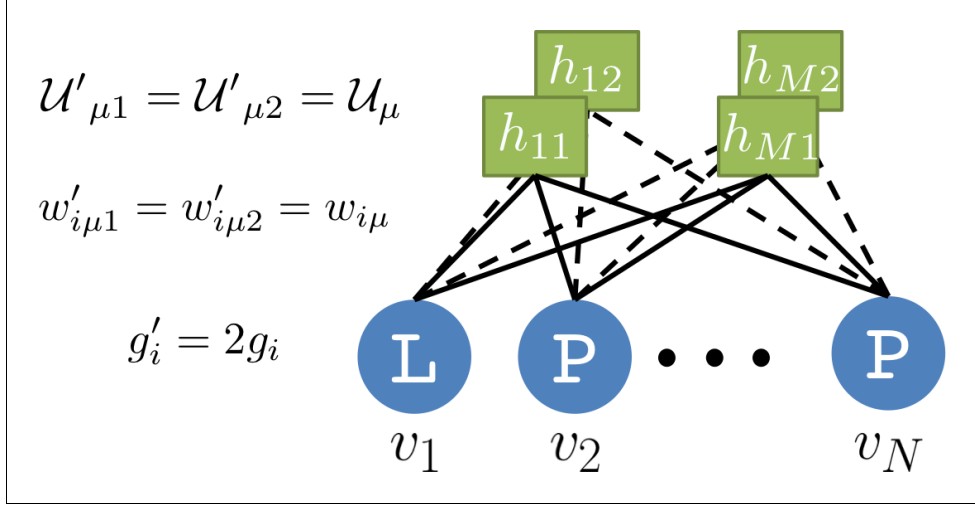

**Figure 11.** Duplicate RBM for biasing sampling toward high-probability sequences. Visible-unit configurations $\mathbf{v}$ are sampled from $P_2(\mathbf{v}) \propto P(\mathbf{v})^2$.

DOI: https://doi.org/10.7554/eLife.39397.013

## Code availability

The Python 2.7 package for training and visualizing RBMs, which was used to obtain the results reported in this work, is available at https://github.com/jertubiana/ProteinMotifRBM (**Tubiana, 2019**; copy archived at https://github.com/elifesciences-publications/ProteinMotifRBM). In addition, Jupyter notebooks are provided for reproducing most of the figures in this article.

## Acknowledgments

We thank D Chatenay for useful comments on the manuscript and L Posani for his help on lattice proteins.

## Additional information

### Funding

| Funder | Grant reference number | Author |
| --- | --- | --- |
| École Normale Supérieure | Allocation Specifique | Jérôme Tubiana |
| Agence Nationale de la Recherche | CE30-0021-01 RBMPro | Simona Cocco Rémi Monasson |

The funders had no role in study design, data collection and interpretation, or the decision to submit the work for publication.

### Author contributions

Jérôme Tubiana, Conceptualization, Resources, Data curation, Software, Formal analysis, Validation, Investigation, Visualization, Methodology, Writing—original draft, Writing—review and editing; Simona Cocco, Conceptualization, Formal analysis, Supervision, Funding acquisition, Validation, Investigation, Methodology, Writing—original draft, Project administration, Writing—review and editing; Rémi Monasson, Conceptualization, Formal analysis, Supervision, Funding acquisition, Validation, Investigation, Visualization, Methodology, Writing—original draft, Project administration, Writing—review and editing

**Author ORCIDs**
Jérôme Tubiana (ID) http://orcid.org/0000-0001-8878-5620
Simona Cocco (ID) http://orcid.org/0000-0002-1852-7789
Rémi Monasson (ID) http://orcid.org/0000-0002-4459-0204

**Decision letter and Author response**
Decision letter https://doi.org/10.7554/eLife.39397.090
Author response https://doi.org/10.7554/eLife.39397.091

## Additional files

### Supplementary files

• Supplementary file 1. Weight logo for all hidden units inferred from the Kunitz domain MSA.
DOI: https://doi.org/10.7554/eLife.39397.014

• Supplementary file 2. Weight logo for all hidden units inferred from the WW domain MSA.
DOI: https://doi.org/10.7554/eLife.39397.015

• Supplementary file 3. Weight logo for all hidden units inferred from the LP MSA.
DOI: https://doi.org/10.7554/eLife.39397.016

• Supplementary file 4. Weight logo of 12 Hopfield-Potts pattern inferred from the Hsp70 protein MSA. The format is the same as that used for *Appendix 1—figures 14–16*.
DOI: https://doi.org/10.7554/eLife.39397.017

• Supplementary file 5. Weight logo and associated structures of the 10 weights with highest norms, excluding the gap modes for each of the 16 additional domains shown in *Figure 9*.
DOI: https://doi.org/10.7554/eLife.39397.018

• Supplementary file 6. Weight logo and associated structures of the 10 sparse (i.e. within the 30% most sparse weights of the RBM) weights with highest norms, excluding the gap modes for each of the 16 additional domains shown in *Figure 9*.
DOI: https://doi.org/10.7554/eLife.39397.019

### Data availability

The Python 2.7 package for training and visualizing RBMs, used to obtained the results reported in this work, is available at https://github.com/jertubiana/ProteinMotifRBM (copy archived at https://github.com/elifesciences-publications/ProteinMotifRBM). It can be readily used for any protein family. Moreover, all four multiple sequence alignments presented in the text, as well as the code for reproducing each panel are also included. Jupyter notebooks are provided for reproducing most figures of the article.

The following previously published datasets were used:

| Author(s) | Year | Dataset title | Dataset URL | Database and Identifier |
|---|---|---|---|---|
| Merigeau K, Arnoux B, Ducruix A | 1997 | THE 1.2 ANGSTROM STRUCTURE OF KUNITZ TYPE DOMAIN C5 | https://www.rcsb.org/structure/2KNT | Protein Data Bank, 2KNT |
| Macias MJ | 2000 | PROTOTYPE WW domain | https://www.rcsb.org/structure/1e0m | Protein Data Bank, 1E0M |
| Zuiderweg ERP, Bertelsen EB | 2009 | NMR-RDC / XRAY structure of E. coli HSP70 (DNAK) chaperone (1-605) complexed with ADP and substrate | https://www.rcsb.org/structure/2KHO | Protein Data Bank, 2KHO |
| Qi R, Sarbeng EB, Liu Q, Le KQ, Xu X | 2013 | Allosteric opening of the polypeptide-binding site when an Hsp70 binds ATP | https://www.rcsb.org/structure/4JNE | Protein Data Bank, 4JNE |
| Gaboriaud C, Rossi V, Bally I, Arlaud G | 2001 | CRYSTAL STRUCTURE OF THE CATALYTIC DOMAIN OF HUMAN COMPLEMENT C1S PROTEASE | https://www.rcsb.org/structure/1ELV | Protein Data Bank, 1ELV |
| Stamler RJ, Kappe G, Boelens WC, Slingsby C | 2005 | CRYSTAL STRUCTURE AND ASSEMBLY OF TSP36, A METAZOAN SMALL HEAT SHOCK | https://www.rcsb.org/structure/2BOL | Protein Data Bank, 2BOL |

| | | PROTEIN | | |
|---|---|---|---|---|
| Camara-Artigas A, Luque I, Ruiz-Sanz J, Mateo PL, Martin-Garcia JM | 2007 | Yes SH3 domain | https://www.rcsb.org/structure/2HDA | Protein Data Bank, 2HDA |
| Jauch R | 2008 | Crystal Structure of the Nanog Homeodomain | https://www.rcsb.org/structure/2VI6 | Protein Data Bank, 2VI6 |
| Baumann H, Paulsen K, Kovacs H, Berglund H, Wright APH, Gustafsson J-A, Hard T | 1994 | REFINED SOLUTION STRUCTURE OF THE GLUCOCORTICOID RECEPTOR DNA-BINDING DOMAIN | https://www.rcsb.org/structure/1GDC | Protein Data Bank, 1GDC |
| Kim C | 2009 | Crystal structure of a complex between the catalytic and regulatory (RI{alpha}) subunits of PKA | https://www.rcsb.org/structure/3FHI | Protein Data Bank, 3FHI |
| Wang X, Hall TMT | 2001 | CRYSTAL STRUCTURE OF HUD AND AU-RICH ELEMENT OF THE TUMOR NECROSIS FACTOR ALPHA RNA | https://www.rcsb.org/structure/1G2E | Protein Data Bank, 1G2E |
| Lange G, Loenze P, Liesum A | 2004 | CRYSTAL STRUCTURE OF SH2 IN COMPLEX WITH RU82209 | https://www.rcsb.org/structure/1O47 | Protein Data Bank, 1O47 |
| He Y-X, Zhao M-X, Zhou C | 2008 | The crystal structure of Sod2 from Saccharomyces cerevisiae | https://www.rcsb.org/structure/3BFR | Protein Data Bank, 3BFR |
| Wilce MCJ, Wilce JA, Sidiqu M | 2005 | Crystal Structure of domain 3 of human alpha polyC binding protein | https://www.rcsb.org/structure/1WVN | Protein Data Bank, 1WVN |
| Bravo J, Staunton D, Heath JK, Jones EY | 1998 | CYTOKYNE-BINDING REGION OF GP130 | https://www.rcsb.org/structure/1BQU | Protein Data Bank, 1BQU |
| Joint Center for Structural Genomics | 2002 | Crystal structure of Ribonuclease III (TM1102) from Thermotoga maritima at 2.0 A resolution | https://www.rcsb.org/structure/1O0W | Protein Data Bank, 1O0W |
| Colby TD, Bahnson BJ, Chin JK, Klinman JP, Goldstein BM | 1998 | TERNARY COMPLEX OF AN ACTIVE SITE DOUBLE MUTANT OF HORSE LIVER ALCOHOL DEHYDROGENASE, PHE93=>TRP, VAL203=>ALA WITH NAD AND TRIFLUOROETHANOL | https://www.rcsb.org/structure/1A71 | Protein Data Bank, 1A71 |
| Parisini E, Wang J-H | 2007 | Crystal Structure Analysis of human E-cadherin (1-213) | https://www.rcsb.org/structure/2O72 | Protein Data Bank, 2O72 |
| Xiao G, Ji X, Armstrong RN, Gilliland GL | 1996 | FIRST-SPHERE AND SECOND-SPHERE ELECTROSTATIC EFFECTS IN THE ACTIVE SITE OF A CLASS MU GLUTATHIONE TRANSFERASE | https://www.rcsb.org/structure/6gsu | Protein Data Bank, 6GSU |
| Binda C, Coda A, Mattevi A, Aliverti A, Zanetti G | 1998 | SPINACH FERREDOXIN | https://www.rcsb.org/structure/1A70 | Protein Data Bank, 1A70 |

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

## Appendix 1

DOI: https://doi.org/10.7554/eLife.39397.020

## Supporting methods and figures

### Lattice-protein synthetic sequences

LP models have been introduced in the '90 to investigate the uniqueness of folding shared by the majority of real proteins *Shakhnovich and Gutin, 1990*, and have been more recently used to benchmark graphical models inferred from sequence data *Jacquin et al., 2016*). There are $\mathcal{N} = 103,406$ possible folds, that is self-avoiding paths of the 27 amino-acid-long chains, on $3 \times 3 \times 3$ a lattice cube *Shakhnovich and Gutin, 1990*. The probability that the protein sequence $\mathbf{v} = (v_1, v_2, ..., v_{27})$ folds in one of these, say, $S$, is

$$P_{nat}(\mathbf{v}; S) = \frac{e^{-\mathcal{E}(\mathbf{v}; S)}}{\displaystyle\sum_{S'=1}^{\mathcal{N}} e^{-\mathcal{E}(\mathbf{v}; S')}}, \tag{17}$$

where the energy of sequence $\mathbf{v}$ in structure $S$ is given by

$$\mathcal{E}(\mathbf{v}; S) = \sum_{i<j} c_{ij}^{(S)} E(v_i, v_j). \tag{18}$$

In the formula above, $c^{(S)}$ is the contact map: $c_{ij}^{(S)} = 1$ if the pair of sites $ij$ is in contact, that is $i$ and $j$ are nearest neighbors on the lattice, and zero otherwise. The pairwise energy $E(v_i, v_j)$ represents the amino-acid physico-chemical interactions, given by the the Miyazawa-Jernigan (MJ) knowledge-based potential *Miyazawa and Jernigan, 1996*.

A collection of 36,000 sequences that specifically fold on structure $S_A$ (*Figure 7A*) with high probability $P_{nat}(\mathbf{v}; S_A) > 0.995$ were generated by Monte Carlo simulations as described in *Jacquin et al. (2016)*. Like real MSA, Lattice Protein data feature short- and long-range correlations between amino-acid on different sites, as well as high-order interactions that arise from competition between folds *Jacquin et al., 2016*).

### Model selection

We discuss here the choice of parameters (strength of regularization, number of hidden units, shape of hidden-unit potentials, ...) for the RBM used in the main text. Our goal is to achieve good generative performances and to learn biologically interpretable representations. We estimate the accuracy of the fit to the data distribution using the average log-likelihood, divided by the number of visible units $\frac{1}{N} \langle \log P(\mathbf{v}) \rangle_{MSA}$. For visible-unit variables with $q = 21$ possible values (i.e. 20 amino acids + gap symbol), this number typically ranges from $-\log 21 \simeq -3.04$ (uniform distribution) to 0. Evaluating $P(\mathbf{v})$ (Methods *Equation (1)*) requires knowledge of the partition function, $Z = \sum_{\mathbf{v}} \exp\left(\sum_{i=1}^{N} g_i(v_i) + \sum_{\mu=1}^{M} \Gamma_\mu(I_\mu(\mathbf{v}))\right)$ (see section titled 'Evaluating the partition function').

### Number of hidden units

The number of hidden units is critical for the generative performance. We trained RBMs on the Lattice Protein data set for various potentials (Bernoulli, quadratic and dReLU), numbers of hidden units (1–400) and regularizations ($\lambda_1^2 = 0$, $\lambda_1^2 = 0.025$). The likelihood estimation shows that, as expected, the larger $M$, the better the ability to fit the training data (*Appendix 1—figure 1*). Overfitting resulting in a decrease in test set performance may occur for large $M$. For the regularized case, the likelihood saturates at about 100 hidden units. Similar results were obtained for WW (see *Appendix 1—figure 2*).

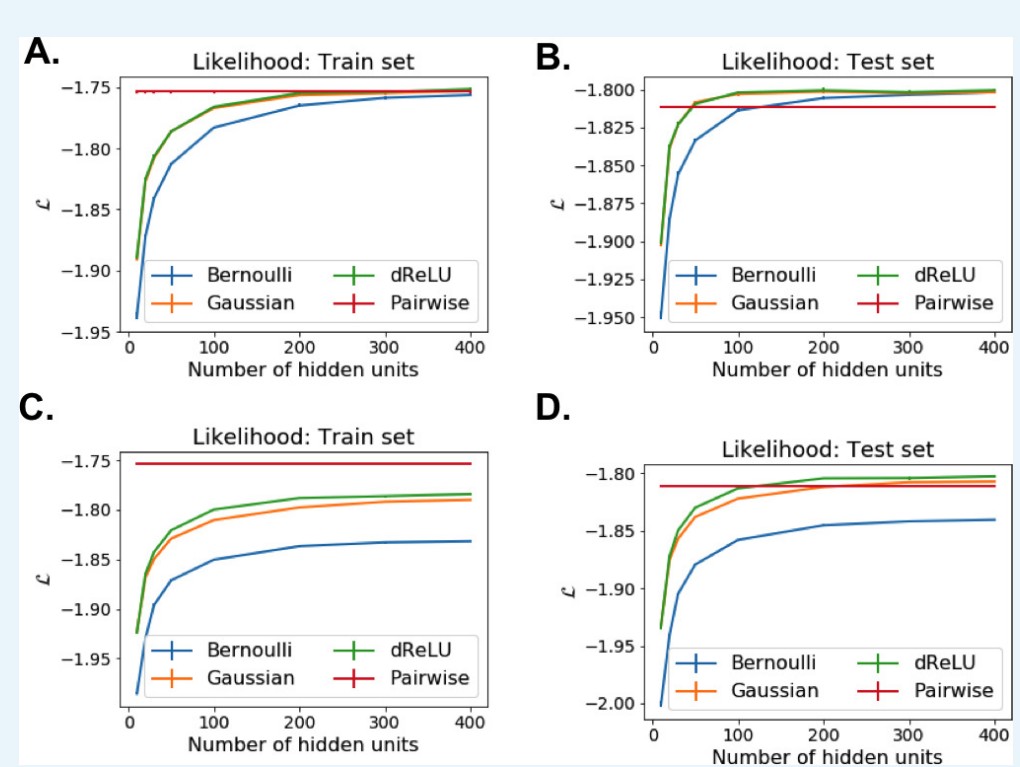

**Appendix 1—figure 1.** Model selection for RBM trained on the Lattice Proteins MSA. Likelihood estimates for various potentials and number of hidden units, evaluated on train and held-out test sets. Top row: without regularization ($\lambda_1^2 = 0$). Bottom row: with regularization ($\lambda_1^2 = 0.025$).

DOI: https://doi.org/10.7554/eLife.39397.021

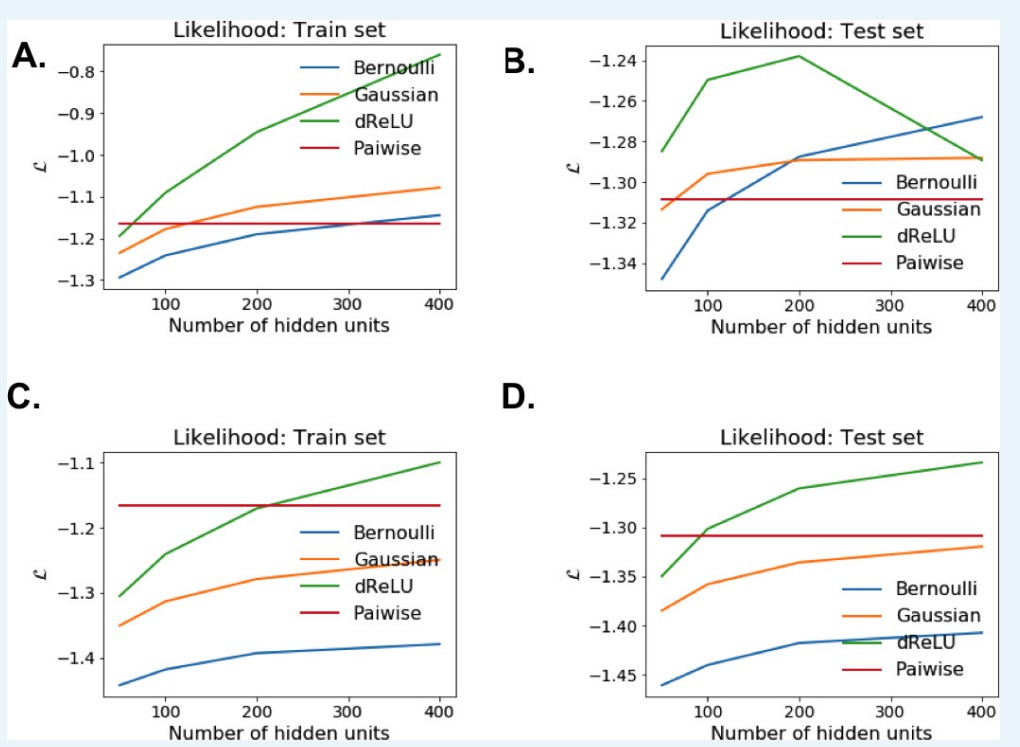

**Appendix 1—figure 2.** Model selection for RBM trained on the WW domain MSA. Likelihood estimates for various potentials and number of hidden units, evaluated on train and held-out test sets. Top row: without regularization ($\lambda_1^2 = 0$). Bottom row: with regularization ($\lambda_1^2 = 0.25$).
DOI: https://doi.org/10.7554/eLife.39397.022

Besides generative performance, the representation also changes as $M$ increases. For very low values of $M$, each hidden unit tries to explain as much covariation as possible and its corresponding weight vector is extended, as in PCA. For larger numbers of hidden units, weights tend to become more sparse; they stabilize at some point, after which new hidden units simply duplicate previous ones.

## Sparse regularization

We first investigate the importance of the sparsifying penalty term. Our study shows that, unlike in the case of MNIST digit data (**Tubiana and Monasson, 2017**), sparsity does not arise naturally from training RBM on protein sequences but requires the introduction of a specific sparsifying regularization (see **Figure 8**). On the one hand, sparse weights, such as those shown in **Figures 2**, **3**, **4** and **7**, are easier to interpret, but, on the other hand, regularization generally leads to a decrease in the generative performance. We show below that the choice of regularization strength used in this work is a good compromise between sparsity and generative performance.

We train several RBM on the Lattice Proteins MSA, with a fixed number of hidden units ($M = 100$), fixed potential, and varying strength of the sparse penalty $\lambda_1^2$ (defined in 'Materials and methods, **Equation (8)**), and evaluate their likelihoods. We repeat the same procedure using the standard $L_1$ regularization ($\lambda_1 \sum_{i,v,\mu} |w_{i\mu}(v)|$) instead of $L_1^2$. Results are shown in **Appendix 1—figure 3**. In both cases, the likelihood on the test set decreases mildly with the regularization strength. However, for $L_1$ regularization, several hidden units become disconnected (i.e. $w_{i\mu}(v) = 0$ for all $i, v$) as we increase the penalty strength. The $L_1^2$ penalty achieves sparse weights without disconnecting hidden units when the penalty is too large, hence it is more robust and requires less fine tuning.

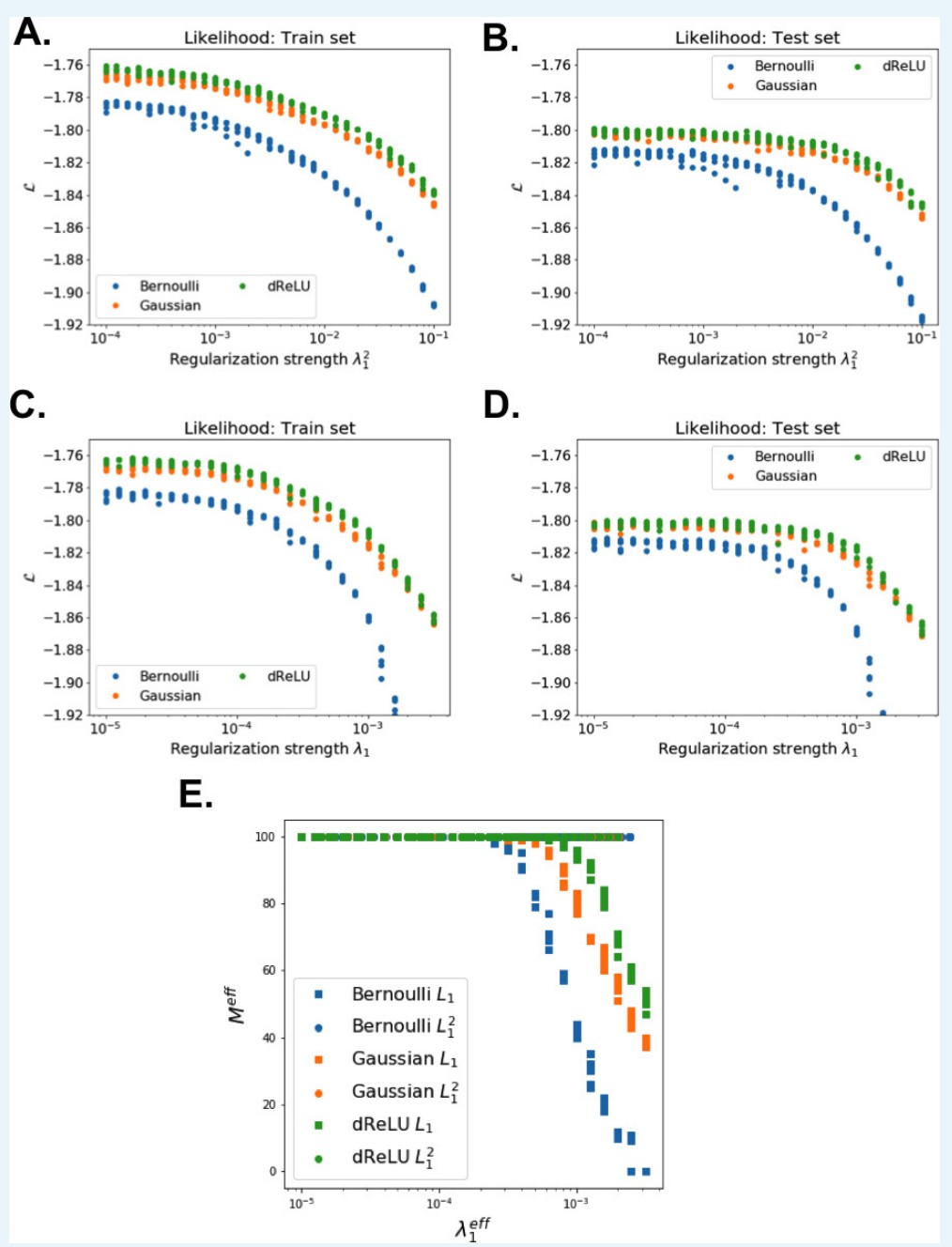

**Appendix 1—figure 3.** Sparsity-generative performance trade-off for RBM trained on the MSA of the Lattice Protein $S_A$. (**A–D**) Likelihood as function of regularization strength, for $L_1^2$ (top) and $L_1$ (bottom) sparse penalties, on train(left) and test (middle) sets. (**E**) Number $M_{eff}$ of connected hidden units (such that $\max_{i,v} |w_{i\mu}(v)| > 0$) against effective strength penalty, for $L_1$ and $L_1^2$ penalties. For $L_1$ penalty, $\lambda_1^{eff} = \lambda_1$; for $L_1^2$, $\lambda_1^{eff} = \lambda_1^2 \frac{1}{NMq} \sum_{\mu,i,v} |w_{\mu i}(v)|$.

DOI: https://doi.org/10.7554/eLife.39397.023

## Hidden-unit potentials

Last, we discuss the choice of the hidden-unit potentials. A priori, the major difference between Bernoulli, quadratic and dReLU potentials are that: (i) the Bernoulli hidden unit takes discrete values whereas quadratic and dReLU hidden units take continuous ones; and (ii) after marginalization, quadratic potentials create pairwise effective interactions whereas Bernoulli and dReLU potentials create non-pairwise ones. It was shown in the context of image

processing and text mining that non-pairwise models are more efficient in practice, and theoretical arguments also highlight the importance of high-order interactions (*Tubiana and Monasson, 2017*).

In terms of generative performance, our results on Lattice Proteins and WW domain MSAs show that, for the same number of parameters, dReLU RBM perform better than Gaussian and Bernoulli RBM. Similar results, not shown, were obtained for the Kunitz domain MSA. Although RBM with Bernoulli hidden units are known to be universal approximators as $M \to \infty$

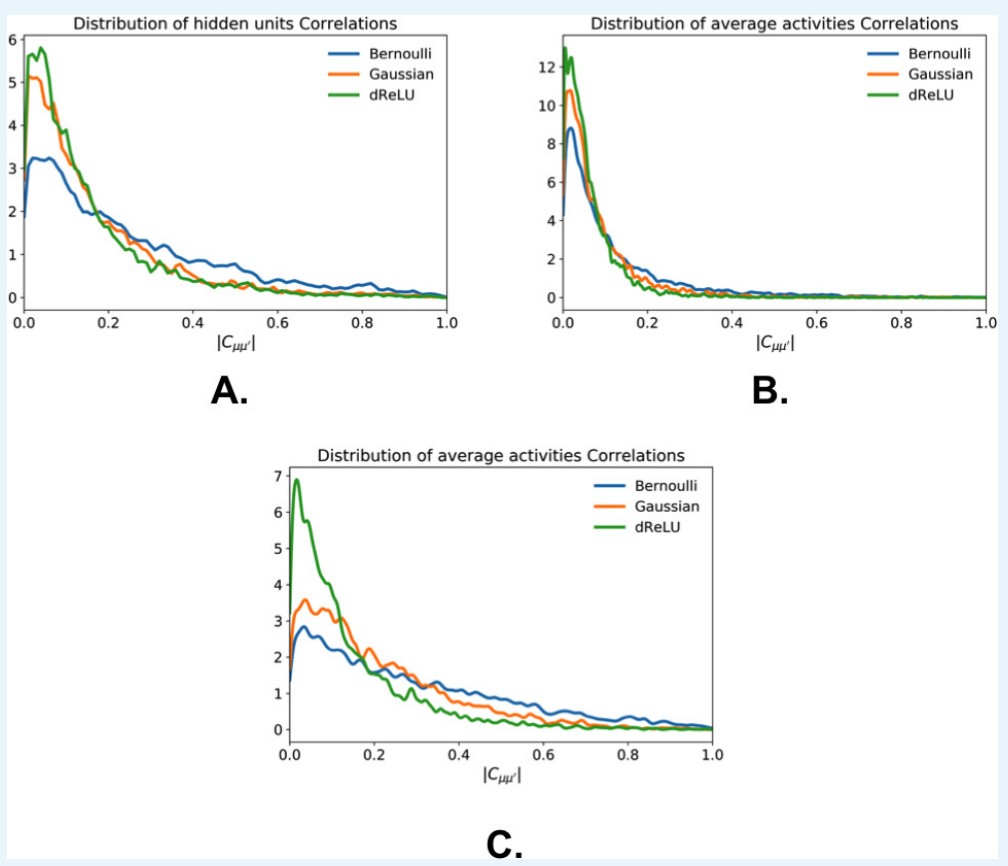

**Appendix 1—figure 4.** Hidden layer representation redundancy as a function of the hidden-unit potentials. Distribution of Pearson correlation coeffcients between hidden-unit average activities, for RBM trained with $M = 100$, on (**a**) Lattice Proteins MSA, (**b**) Kunitz domain MSA, and (**c**) WW domain MSA. Bernoulli RBM feature the highest correlations.
DOI: https://doi.org/10.7554/eLife.39397.024

One of the key aspects that explains the difference in performance between dReLU and Gaussian RBM is the ability of the former to better model 'outlier' sequences, with rare extended features such as Bikunin-AMBP (Weight 5 in the main text, *Figure 2*) or the non-aromatic W28-substitution feature (Weight 3 in the main text, *Figure 3*). Indeed, thanks to the thresholding effect of the average activity, dReLU (unlike quadratic potentials) can account for outliers without altering the distribution for the bulk of the other sequences. To illustrate this property, in *Appendix 1—figure 5*, we compare the likelihoods for all sequences of two RBMs trained with quadratic (resp. dReLU) potentials, $M = 100$, $\lambda_1^2 = 0.25$ on the Kunitz domain MSA. The color coding indicates the degree of anomaly of the sequence, which is obtained as follows:

1. Compute the average activity $h_\mu^l$ of dReLU RBM for all data sequences $\mathbf{v}^l$.

2. Normalize (z-score) each dimension: $\hat{h}_\mu = \frac{h_\mu - \langle h_\mu \rangle_{MSA}}{\sqrt{\mathrm{Var}[h_\mu]_{MSA}}}$.

3. Define:

$$c^I = \arg\max_{\mu} |\hat{h}_{\mu}^I| \qquad (19)$$

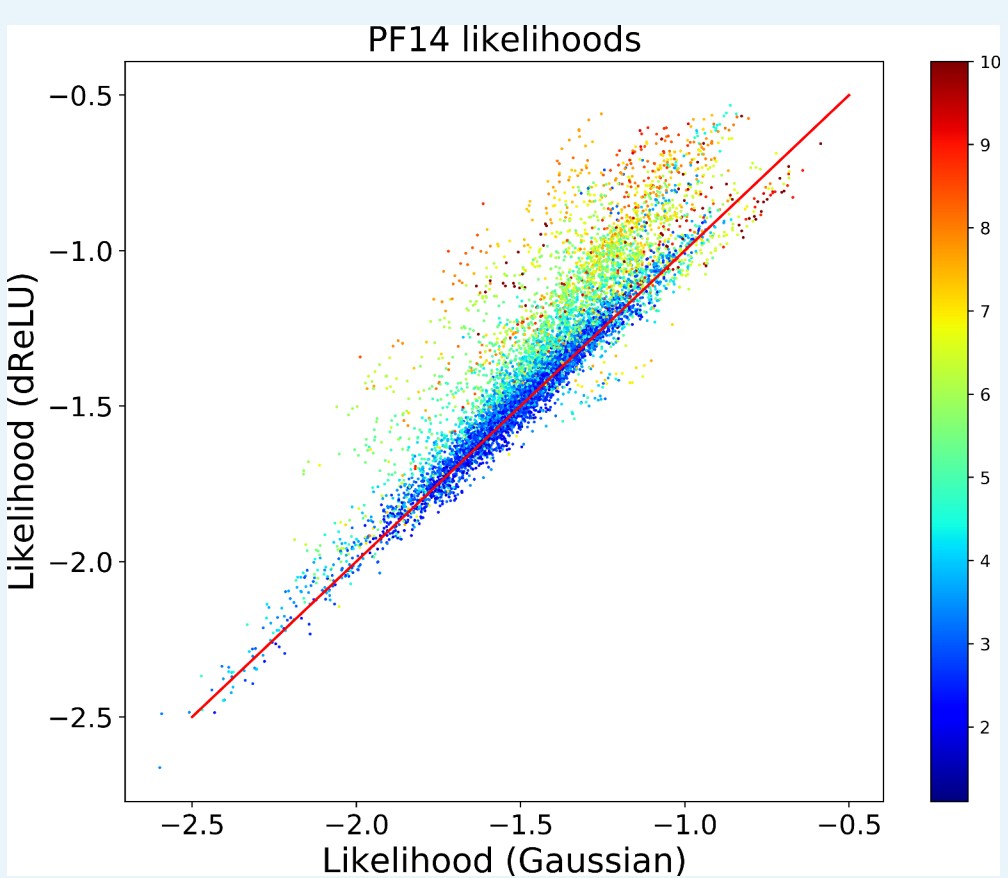

**Appendix 1—figure 5.** Comparison of Gaussian and dReLU RBM with $M = 100$ trained on the Kunitz domain MSA. Scatter plot of likelihoods for each model, where each point represents a sequence of the MSA. The color code is defined in *Equation 19*; hot colors indicate 'outlier' sequences.

DOI: https://doi.org/10.7554/eLife.39397.025

For instance, a sequence $\mathbf{v}^I$ with $c^I = 10$ has at least one hidden-unit average activity that is 10 standard deviations away from the mean. Clearly, most sequences have very similar likelihood but the outlier sequences are better modeled by dReLU potentials.

The features that are extracted are fairly robust with respect to the choice of potential when regularization is used. Clearly, the nature of the potentials does not matter for finding contacts features because for any potential, a hidden unit connected to only two sites will create only pairwise effective interaction. For larger collective modes, some difference arise. As discussed above, Bernoulli features are more redundant, and Gaussian RBM tend to miss outlier features.

## Summary

To summarize, the systematic study suggests that:

- More general potentials, such as dReLU, perform better than the simpler quadratic and Bernoulli potentials.

- There exist values of sparsity regularization penalties that allow for both good generative performance and interpretability.
- As the number of hidden units increases, more features are captured and generative performance improves. Beyond some point, increasing $M$ simply adds duplicate hidden units and does not enhance performance.

## Sequence generation

We use Lattice Proteins to check that our RBM is a good generative model, that is able to generate sequences that have both high fitness and high diversity (far away from one another and from the sequences provided in the training data set), as was done for Boltzmann Machines *Jacquin et al., 2016*). Various RBM are trained, sequences are generated for each RBM and scored using the ground truth $p_{nat}$ (see *Appendix 1—figure 6*). We find that: (i) RBMs with low likelihood (Bernoulli and/or small $M$) generate low-quality sequences; (ii) unregularized BMs and RBMs, which tend to overfit, generate sequences with higher fitness but low diversity; and (iii) the true fitness function is predicted well by the inferred log probability. Moreover, conditional sampling also generates high-quality sequences, even when conditioning on unseen combination of features.

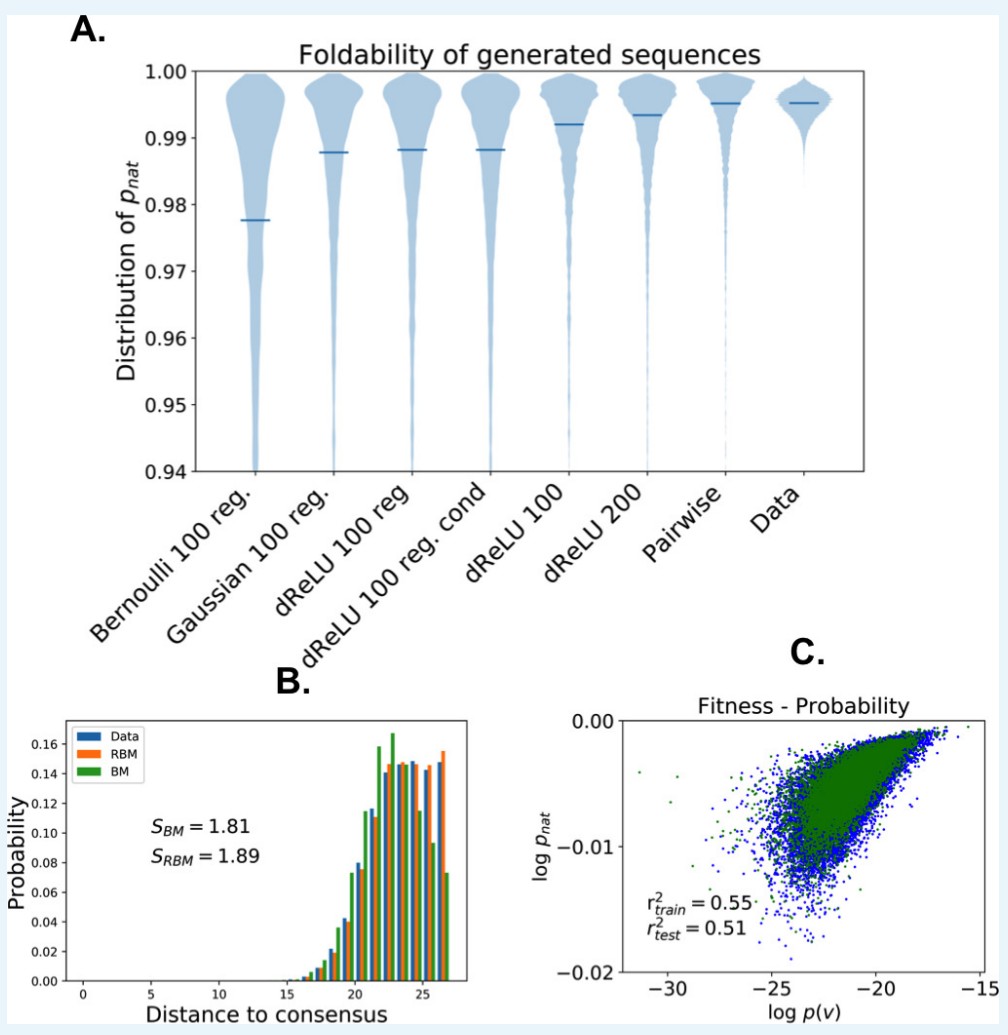

**Appendix 1—figure 6.** Quantitative quality assessment of sequences generated by RBM trained on the Lattice Protein MSA. (**a**) Distributions of the probability $p_{nat}$ of folding into the native structure $S_A$ (*Equation (14)* in 'Materials and methods'), for sequences generated by various models. The horizontal bars locate the average values of $p_{nat}$. Models with higher capacity

(more parameters, less regularization) generate sequences with higher quality but lower diversity. (**b**) Distribution of distances from a randomly selected wildtype. The unregularized BM samples have lower diversity, whereas the regularized RBM samples better reproduce the data distribution. (**c**) Log-probability of dReLU RBM $M = 100$ shown in the main text (**Figure 7**) vs true fitness evaluated on sequences from the MSA used (train) or not (test) for training.
DOI: https://doi.org/10.7554/eLife.39397.026

For RBMs trained on real proteins sequences, no ground-truth fitness is available and sequence quality cannot be assessed numerically. **Appendix 1—figure 7** shows nonetheless that the generated sequences, including those with recombined features that do not appear in nature, are consistent with a pairwise model trained on the same data.

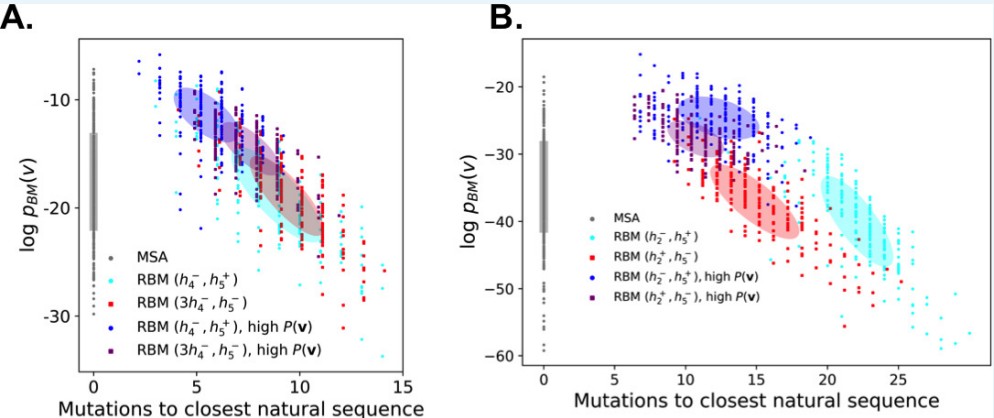

**Appendix 1—figure 7.** Quality assessment of sequences generated by RBM trained on (**a**) the Kunitz domain MSA and (**b**) the WW domain MSA. Scatter plot of the number of mutations to the closest natural sequence vs log-probability of a BM trained on the same data, for natural (gray) and RBM-generated (colored) WW domain sequences. The color code is that same as that used in **Figure 5A**. Note similar likelihoods values for RBM-generated sequences and natural ones, including the unseen $(h_4^-, h_5^+)$ combinations.
DOI: https://doi.org/10.7554/eLife.39397.027

Finally, in **Appendix 1—figure 8**, we show the role of regularization and sequence reweighting on sequence generation. Sequences drawn from the unregularized model are closer to those of the training data, and the corresponding sequence distribution has significantly lower entropy $S = -\sum_{\mathbf{v}} P(\mathbf{v}) \log P(\mathbf{v})$ (i.e. the average negative log-probability of the generated sequences). There are respectively about $e^S \sim 10^{12}$ and $10^{18}$ distinct sequences for the unregularized and regularized models, respectively. We find that sequence reweighting plays a similar role as regularization: with reweighting, sequences are slightly further away from the training set and the model has higher entropy.

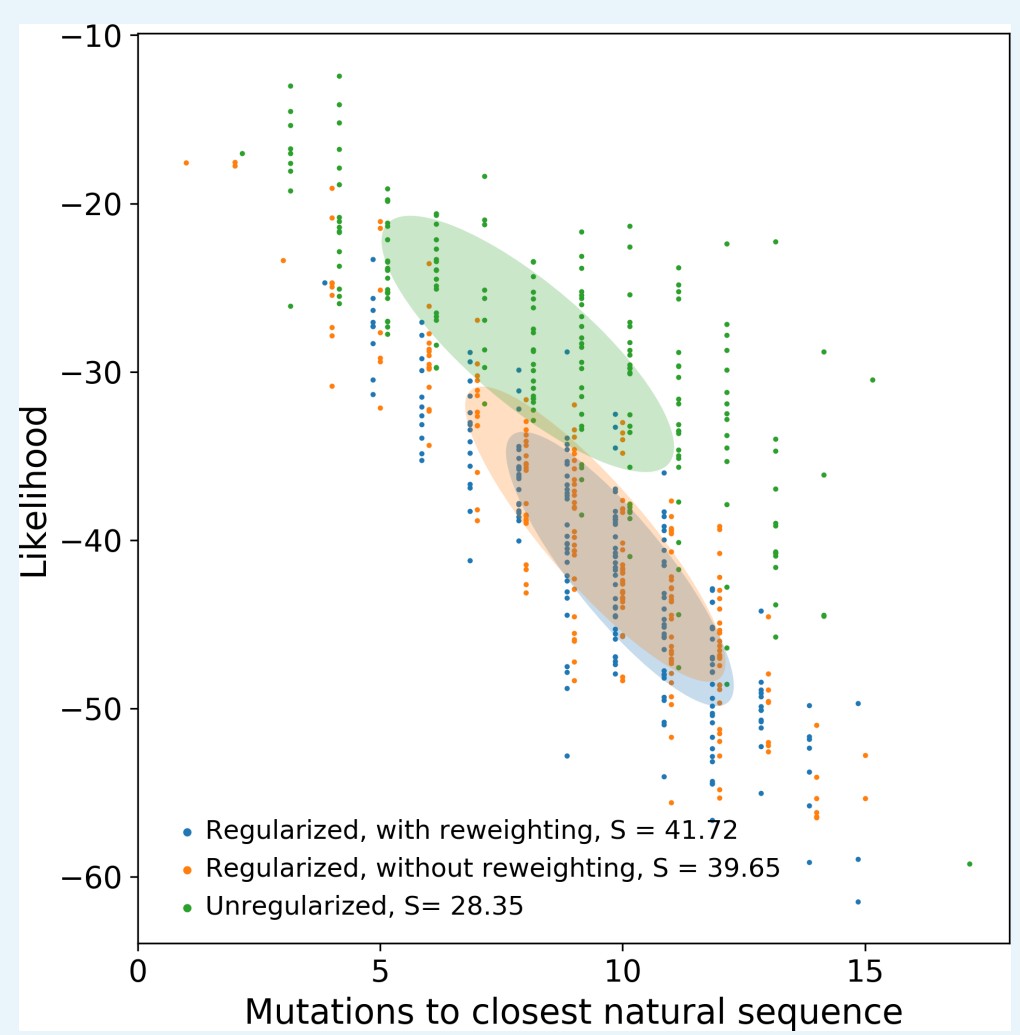

**Appendix 1—figure 8.** Evaluating the role of regularization and sequence reweighting on generated sequence diversity for the WW domain. The y-axis indicates the log-likelihood of the data generated by the model; entropy is the negative average log-likelihood.

DOI: https://doi.org/10.7554/eLife.39397.028

## Contact predictions

Since RBMs learn a full energy landscape, they can predict epistatic interactions (see 'Materials and methods'), and therefore contacts, as shown in *Figure 6*. The effective couplings derived with RBM are consistent with those inferred from a pairwise model (see *Appendix 1—figure 9*). Predictions for distant contacts in the Kunitz domain are shown in *Appendix 1—figure 10*, and are slightly worse than with DCA.

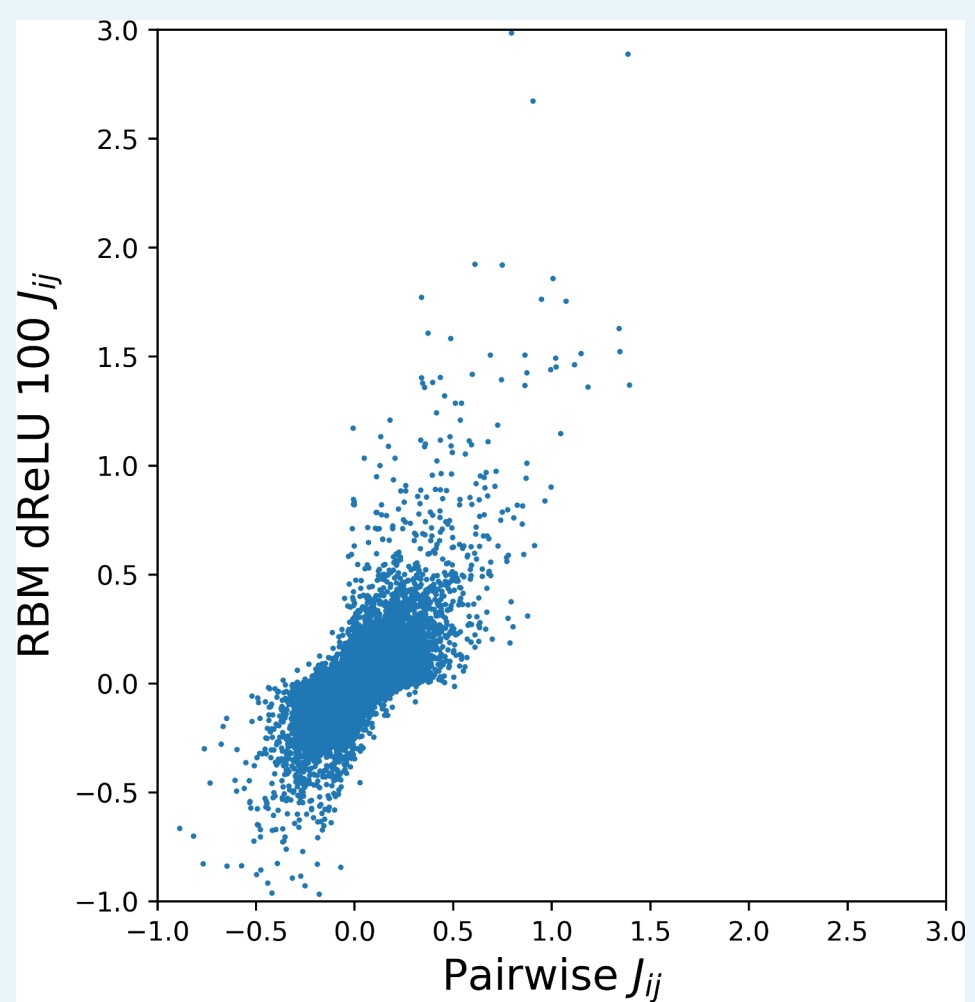

**Appendix 1—figure 9.** Pairwise couplings learned from Kunitz domain MSA. Scatter plot of inferred pairwise direct couplings learned by BM vs effective pairwise couplings computed from the RBM through **Equation (15)** in the 'Materials and methods'.

DOI: https://doi.org/10.7554/eLife.39397.029

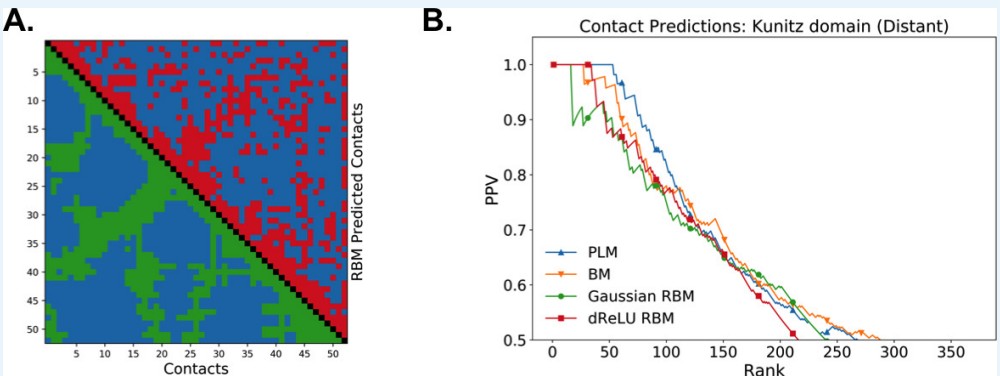

**Appendix 1—figure 10.** Contact map and contact predictions for the Kunitz domain. (**a**) Lower diagonal: the 551 pairs of residues at $D < 0.8$ nm in the structure. Upper diagonal: top 551 contacts predicted by dReLU RBM with $M = 100$, shown in **Figure 2**. (**b**) Positive Predicted Value vs rank for distant contacts $|i - j| > 4$ for RBM ($M = 100$) and pairwise models. Distant contacts are well predicted, including those involved in the secondary structure.

DOI: https://doi.org/10.7554/eLife.39397.030

We briefly discuss the best set of parameters for contact prediction. As seen from *Appendix 1—figure 11*, all RBMs can predict contacts maps on Lattice Proteins more or less accurately. As for the likelihood and generative performance, increasing the number of hidden units significantly improves contact prediction. The best hidden unit potentials for predicting contacts are dReLU and quadratic.

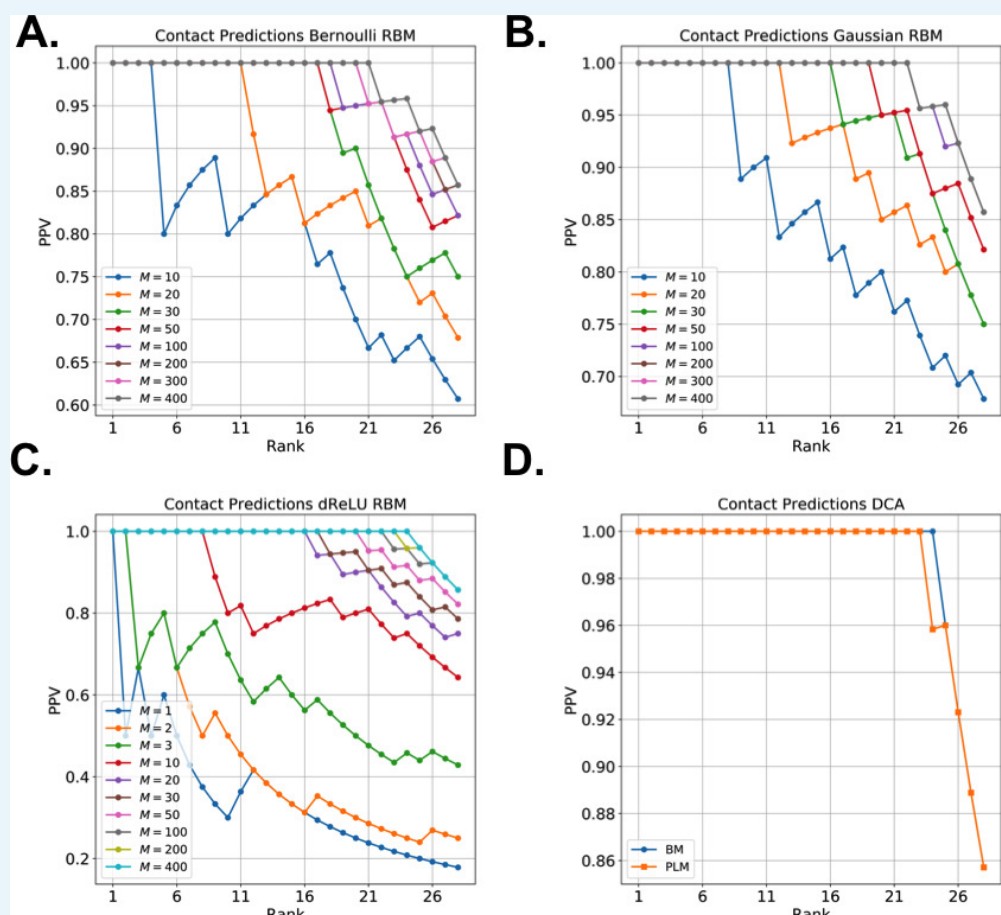

**Appendix 1—figure 11.** Contact predictions for Lattice Proteins, with (**a**) Bernoulli (**b**) Gaussian (**c**) dReLU RBM and (**d**) BM potentials. Models with quadratic or dReLU potentials and large number of hidden units are typically similar in performance to pairwise models, trained either with Monte Carlo or Pseudo-likelihood Maximization.

DOI: https://doi.org/10.7554/eLife.39397.031

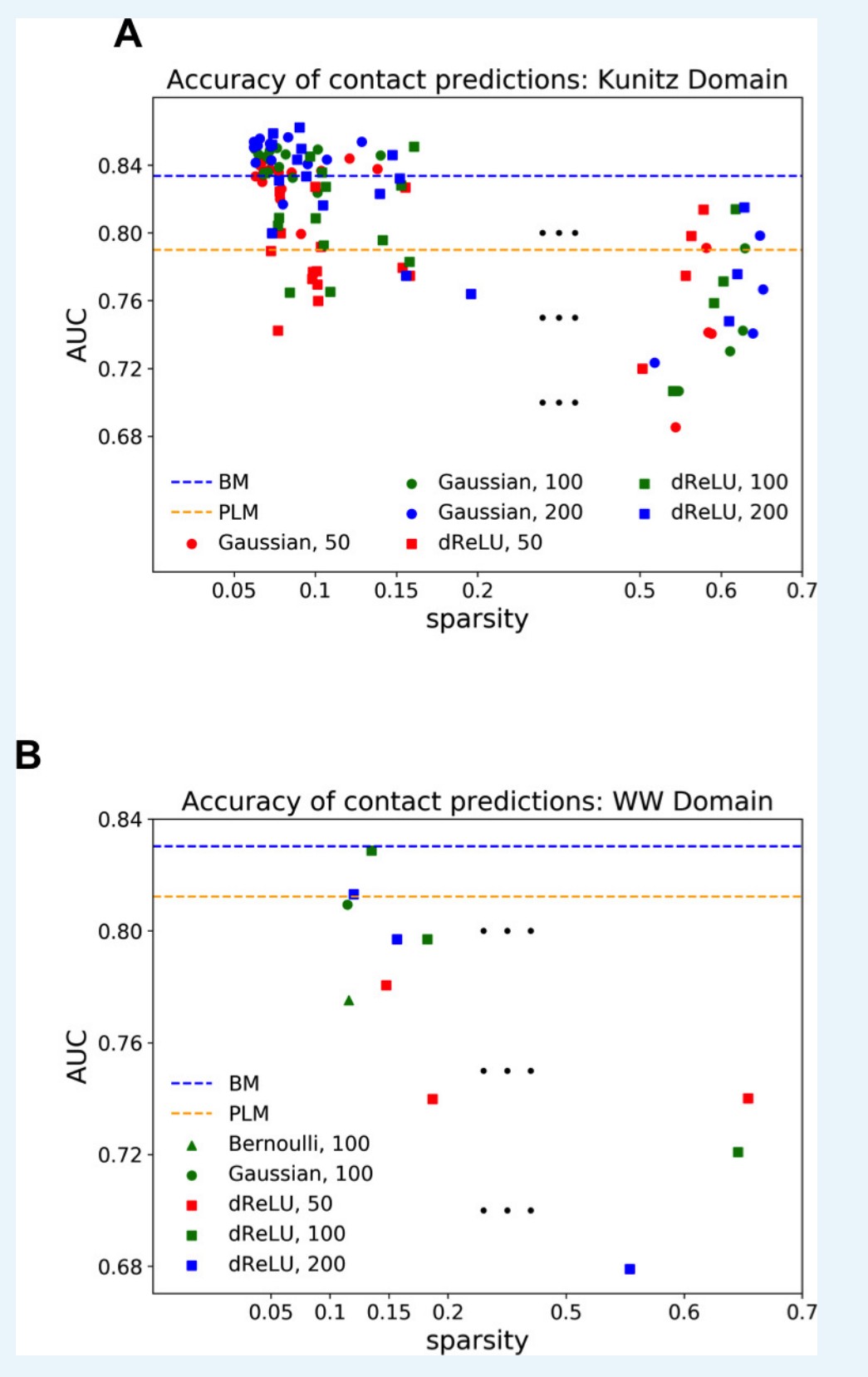

**Appendix 1—figure 12.** Contact predictions as a function of RBM parameters for (**a**) Kunitz and (**b**) WW domains. Both panels show the area under curve metric (integrated up to the true number of contacts) for various trainings, with different training parameters, regularization choice and hidden units number/potentials, against the weight sparsity. In both cases, large

sparse regularization and a high number of hidden units reproduce the performance of the pairwise models.

DOI: https://doi.org/10.7554/eLife.39397.032

We also studied how constraints on the sparsity of weights, tuned by the regularization penalty $\lambda_1^2$, influenced the performance. Because weights are never exactly zero, proxies are required for an appropriate definition of sparsity. In order to avoid arbitrary thresholds, we use Participation Ratios. The Participation Ratio ($PR_e$) of a vector $\mathbf{x} = \{x_i\}$ is

$$PR_e(\mathbf{x}) = \frac{\left(\sum_i |\mathbf{x_i}|^e\right)^2}{\sum_i |\mathbf{x_i}|^{2e}} \tag{20}$$

If $\mathbf{x}$ has $K$ nonzero and equal (in modulus) components, PR is equal to $K$ for any $e$. In practice, we use the values $e = 2$ and $3$: the higher $e$ is, the more small components are discounted against strong components in $\mathbf{x}$. Also note that it is invariant under rescaling of $\mathbf{x}$. We then define the weight sparsity $p_\mu$ of a hidden unit, through

$$p_\mu = \frac{1}{N}PR_3(\mathbf{x}_\mu) \quad \text{with} \quad (\mathbf{x}_\mu)_\mathbf{i} \equiv \sqrt{\sum_\mathbf{v} \mathbf{w_{i\mu}(v)}^2}$$

$$\tag{21}$$

and average it over $\mu$ to get a unique estimator of weight sparsity across the RBM. The results are reported in *Appendix 1—figure 12*, and show that performance strongly worsens when sparsity increases, both in Lattice Proteins and in real families.

## Feature robustness

To assess feature robustness, we repeat the training on WW using only one of the two halves of the sequences data, and look for the closest features to those shown in the main text. The closest features, shown below, are quite similar to the original ones.

## Comparison with the Hopfield-Potts model

The Hopfield-Potts model is a special case of RBM with: (i) quadratic potentials for hidden units, ii) no regularization but orthogonality constraints on the weights, and (iii) mean-field inference rather than PCD Monte Carlo learning. The consequences are that: (i) we cannot model high-order interactions, (ii) we do not observe a compositional regime in which the weights are sparse and typical configurations are obtained by combinations of these weights, instead, the representation is entangled and the weights attached to high eigenvalues are extended over most sites of the protein; and (iii) the model is not generative, that is, it does not reproduce the data moments and cannot generate a diverse set of sequences. To illustrate this fact, we show:

- Examples of weights inferred from the the Kunitz and WW domains, and for Lattice Proteins (weights corresponding to Hsp70 can be found in a 'Supporting information' file). Low-eigenvalue weights are sparse, as reported in *Cocco et al. (2013)*, but high eigenvalue weights that encode collective modes are extended, and therefore hard to interpret and to relate to function.
- Contact predictions with Hopfield-Potts, showing worse performance than RBM or plmDCA.
- Benchmarking of generated sequences with Hopfield-Potts on Lattice Proteins (similar to *Figure 7F*). Using a small pseudo-count, sequences are very poor (have a very low folding probability). Using a larger pseudo-count, sequences have reasonable fitness $p_{\text{nat}}$, although lower than those for high-$P(\mathbf{v})$ RBM, but quite low diversity. This phenomenon is characteristic of sequences generated with mean-field models (see figure 3A in *Jacquin et al. (2016)*. We also note that the Lattice Protein benchmark is actually optimistic for the Hopfield-Potts model, as the pseudo-count trick does not work as well whenever a sequence has many conserved sites.

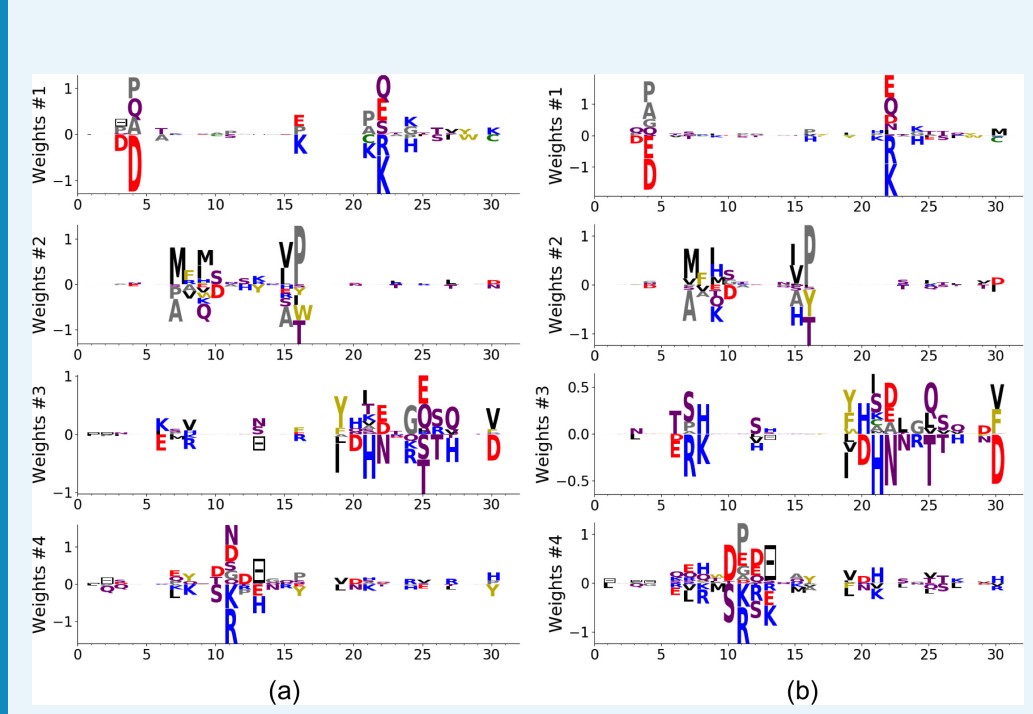

**Appendix 1—figure 13.** Features inferred using the first and second half of the sequences.

DOI: https://doi.org/10.7554/eLife.39397.033

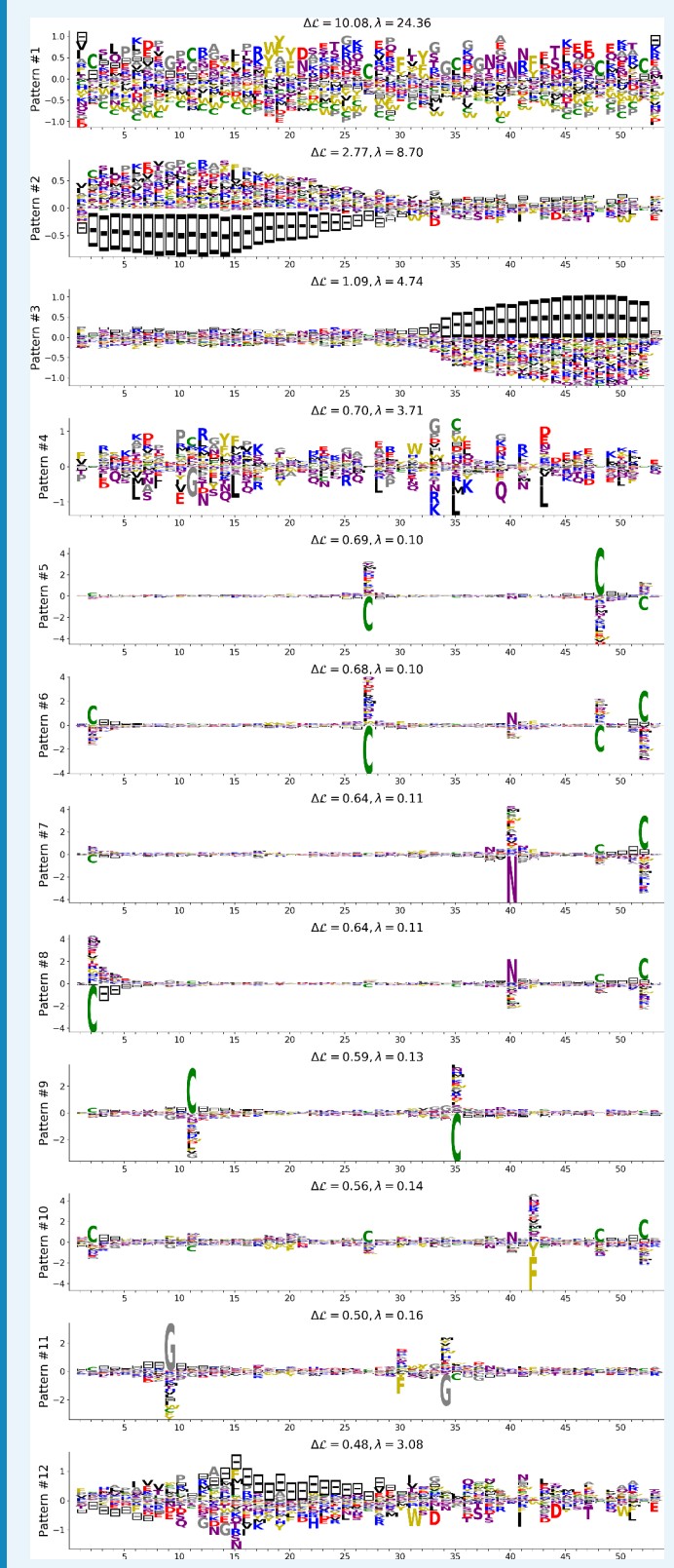

**Appendix 1—figure 14.** Top 12 patterns with highest contributions to the log-probability, see eqn (23) in *Cocco et al. (2013)*, inferred by the Hopfield-Potts model on the Kunitz domain.

DOI: https://doi.org/10.7554/eLife.39397.034

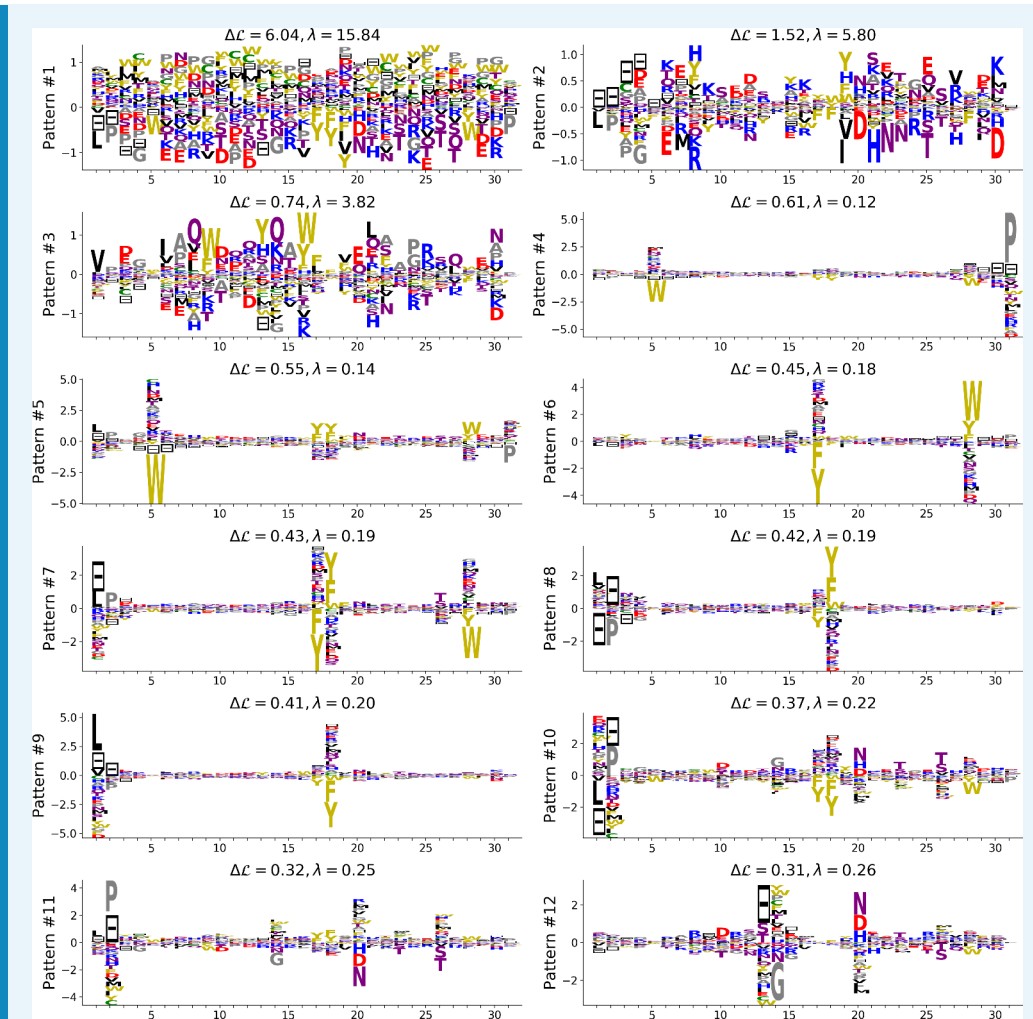

**Appendix 1—figure 15.** Top 12 patterns with the highest contributions to the log-probability (see equation (23) in *Cocco et al. (2013)*), inferred by the Hopfield-Potts model on the WW domain.

DOI: https://doi.org/10.7554/eLife.39397.035

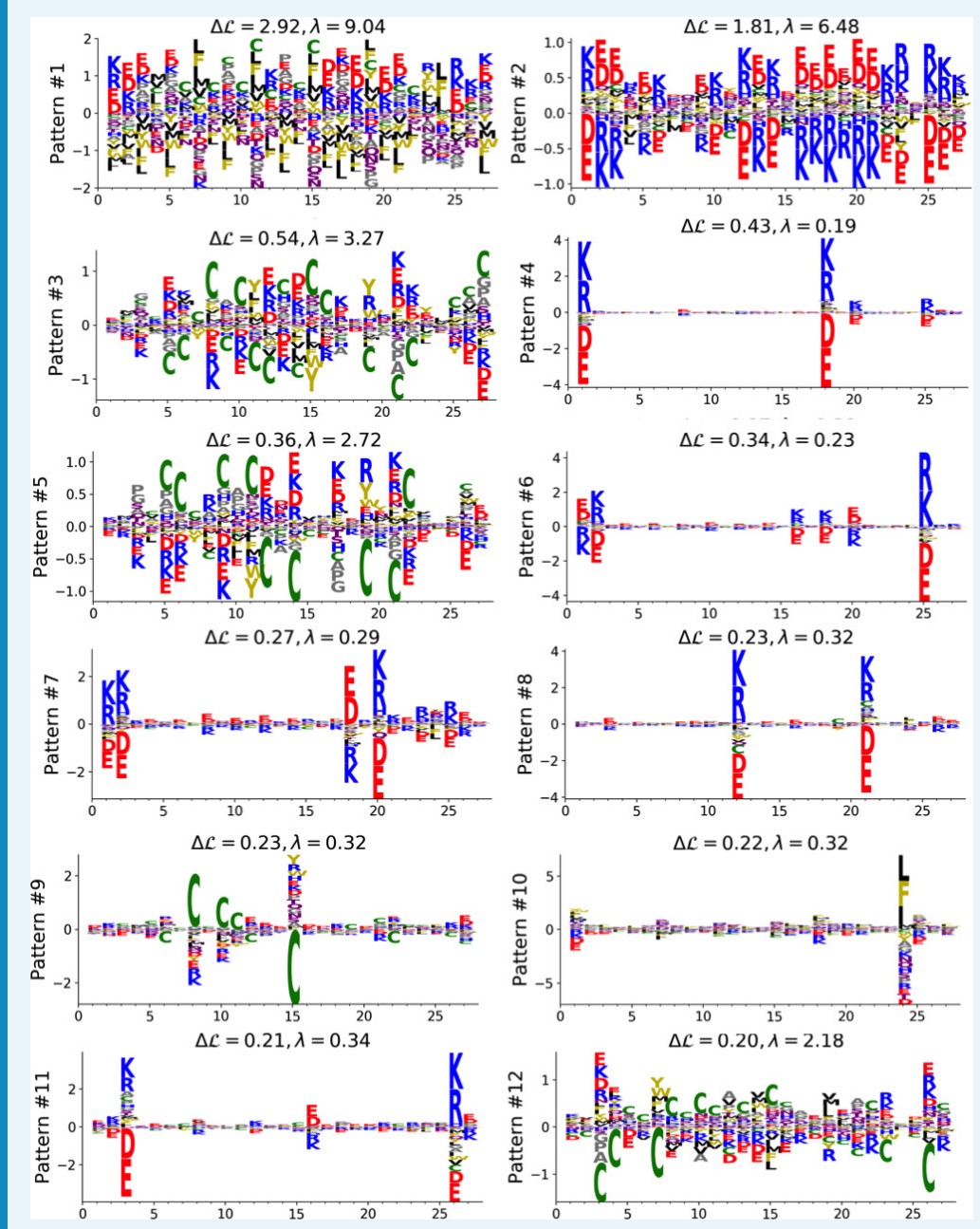

**Appendix 1—figure 16.** Top 12 patterns with the highest contributions to the log-probability (see equation (23) in *Cocco et al. (2013)*, inferred by the Hopfield-Potts model on the Lattice Proteins data.

DOI: https://doi.org/10.7554/eLife.39397.036

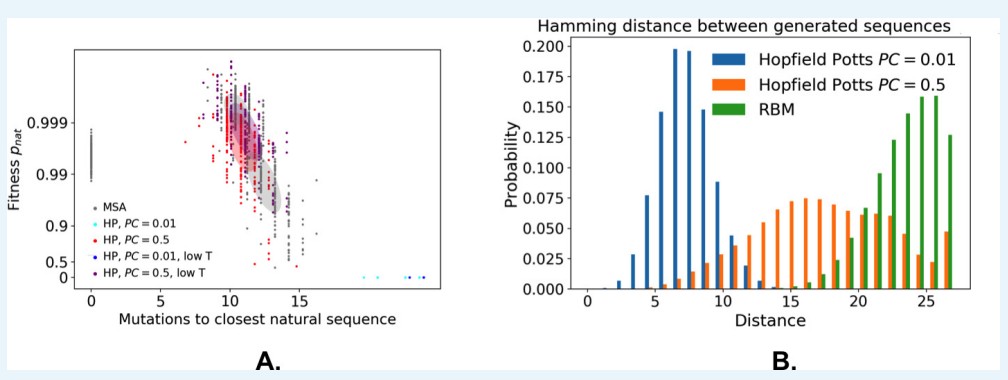

**Appendix 1—figure 17.** Hopfield-Potts model for sequence generation. (**A**) Fitness $p_{nat}$ against distance to closest sequence for the Hopfield-Potts model with pseudo-count 0.01 or 0.5, sampled with or without the high $P(\mathbf{v})$ bias. Gray ellipses denote the corresponding values for the RBM. (**B**) Distribution of distances between generated sequences.

DOI: https://doi.org/10.7554/eLife.39397.037

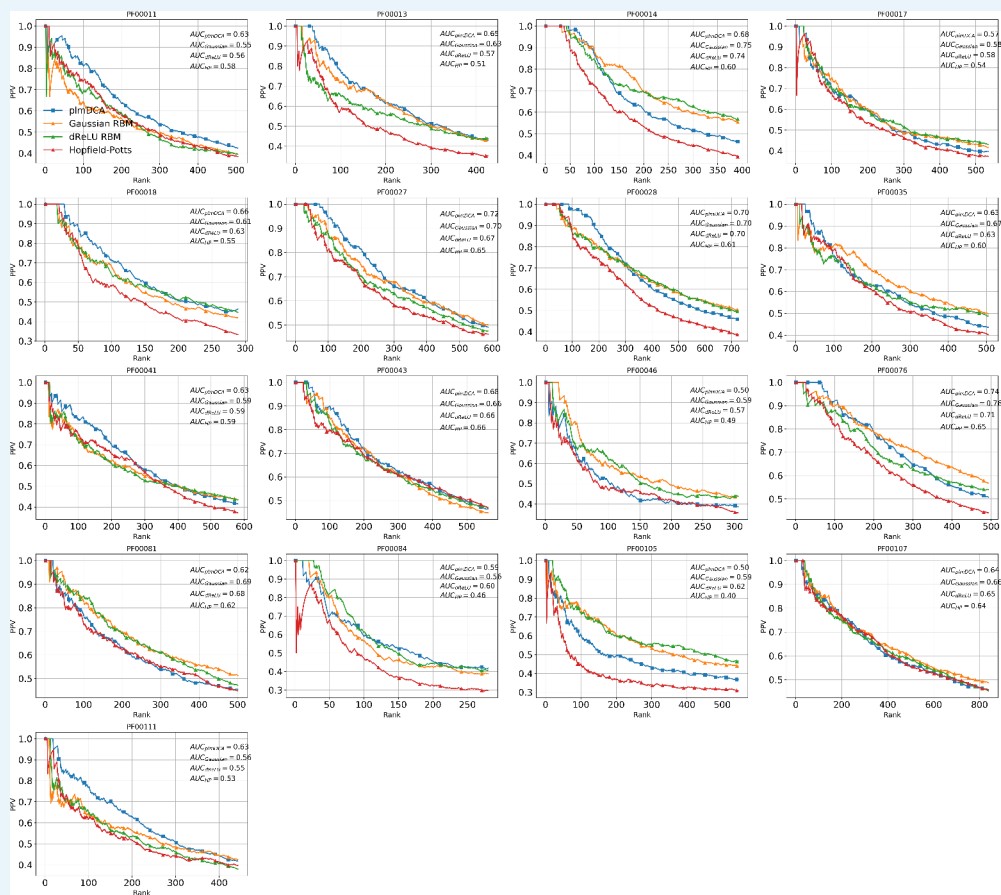

**Appendix 1—figure 18.** Contact prediction for 17 protein families including the Hopfield-Potts model.

DOI: https://doi.org/10.7554/eLife.39397.038

## Additional figure: hidden-input distribution for the Kunitz domain, separated by phylogenetic identity and genes

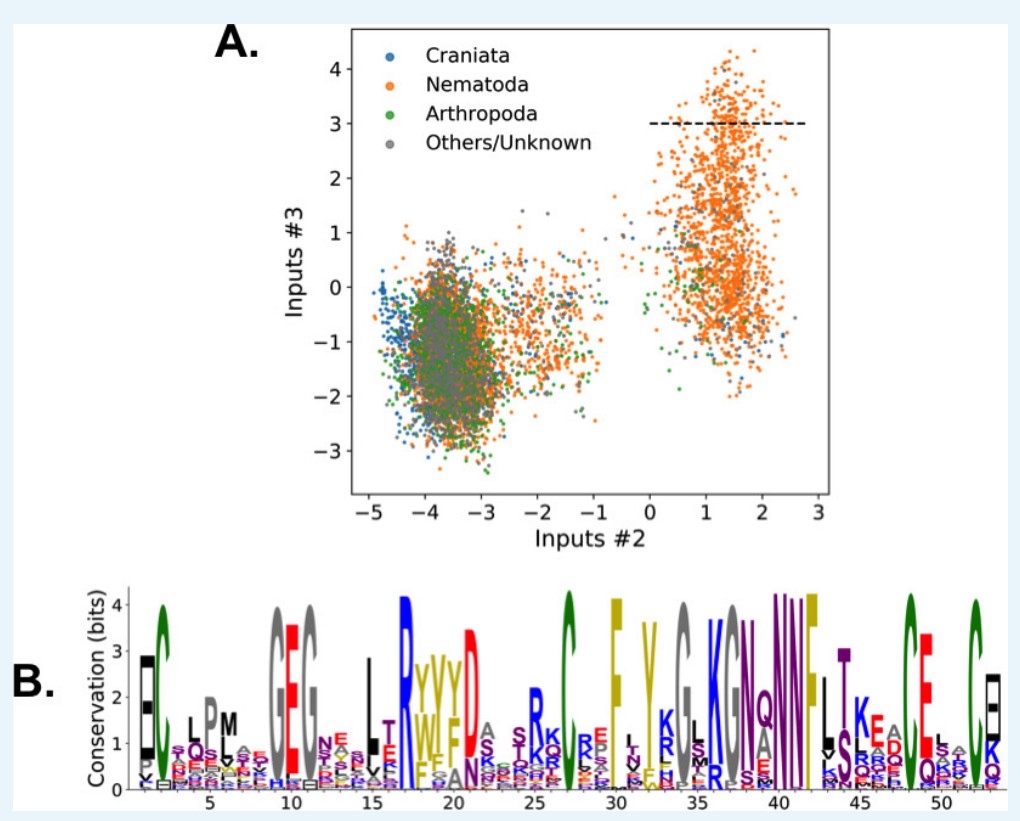

**Appendix 1—figure 19.** Phylogenetic identity of feature-activating Kunitz sequences with the RBM shown in *Figure 2*. (**A**) Scatter plot of inputs of hidden units 2 and 3; color depicts the organisms' position in the phylogenic tree of species. Most of the sequences that lack the disulfide bridge are nematodes. (**B**) Sequence logo of the 137 sequences above the dashed line ($I_3 > 3$), showing the electrostatic triangle that putatively replaces the disulfide bridge.

DOI: https://doi.org/10.7554/eLife.39397.039

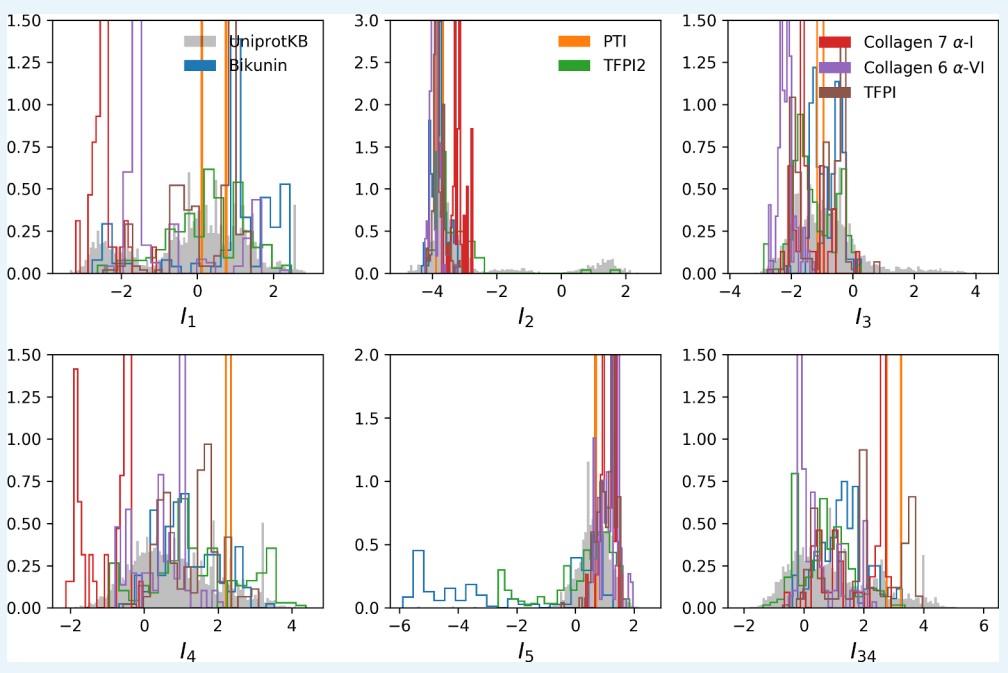

**Appendix 1—figure 20.** Distribution of inputs for the five features shown in main text plus

hidden unit 34. Distributions of inputs for Kunitz domains belonging to specific genes are shown.

DOI: https://doi.org/10.7554/eLife.39397.040

## Additional figure: weight logos, 3D visualizations, input distributions of 10 hidden units for Hsp70

Hidden unit numbering: 1 = short vs long loop; 2 = function feature on SBD; 3 = LID/SBD interdomain; 4 = NBD/SBD interdomain and non-allosteric specific; 5 = unstructured tail; 6 = short/long vs very short loop; 7 = long loop variant; 8 = ER specific; 9 = second non-allosteric specific; 10 = dimer contacts.

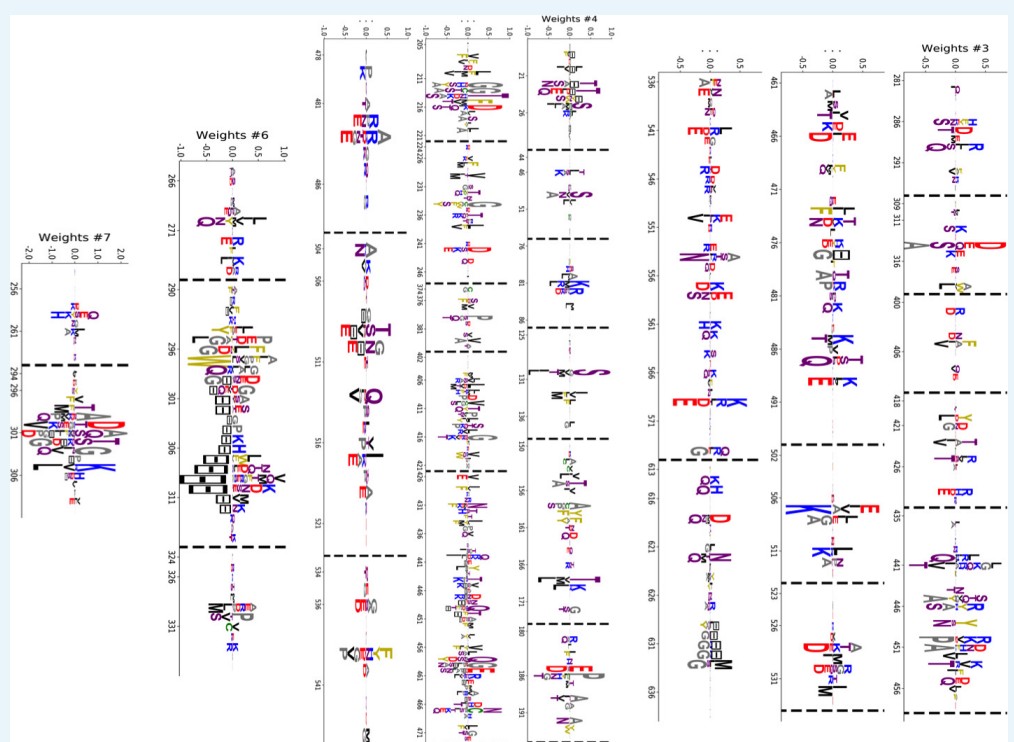

**Appendix 1—figure 21.** Truncated weight logo of 10 selected HSP70 hidden units (1/2).

DOI: https://doi.org/10.7554/eLife.39397.041

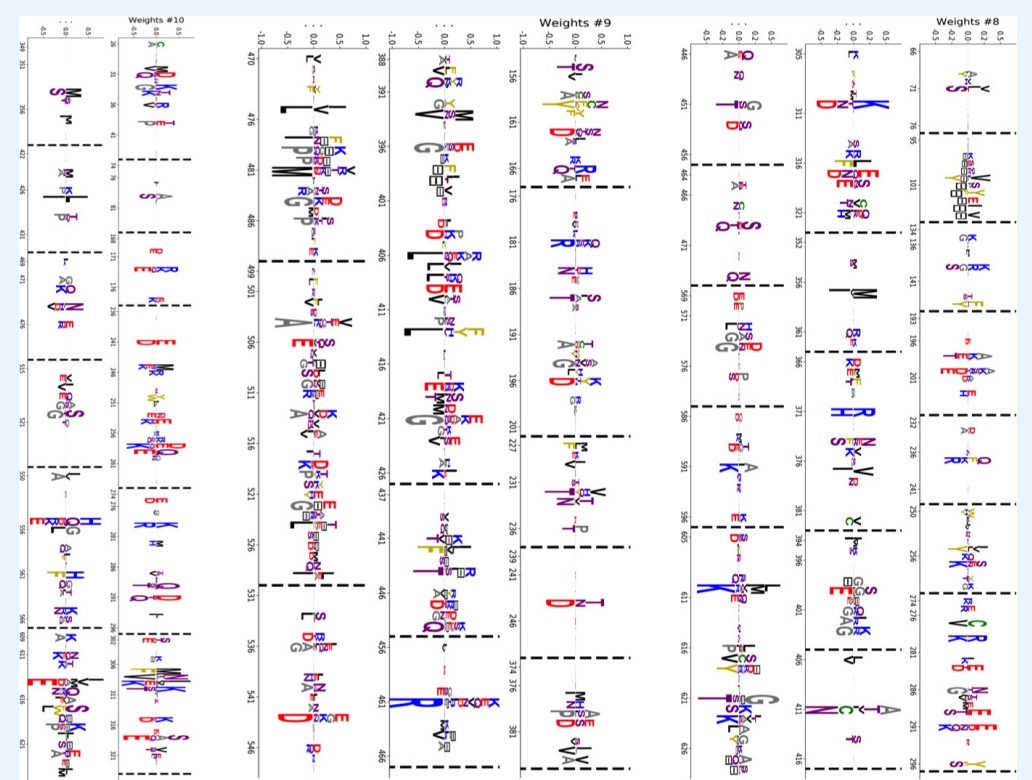

**Appendix 1—figure 22.** Truncated weight logo of 10 selected HSP70 hidden units (2/2).

DOI: https://doi.org/10.7554/eLife.39397.042

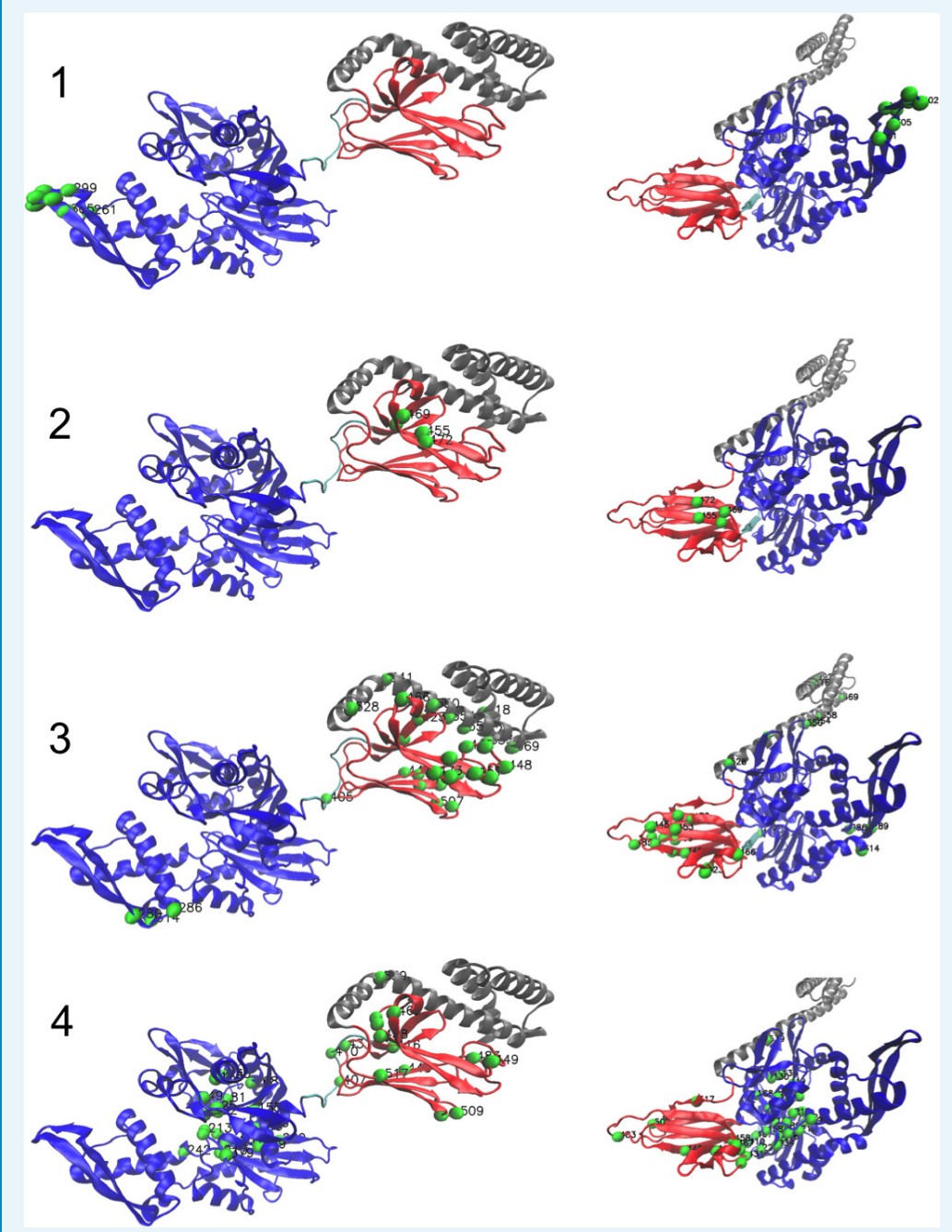

**Appendix 1—figure 23.** Corresponding structures (1/3). Left: ADP-bound conformation (PDB: 2kho). Right: ATP-bound conformation (PDB: 4jne). For the last hidden unit, we show the structure of the dimer Hsp70–Hsp70 in ATP conformation (PDB: 4JNE), highlighting dimeric contacts.

DOI: https://doi.org/10.7554/eLife.39397.043

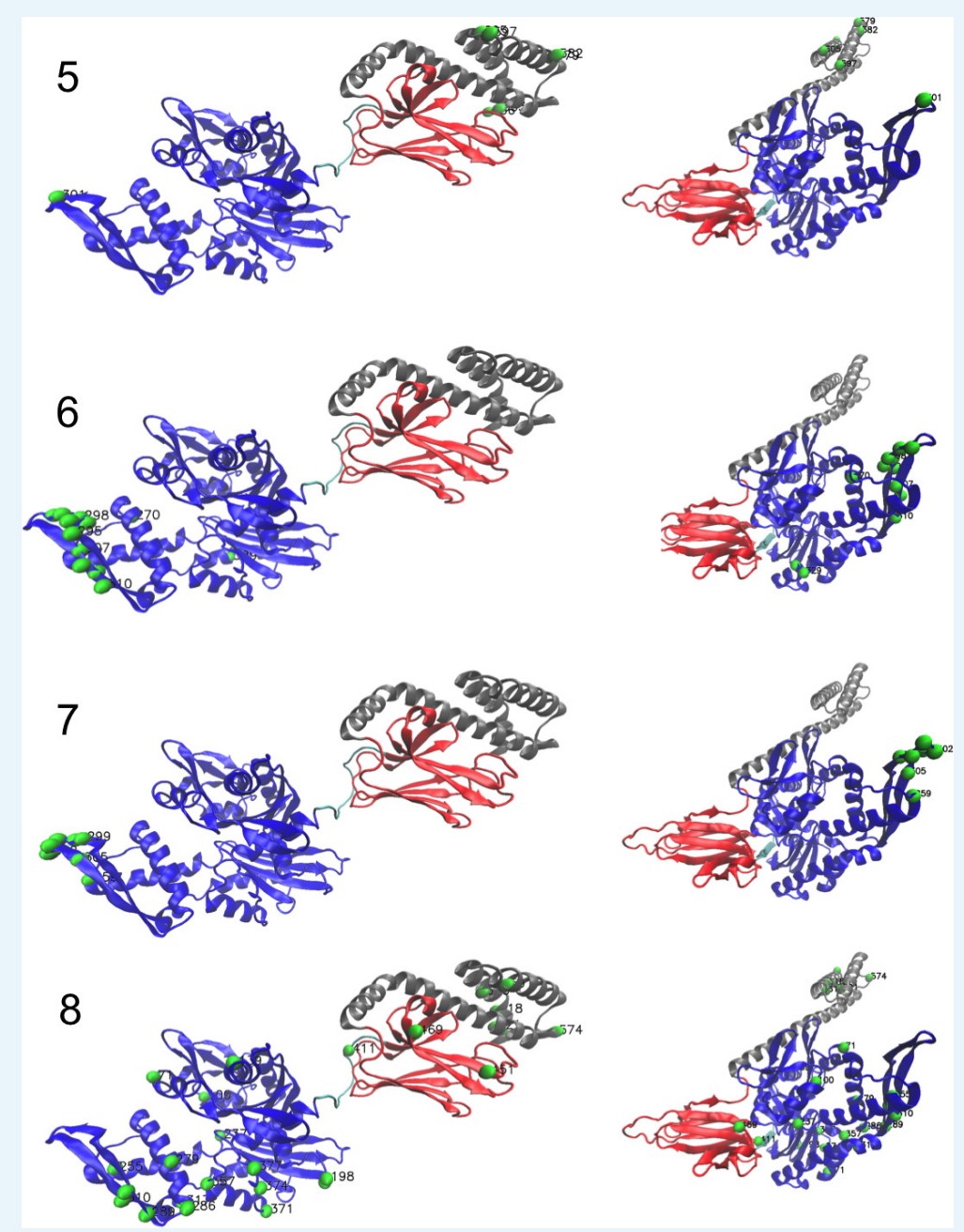

**Appendix 1—figure 24.** Corresponding structures (2/3). Left: ADP-bound conformation (PDB: 2kho). Right: ATP-bound conformation (PDB: 4jne). For the last hidden unit, we show the structure of the dimer Hsp70–Hsp70 in ATP conformation (PDB: 4JNE), highlighting dimeric contacts.

DOI: https://doi.org/10.7554/eLife.39397.044

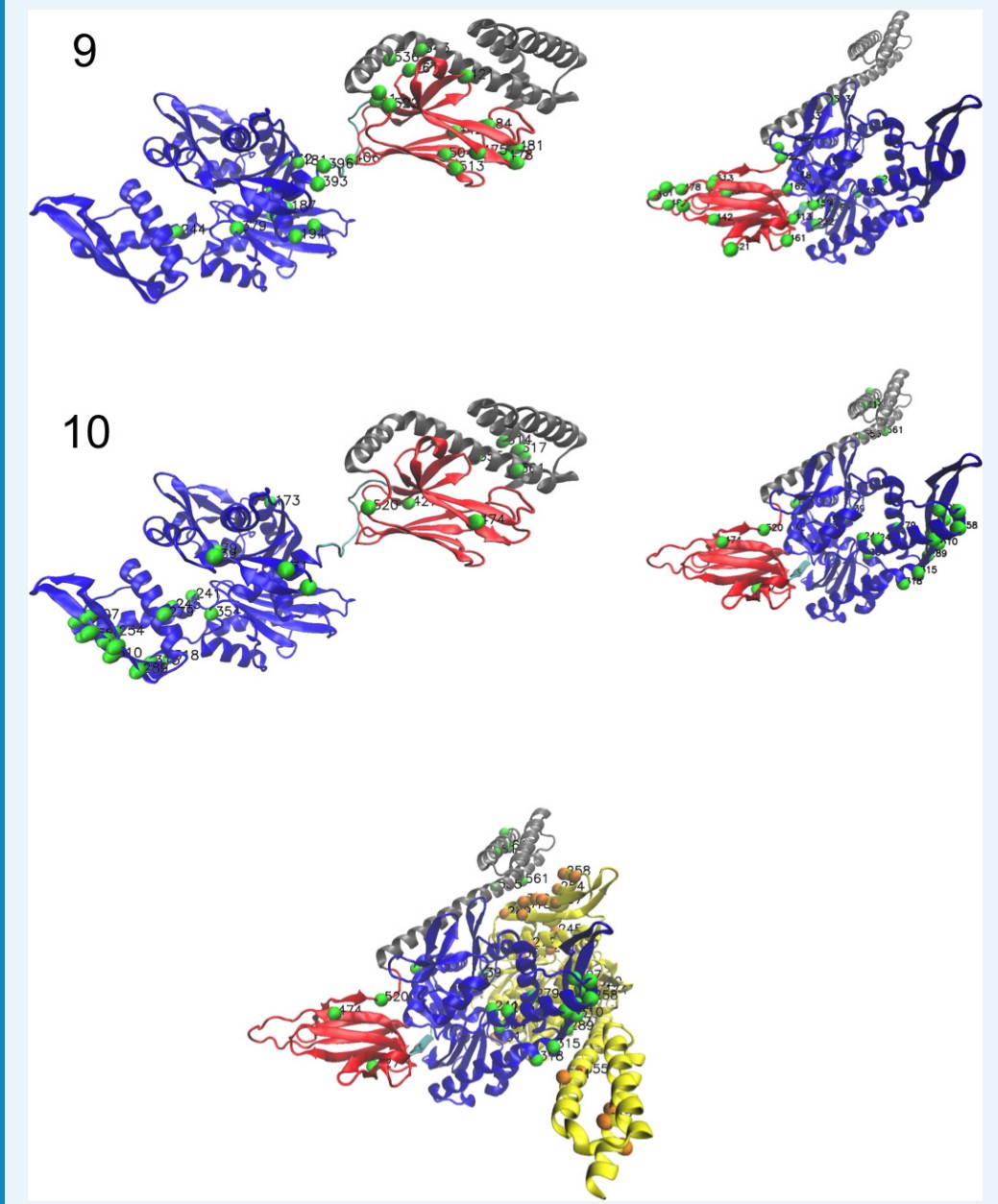

**Appendix 1—figure 25.** Corresponding structures (3/3). Left: ADP-bound conformation (PDB: 2kho). Right: ATP-bound conformation (PDB: 4jne). For the last hidden unit, we show the structure of the dimer Hsp70–Hsp70 in ATP conformation (PDB: 4JNE), highlighting dimeric contacts.

DOI: https://doi.org/10.7554/eLife.39397.045

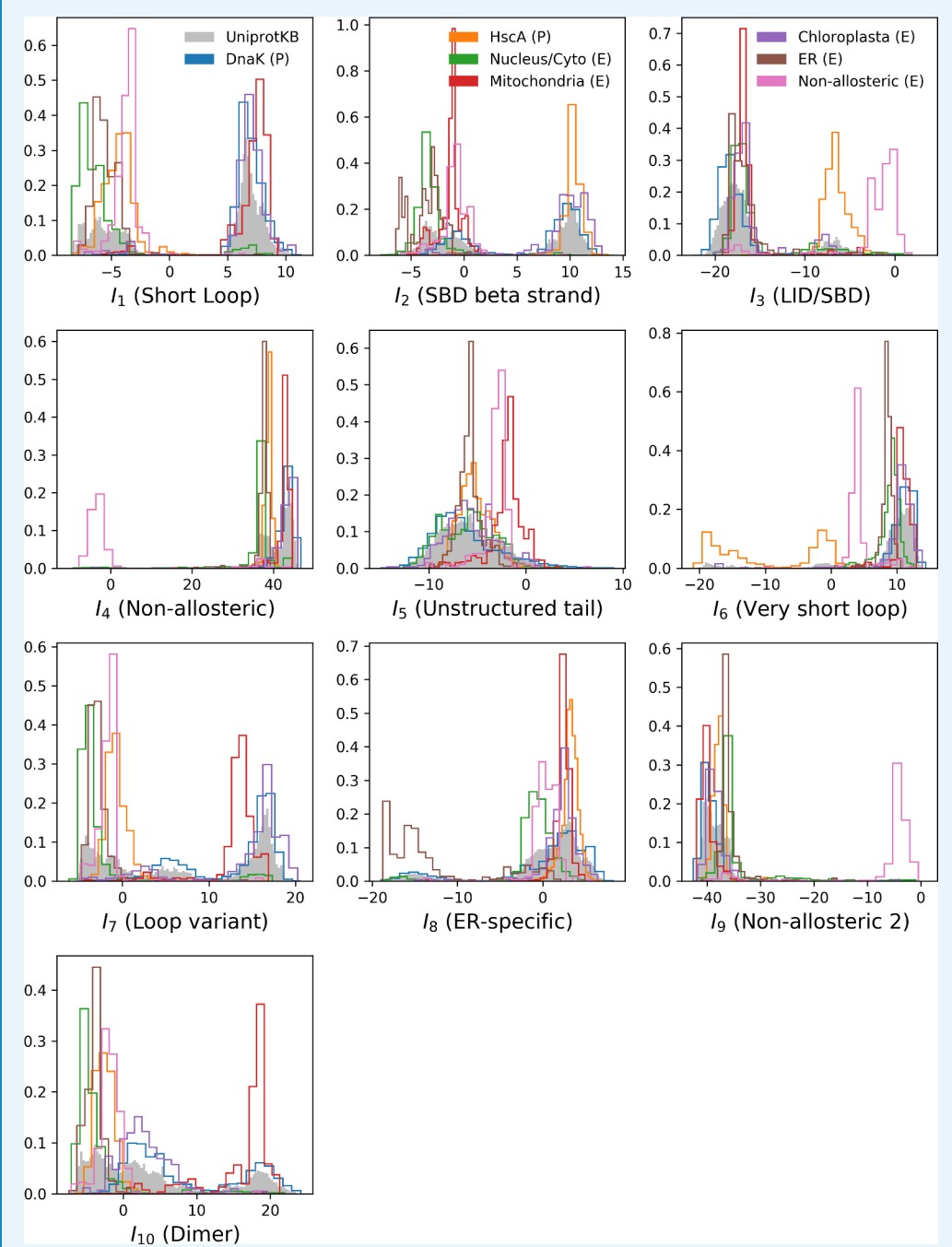

**Appendix 1—figure 26.** Corresponding input distributions. Note that both hidden unit 4 and 9 discriminate the non-allosteric subfamily from the rest; and that hidden unit 8 discriminates eukaryotic Hsp expressed in the endoplasmic reticulum from the rest.

DOI: https://doi.org/10.7554/eLife.39397.046

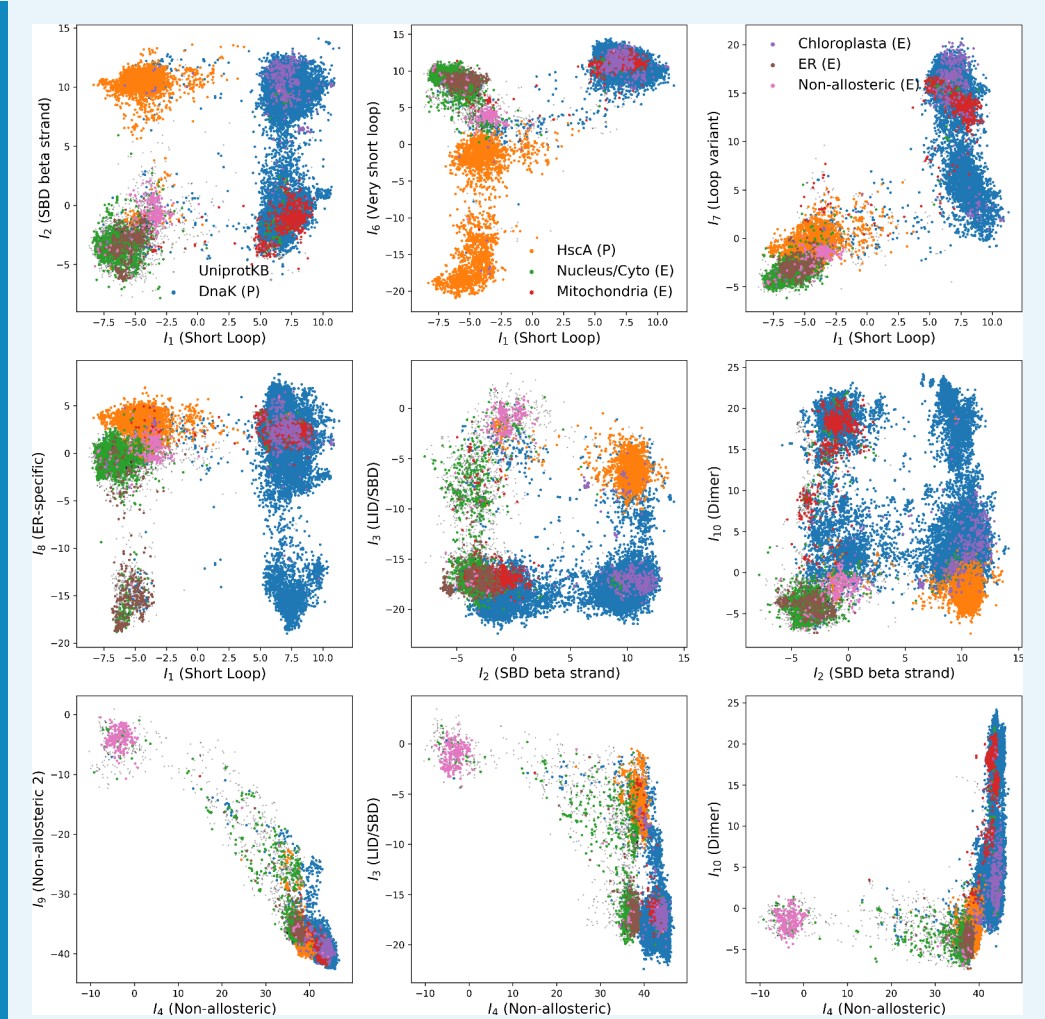

**Appendix 1—figure 27.** Some scatter plots of inputs for the 10 hidden units shown.

DOI: https://doi.org/10.7554/eLife.39397.047

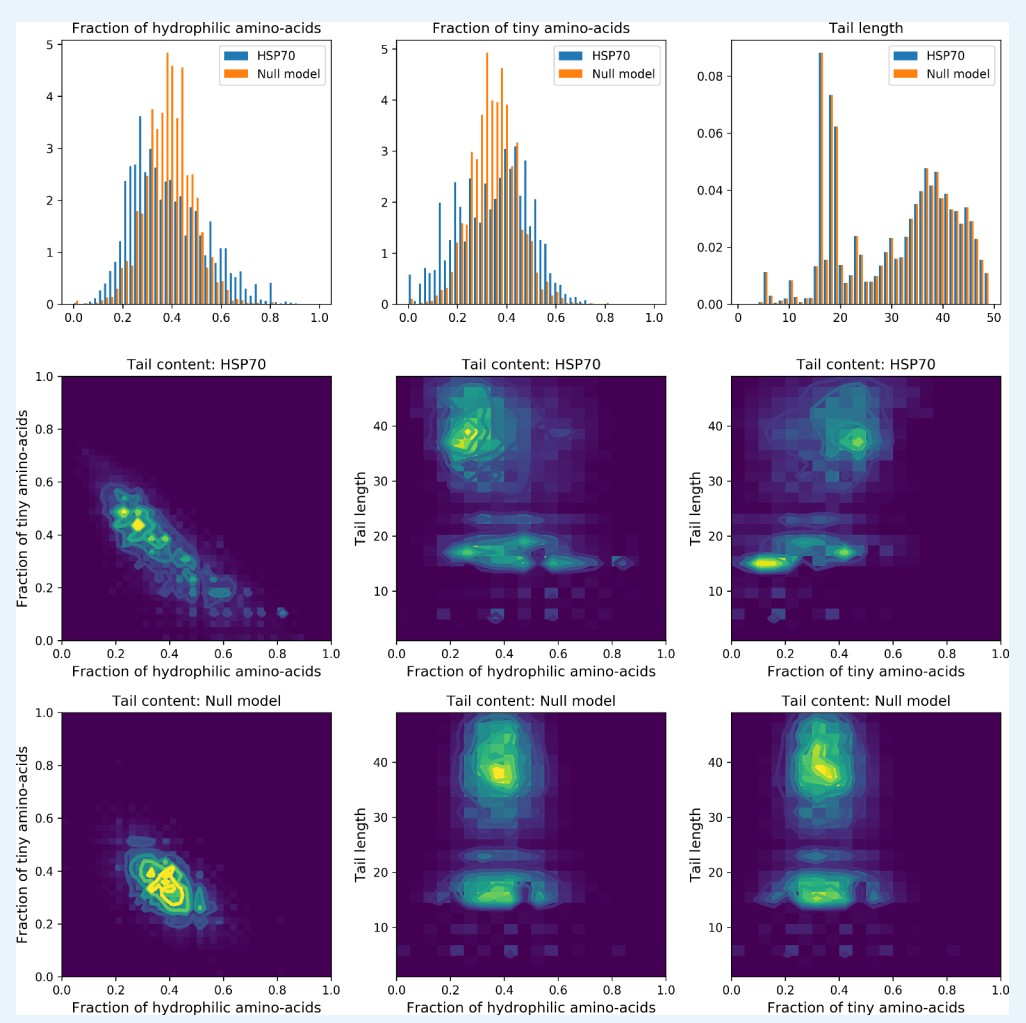

**Appendix 1—figure 28.** Statistics of the length and amino-acid content of the unstructured tail of Hsp70. Hidden unit 5 defines a set of sites, mostly located on the unstructured tail of Hsp70; its sequence logo and input distribution suggests that for a given sequence, the tail can be enriched either in tiny (A, G)or hydrophilic amino-acids (E,D,K,R,T,S,N,Q). This is qualitatively confirmed by the non-gaussian statistics of the distributions of the fractions of tiny and hydrophilic amino-acids in the tail (blue histograms and top left contour plots). This effect could, however, be due to the variable length of the loop (bottom histogram). To assess this enrichment, we built a null model where the tail size was random (same statistics as Hsp70), and each amino-acid was drawn randomly, independently from the others, using the same amino-acid frequency as that in the tail of Hsp70. The null model statistics (orange histograms and lower left contour plots) are clearly different, validating the collective mode.
DOI: https://doi.org/10.7554/eLife.39397.048

