## [Decision Letter]

Thank you for submitting your article "Learning protein constitutive motifs from sequence data" for consideration at *eLife*. Your article has been favorably evaluated by three reviewers, one of whom served as a guest Reviewing Editor, and the evaluation has been overseen by Aviv Regev as the Senior Editor.

The reviewers have discussed the reviews with one another and the Reviewing Editor has drafted this decision to help you prepare a revised submission.

1) There is a difference of opinion among the reviewers about whether your methodology needs to be compared with other protein structures, and in particular examples that are not so well known and annotated. However the consensus opinion is that the manuscript would benefit from additional comparisons. Reviewer 3 explicitly calls out the possibility that there is overfitting involved, and it is hard to dismiss this out of hand without more effort to compare to the types of examples that they mention. Reviewer 3 makes reasonable suggestions of other examples you might compare to – and I look forward to seeing how these work in the revised manuscript.

2) All of the reviewers have major comments for clarifications and addressing these will improve the manuscript. For example, reviewer 1 asks you to comment on the weights (how they are chosen, how they depend on model hyperparameters, etc.); Reviewer 2 points out that the methodology is not fully developed in this manuscript, but instead will be described elsewhere. The consensus view of the reviewers is that it is important for the methods to be fully described in this paper so that the results are reproducible by others, and a promise of publishing these elsewhere is insufficient.

3) There are other explicit comments included in these reviews about ways of clarifying the manuscript that I trust you will address in a revised version.

*Reviewer #1:*

In this manuscript Tubiana et al. show that Restricted Boltzman Machines are able to learn models of protein families from sequence data. The authors observe two contrasting approaches to building models from protein family sequence data that have been shown in the literature to either (i) identify functionally important groups of sequence positions, or ii) provide information about sequence positions that are close in 3D protein structure. Approach (i) often involves applying PCA type approaches to large sets of protein sequences, and examining the weights assigned to different sequence positions on those principal components with largest eigenvalues, while versions of approach (ii) such as DCA have been motivated by maximum-entropy based modelling.

The authors note that a unique framework that can extract both structurally and functionally important features of a protein family is still missing, and propose Restricted Boltzmann Machines as a solution. In the manuscript, they report a method to efficiently learn RBM from sequence data, and describe applications to three diverse protein families, in addition to data generated using a lattice-protein model. Much of the manuscript is devoted to describing the sets of sequence positions that are structurally and/or functionally important in these protein families, and that can be identified by examining the weight coefficients of the RBM.

Within the main text, the authors cite their previous publication, from 2013, in which they propose the Hopfield-Potts model as a means of naturally interpolating between PCA-based approaches to identifying functionally important groups of residues, and DCA-type approaches. There they also observed sparse modes (or patterns) that correspond to contacts in 3D structure, and in addition less sparse modes that correspond to features such as conserved sequence positions that form spatially connected and functionally important regions in the folded protein that are reminiscent of protein sectors, or positions at which many gap symbols are found in the alignments.

This makes me question whether there is really no framework that can extract both structurally and functionally important features of a protein family. I think the authors should be careful to accurately place this manuscript in the context of prior work. Moreover, there appear to be some similarities between the energy function of an RBM, and that of a Hopfield model – this would be an interesting point for the authors to discuss. It would be interesting to understand from the main text why the authors feel that the modeling assumptions of the RBM are applicable to protein sequence data.

My major comment is that the new manuscript focuses on a rather detailed description of how a small number of RBM weights correspond to structurally and functionally important sequence positions. Almost no time at all in the main text describing how they were able to efficiently learn RBM from sequence data, which seems to be the novel contribution of this work. The authors also do not provide any comparison between the weights of RBM and patterns learned from sequence data in prior work, such as their own 2013 paper. Specifically, did the patterns derived from the Hopfield-Potts model differ significantly from those found in the RBM weights? A thorough comparison to prior work is necessary to distinguish the advances made in this work.

Beyond this, I would ask the authors to at least comment on how the diverse weights are chosen, and what happens to the weights as the number of hidden units in the model is varied. Moreover, it would be interesting to understand how the weights change as the number, and diversity of sequences in the protein family alignment is varied. In addition, the authors pay only passing attention to the fact that the protein sequences contained in the alignment are chosen uniformly at random from some underlying distribution, but rather are biased by their shared evolutionary history. The authors should provide some insight into how this affects the ability of the model to generalize.

*Reviewer #2:*

This manuscript introduces restricted Boltzmann machines (RBMs) for protein sequence analysis. There has been a recent revolution in inferring 3D protein structure by statistical analysis of pairwise residue covariations in deep protein sequence alignments, mostly using a class of models called Potts models; these authors have helped pioneer that work. Here the authors show that RBMs can efficiently capture effective pairwise couplings (as Potts models do), while also capturing even higher-order correlation patterns that arise as a result of phylogenetic and functional constraint. The math is well-principled and presented clearly and concisely. The paper shows a well-chosen set of illustrative biological examples that connect this work to the body of related previous work with Potts models and other methods.

This is a mathematical manuscript that introduces a compelling new analysis approach, but there isn't (yet) a new biological result. Maybe it won't have immediate impact on a wide *eLife* audience of biologists, although biologists should be able to get the gist. I expect there might be a discussion amongst reviewers and editors of whether it needs to be buttressed with more explicit biological relevance and biological results. My view is no. I believe this manuscript will be important and influential as it stands. I would compare it to the 1994 Anders Krogh and David Haussler JMB paper that introduced profile hidden Markov models, or the 2009 Martin Weight and Terry Hwa paper that introduced Potts models. These papers ignited new areas of computational biology research that later led to important biological results. The Krogh paper was weakened and diluted in its review process by JMB reviewers and editors asking for more biology. Many of us in the field worked from a pre-dilution preprint more than the final published JMB version. I would advise not similarly diluting this strong manuscript. *eLife* should publish mathematical/technical papers occasionally, when the math is well-principled and clear, and where there's a good chance of founding a new important direction in biological data analysis. This manuscript is a nice example.

- At the beginning of Results, it would help to see even more intuitive rationale for the strengths of RBMs over Potts models, before diving in. For example, Materials and methods has an important line 'RBM<s> offer a practical way to go beyond pairwise models without an explosion in the number of interaction parameters'. Move this line up and clarify. I got a fair way into the paper thinking that RBMs would have to give up detailed pairwise correlation structure, in return for capturing phylogenetic and functional many-residue correlation structure.

- One weakness of the paper is that it does not fully describe the method. In particular, I don't know how to calculate the partition function from reading this paper; I would have to refer to other cited work (Tieleman, 2009; Fischer and Igel, 2012), which might be fair enough. However, Materials and methods says that 'practical details… including several improvements to the standard Persistent Contrastive Divergence algorithm… will be presented elsewhere'. If those details are essential for the results, they need to be in the paper somewhere.

- '(precise Github address here)': I assume that's an oversight, and that code will be provided. Indeed that would go a long way towards addressing the above weakness, as far as I'm concerned.

*Reviewer #3:*

This manuscript describes a novel generative machine learning model called RBM (Restricted Boltzmann Machines) for multiple protein sequence alignment (MSA). Through RBM, a protein sequence can be generated from some hidden nodes trained from the MSA. A unique feature of this RBM model is that each hidden node may represent a small set of residues which may not be adjacent to one another along the primary sequence, but may form contacts or a functional motif. This is very different from the widely-used Potts model, in which mainly pairwise residue correlation is considered. The authors have shown that their RBM model can be used to detect motifs and inter-residue contact prediction.

A major concern is that the authors tested their RBM model on only 4 natural proteins with solved structures and good functional annotations. By testing on very few proteins with solved structures, it is hard to convince the readers that the authors did not overfit their model. There are a few ways to address this issue:

1) Apply the model to analyze proteins without solved structures and then verify their results after the solved structures are available. For example, the authors may test their methods through the online benchmark CAMEO for protein contact prediction. Each week CAMEO will send out some test sequences for prediction. The native structures of these test proteins are not publicly available during the test, but will be public after 4-5 days.

2) The authors may also test their method on a large set of proteinswith solved structures for contact prediction, but using a simple way to choose M (the number of hidden nodes) and the regularization factor for all the test proteins. The authors may compare their result with currently popular contact prediction methods such as Evfold and CCMpred. It will also be interesting to study the relationship between accuracy and the number of effective sequences in MSA.

---

## [Author Response]

1) There is a difference of opinion among the reviewers about whether your methodology needs to be compared with other protein structures, and in particular examples that are not so well known and annotated. However the consensus opinion is that the manuscript would benefit from additional comparisons. Reviewer 3 explicitly calls out the possibility that there is overfitting involved, and it is hard to dismiss this out of hand without more effort to compare to the types of examples that they mention. Reviewer 3 makes reasonable suggestions of other examples you might compare to – and I look forward to seeing how these work in the revised manuscript.

To answer the concerns about overfitting, we have followed the suggestion of reviewer 3, and have further tested our modeling approach on 16 more families, used as a benchmark for contact predictions by plmDCA (an important inference method in the context of Direct-Coupling Analysis – DCA) in the standard 2014 paper by Ekeberg et al. This new analysis allows us to:

a) Carry out a statistical study of the performance of RBM with respect to DCA for contact prediction. The new panel in Figure 6 shows that the performances of RBM under fixed regularization and with a number of hidden units proportional to the maximal rank (M=0.3 R) are comparable with the ones obtained by plmDCA.

b) Compare the most significant weights (with largest norms) across the 16 families, and extract broad classes of structural and functional motifs repeated in many families, including the proteins we had presented in the first version of the manuscript. A new figure (Figure 9) has been added to report these results, accompanied by a discussion in the section “Cross- validation of the model and interpretability of the representations”.

We believe that these new results provide convincing evidence that our approach is not limited to a few “good” proteins or protein domains, but can be applied to any protein (with performances depending, of course, on the number of available sequences).

2) All of the reviewers have major comments for clarifications and addressing these will improve the manuscript. For example, reviewer 1 asks you to comment on the weights (how they are chosen, how they depend on model hyperparameters, etc.); Reviewer 2 points out that the methodology is not fully developed in this manuscript, but instead will be described elsewhere. The consensus view of the reviewers is that it is important for the methods to be fully described in this paper so that the results are reproducible by others, and a promise of publishing these elsewhere is insufficient.

We have added a detailed description of the main steps of the training algorithm in the Materials and methods section, as requested by reviewer 2. In particular, in the subsection "Learning Procedure", we now give the general structure of the equations for the gradient of the log-probability, the tricks used for moment evaluations, and the procedure followed to estimate the partition function. We also better present the main algorithmic innovations of the present paper, that is, the extension of RBM to multi-categorical Potts-like variables, and the introduction of general non-quadratic hidden-unit potentials, interpolating between quadratic, double-well (Bernoulli) and Rectified Linear Unit potentials (with hard-wall constraints). Readers should have no problem in understanding our code for training RBM, which is made available here, together with the scripts necessary to reproduce the figures in our manuscript. In addition, we have expanded the paragraph entitled “Learning” in the main text (section “Restricted Boltzmann Machines”); this paragraph lists the main ingredients of our training procedure, and is intended for readers not interested in the technical details presented in the Materials and methods section.

We have also added a discussion on the comparison of RBM with the Hopfield- Potts approach, another procedure for extracting simultaneously structural and functional information from sequence data proposed by two of us and collaborators in 2013, as required by reviewer 1. We have better developed the discussion on how the hyperparameters of the RBM should be chosen to learn efficiently interpretable features. Following reviewer 1’s request, we now explain how weights are selected, and study the stability of the extracted features under undersampling.

3) There are other explicit comments included in these reviews about ways of clarifying the manuscript that I trust you will address in a revised version.

Reviewer #1:

[…] Within the main text, the authors cite their previous publication, from 2013, in which they propose the Hopfield-Potts model as a means of naturally interpolating between PCA-based approaches to identifying functionally important groups of residues, and DCA-type approaches. There they also observed sparse modes (or patterns) that correspond to contacts in 3D structure, and in addition less sparse modes that correspond to features such as conserved sequence positions that form spatially connected and functionally important regions in the folded protein that are reminiscent of protein sectors, or positions at which many gap symbols are found in the alignments.This makes me question whether there is really no framework that can extract both structurally and functionally important features of a protein family. I think the authors should be careful to accurately place this manuscript in the context of prior work. Moreover, there appear to be some similarities between the energy function of an RBM, and that of a Hopfield model – this would be an interesting point for the authors to discuss.

We fully agree that the Hopfield-Potts model (Cocco, Monasson and Weigt, 2013) was an attempt to extract simultaneously structural and functional information from sequence data, and, in this regard, is conceptually similar to RBM (with the notable exception that the idea of representation was essentially lacking). Actually, the connection between RBM and Hopfield-Potts model is quite deep, as the latter is a particular case of the former, obtained when the hidden-unit potentials are quadratic functions of their arguments h. This connection, which is clearly emphasized in the new manuscript (see section “Restricted Boltzmann Machines”, in between Equations 5 and 6) was, indeed, a major incentive for us to start studying RBM. Some of the results found with RBM, such as very localized modes on two sites coding for contacts or extended modes due to stretches of gaps at the sequence extremities, were already found with Hopfield-Potts models. But, despite those similarities, the work on the Hopfield- Potts approach cited above suffered from several drawbacks (which we feel free to list, being among the authors of this work…):

1) the patterns (here, the weights) w were inferred within the mean-field hypothesis, known to be fast but inaccurate. In particular, 2-point statistics are not reproduced. As a consequence, the inferred Hopfield-Potts model cannot be used to generate new sequences in practice; it is simply too poor an approximation of the underlying sequence distribution.

2) the quadratic nature of the interactions made the patterns not uniquely defined, as they could be rotated in the mu-space with no change on the probability distribution of the sequences. To avoid this problem, we artificially imposed orthogonality between the weight vectors. There was (and there is) no justification for this procedure. The same problem arises in Independent Component Analysis: it is well known that identification of independent components cannot be done based on quadratic moments only, and requires knowledge of higher-order moments (captured in RBM by non-quadratic potentials).

3) the importance of sparsity was not recognized. While some inferred patterns were sparse in practice (such as the ones corresponding to contacts), most were not (see middle panel in Figure 2 of the above-mentioned paper) and could not be interpreted.

We have added in Appendix 1 several figures (for Kunitz, WW, and LP; results for HSP70 are given in a separate file in Appenndix 1) to illustrate these points, and showing in particular:

1) Some of the patterns inferred for the protein families presented in the main text. Low-eigenvalue patterns are sparse (as reported in the 2013 PLoS Comp Bio paper), but high-eigenvalue patterns coding for collective modes are extended, and therefore hard to relate to function.

ii) Contact predictions with the Hopfield-Potts model, showing worse performances than RBM (for the same number of hidden units).

iii) Benchmarking of generated sequences with Hopfield Potts on Lattice Proteins, similar to what is shown in Figure 7F for RBM. With a small pseudo-count, sequences generated by the Hopfield-Potts model are very bad. Using a larger pseudo-count, sequences have decent folding probabilities, but very low diversity (in agreement with findings by Jacquin et al., 2016).

Our current work on RBM does not suffer from any of these problems, and we are convinced that RBM are much more accurate and controlled ways to infer sequence motifs than the Hopfield-Potts approach. Yet, we agree with reviewer 1 that proper acknowledgements about previous works are required, and we have modified the Discussion accordingly.

It would be interesting to understand from the main text why the authors feel that the modeling assumptions of the RBM are applicable to protein sequence data.

RBM are natural candidates to interpolate between PCA-based and DCA-based approaches are they can be thought as either a method for extracting modes/representations (which goes beyond PCA) or an effective way to model complex (pairwise and higher-order) interactions between residues (which extends DCA). We have added a sentence in the section on RBM to explain this important point.

My major comment is that the new manuscript focuses on a rather detailed description of how a small number of RBM weights correspond to structurally and functionally important sequence positions. Almost no time at all in the main text describing how they were able to efficiently learn RBM from sequence data, which seems to be the novel contribution of this work.

As reviewer 1 emphasizes, we primarily aimed at convincing readers that RBM could learn structurally and functionally relevant motifs in protein sequences. This is made possible by:

1) the introduction of parametric class of non-quadratic potentials, called double Rectified Linear Unit potentials, which spans a large variety of behaviors, allowing hidden units to have smooth, bimodal, or constrained (e.g. positive- valued) conditional probabilities. We have added a formula (numbered 6) for dReLU potentials in the main text.

2) the introduction of the so-called L1/L2 sparsity regularization, which is an adaptive L1 penalty on the weights. This regularization is briefly described in the Results sections, and details are given in the Materials and methods section. We also discuss at length in the section "Cross-validation of model and interpretability of representations" how the choice of the regularization strength is crucial to obtain meaningful weights.

3) the selection of hyperparameters (in particular, the number of hidden units) and how they affect performance (for instance, for contact prediction) is also discussed in the main text and in Appendix.

The authors also do not provide any comparison between the weights of RBM and patterns learned from sequence data in prior work, such as their own 2013 paper. Specifically, did the patterns derived from the Hopfield-Potts model differ significantly from those found in the RBM weights? A thorough comparison to prior work is necessary to distinguish the advances made in this work.Beyond this, I would ask the authors to at least comment on how the diverse weights are chosen?

The weights shown in Figures 2, 3, 4 were selected manually based on several criteria:

i) Weight norm, which is a proxy for the importance of the corresponding hidden unit. Hidden units with larger weight norms contribute more to the likelihood, whereas low-weight norms may arise from noise/overfitting.

ii) Weight sparsity. Hidden units with sparse weights are easier to interpret in terms of structural/functional constraints.

iii) Shape of input distribution. Hidden units with bimodal input distribution separate the family in two subfamilies and are therefore potentially interesting.

iv) Comparison with available literature, in particular mutagenesis experiments.

v) Diversity of features in the weights (many weights correspond to contacts, and we do show all of them).

We have made these criteria explicit in the manuscript.

The weights shown in Figure 9 (one for each one of the 16 families) were picked up arbitrarily among the top (having the largest norms) 10 weights of each family to illustrate the different broad classes of motifs. A large scale, statistical approach would be necessary to better define those classes, as underlined in Discussion.

And what happens to the weights as the number of hidden units in the model is varied.

At low number M of hidden units (and fixed large regularization), the weights are not sparse, which is similar to what is found with PCA or Hopfield-Potts. As M increases, weights get sparser, as more hidden units are available to code for the various statistical features of the data. Above a certain value of M, the weights stabilize, and extra hidden units are essentially copies of the previous ones. The results shown here correspond to after the compositional regime emergence, prior to the ‘copy’ regime.

Moreover, it would be interesting to understand how the weights change as the number, and diversity of sequences in the protein family alignment is varied.

To show how the weights change with sampling, we have trained again our RBM after random removal of 50% the sequences in the MSA of the WW domain. We compare in Appendix 1—figure 13 the inferred weights to the ones obtained with the full MSA and shown in Figure 3B. We find that the weights are quite robust despite the removal of half of the data. Not surprisingly, weights with small norms (and not selected according to criterion i) above) are more sensitive to sampling noise.

The question on diversity is related to the presence of evolutionary correlations between sequences; the less diverse the sequences are, the stronger should be the reweighting factor, see point below.

In addition, the authors pay only passing attention to the fact that the protein sequences contained in the alignment are chosen uniformly at random from some underlying distribution, but rather are biased by their shared evolutionary history. The authors should provide some insight into how this affects the ability of the model to generalize.

Reviewer 1 is right that the sequences sampled in databases are not independent. This problem is shared by all sequence analysis methods such as Hidden Markov Models or DCA. We treat this dependence here in a rough way: each sequence S is assigned a weight factor (subsequently used for computing averages over data) proportional to the inverse of the number of neighbor sequences S’ in the alignment; S’ and S are considered as neighbors if their Hamming distance is smaller than a cut- off, generally 20% of the sequence length. This flattening procedure is known to improve contact prediction with DCA.

For RBM, this flattening reduces overfitting as well. To see this, we have repeated the training on the WW domain sequence data without reweighting (or regularization). We have generated sequences and plotted their log-probabilities vs. their distances to closest sequences in the MSA (same as Figures 5E and F). Without regularization or reweighting, the generated sequences are closer to the set of natural sequences, and their log-probability is higher, meaning that they are less diverse. We have added a comment in the "Cross-validation…" section and the corresponding figure in the Appendix.

We stress that reweighting is not necessary for *well* sampled sequences, i.e. for rather uniformly distributed sequences. In particular, for Lattice Proteins where we have full control and can sample sequences by Monte Carlo at equilibrium, reweighting does not improve performance, e.g. for contact prediction. Conversely, when the sampling of sequences is biased around a given (wild type) sequence, reweighting is needed to get back to good performance levels, see Jacquin et al., 2016.

Reviewer #2:

[…] At the beginning of Results, it would help to see even more intuitive rationale for the strengths of RBMs over Potts models, before diving in. For example, Materials and methods has an important line 'RBM<s> offer a practical way to go beyond pairwise models without an explosion in the number of interaction parameters'. Move this line up and clarify. I got a fair way into the paper thinking that RBMs would have to give up detailed pairwise correlation structure, in return for capturing phylogenetic and functional many-residue correlation structure.

We agree that this sentence was unclear. As one key advantage of RBMs with respect to Potts models is that they can capture higher-order interactions, we have now expanded this notion in two places in the manuscript. First, we have expanded the section on the distribution of sequences in the section “Restricted Boltzmann Machines”; we now explain that RBMs reduce to pairwise interaction models when the hidden-unit potentials are quadratic, and offer a possibility to take into account higher-order interactions with non-quadratic potentials. We have introduced a new formula (numbered 6) to explicitly show the class of (dReLU) potentials considered in our work and fitted from the sequence data. Secondly, we come back on this notion in the Discussion and explain why RBM are efficient ways of introducing high-order interactions (whose number is a priori exponential in the sequence length N) from a limited number of parameters (the weights w).

- One weakness of the paper is that it does not fully describe the method. In particular, I don't know how to calculate the partition function from reading this paper; I would have to refer to other cited work (Tieleman, 2009; Fischer and Igel, 2012), which might be fair enough. However, Materials and methods says that 'practical details… including several improvements to the standard Persistent Contrastive Divergence algorithm… will be presented elsewhere'. If those details are essential for the results, they need to be in the paper somewhere.

We agree with reviewer 2 that the paper should be self-contained. We have therefore added a detailed description of the main steps of the training algorithm in the Materials and methods section. In particular, in the subsection "Learning Procedure", we now give the general structure of the equations for the gradient of the log-probability, the tricks used for moment evaluations, and the procedure followed to estimate the partition function (with a paragraph explaining the principle of the Annealed Importance Sampling algorithm). We also better present the main algorithmic innovations of the present paper, that is, the extension of RBM to multi-categorical Potts-like variables, and the introduction of general non-quadratic hidden-unit potentials, interpolating between quadratic, double-well (Bernoulli) and Rectified Linear Unit potentials (with hard-wall constraints). Readers should have no problem in understanding our code for training RBM, which is made available here, together with the scripts necessary to reproduce the figures in our manuscript. In addition, we have expanded the paragraph entitled “Learning” in the main text (section “Restricted Boltzmann Machines”); this paragraph lists the main ingredients of our training procedure, and is intended for readers not interested in the technical details presented in the Methods section.

- '(precise Github address here)': I assume that's an oversight, and that code will be provided. Indeed that would go a long way towards addressing the above weakness, as far as I'm concerned.

Yes, all the codes (training algorithm and scripts for reproducing the figures of the paper) are provided.

Reviewer #3:

[…] A major concern is that the authors tested their RBM model on only 4 natural proteins with solved structures and good functional annotations. By testing on very few proteins with solved structures, it is hard to convince the readers that the authors did not overfit their model. There are a few ways to address this issue:1) Apply the model to analyze proteins without solved structures and then verify their results after the solved structures are available. For example, the authors may test their methods through the online benchmark CAMEO for protein contact prediction. Each week CAMEO will send out some test sequences for prediction. The native structures of these test proteins are not publicly available during the test, but will be public after 4-5 days.2) The authors may also test their method on a large set of proteinswith solved structures for contact prediction, but using a simple way to choose M (the number of hidden nodes) and the regularization factor for all the test proteins. The authors may compare their result with currently popular contact prediction methods such as Evfold and CCMpred. It will also be interesting to study the relationship between accuracy and the number of effective sequences in MSA.

We understand the concerns about overfitting expressed by reviewer 3. Following his/her second suggestion we have further analyzed 16 protein families that were used to benchmark plmDCA by Ekeberg et al. in their original paper (Ekeberg, Hartonen and Aurell, 2014), which is exactly the same method as Evfold and CCMpred: Evfold directly uses the plmDCA code, and CCMpred is a GPU implementation of plmDCA.

This new analysis allows us to:

1) Carry out a statistical study of the performance of RBM with respect to DCA for contact prediction. The new panel in Figure 6 shows that the performances of RBM under fixed regularization and with a number of hidden units proportional to the maximal rank (*M=0.3 R*) are comparable with the ones obtained by plmDCA.

2) Compare the most significant weights (with largest norms) across the 16 families, and extract broad classes of structural and functional motifs repeated in many families, including the proteins we had presented in the first version of the manuscript. A new figure (Figure 9) has been added to report these results, accompanied by a discussion in the section “Cross-validation of the model and interpretability of the representations”.

We believe that these new results provide convincing evidence that our approach is not limited to a few “good” proteins or protein domains, but can be applied to any protein (with performances depending of course on the number of available sequences).